# Monitoring the record-breaking wave event in Melilla harbour (SW Mediterranean Sea)

Pablo Lorente[1], Marta De Alfonso[1], Pilar Gil[1], Fernando Manzano[1], Anna Magdalena Matulka[1], Begoña Pérez-Gómez[1], Susana Pérez-Rubio[1] and M. Isabel Ruiz[1]

[1]Puertos del Estado, Madrid, 28042, Spain

*Correspondence to*: Pablo Lorente (plorente@puertos.es)

**Abstract.** During the 4th-5th of April 2022, a record-breaking wave storm hit Melilla harbour (SW Mediterranean Sea) with the violent overtopping of breakwaters. This unprecedented episode was compared against the six most extreme events previously registered by Melilla coastal buoy during 2011-2022 to disentangle their common atmospheric driving

mechanisms. A dipole-like sea level pressure (SLP) pattern, characterised by two adjacent (northwestern) high and (southeastern) low pressure systems, induced intense easterly winds and high waves over the entire SW Mediterranean Sea. The 2022 record-breaking event differed from the rest in the much stronger SLP gradient (2 Pa·km$^{-1}$) and north-easterly winds (above 20 m·s$^{-1}$), which concurrently gave rise to a maximum significant wave height (SWH$_o$) and mean period (T$_m$) of 7.32 m and 9.42 s, respectively, beating previous historical records. The associated return period decreased from 53 to 25

years, which must be considered for updated security protocols and sound design of future port facilities. Hourly observations from Melilla tide gauge covering the 2011-2022 period were used to investigate the relationship between offshore energetic waves penetrating into the harbour and the sea state inside. The harbour agitation, which also reached a record-breaking value (1.41 m) during the storm, was proved to be modulated by both the offshore SWH$_o$ (correlation coefficient of 0.87) and T$_m$. The highest values of agitation (above 1 m) were registered for incident high waves coming from

the angular sector comprised between 50º and 70º (clockwise from true north) with T$_m$ and peak period (T$_p$) values above 7 s and 10 s, respectively. By contrast, the astronomical tide and the storm surge had negligible effects on harbour agitation during the seven extreme wave events. Infragravity waves, with periods between 30 s and 300 s and maximum values up to 0.58 m during the 2022 storm, were also detected within the harbour basins and exceeded previously reported peaks. The energy in the infragravity band (IGE) was significantly correlated (0.96) with an offshore forcing parameter proportional to

SWH$_o^2$·T$_p$, evidencing that energetic swell was responsible for the highest IGE values (above 2000 m$^2$·s). Furthermore, a 30-year (1993-2022) regional wave reanalysis was used to characterise the intra-annual variability of the 99$^{th}$ percentile of SWH$_m$ over the Alborán Sea at monthly timescale and identify the existence of trends. Results revealed that the intensity of extreme wave events impacting Melilla harbour and surrounding areas have increased for April while observed trends indicate a significant decrease of the 99$^{th}$ percentile of SWH$_m$ for June and October. Finally, outcomes from this work could

be useful to implement a multi-hazard early warning system and ad hoc mitigation plans within the harbour territory.

## 1 Introduction

Over recent decades, climate change and extreme weather events have attracted growing public concern and political attention due to its widespread detrimental impact on the marine environment and human wellbeing (Konisky et al., 2015). While the global ocean is already experiencing anthropogenic-driven variations such as gradual warming, acidification, and sea level rise (IPCC, 2022), sustained pressure from climate change is even more significant on semi-enclosed basins like the Mediterranean Sea (Chiggiato et al., 2023; Juza and Tintoré, 2021) and also on exposed sectors like harbour systems (Verschuur et al., 2023; Izaguirre et al., 2021).

The Mediterranean Sea has long been recognized as a vulnerable climate change hot spot (Tuel and Eltahir, 2020), seriously jeopardised by marine pollution episodes or litter accumulation (Soussi et al., 2020; Ramirez-Llodra et al., 2013). This region is often affected by marine heat waves, mass mortality events and violent hazards, ranging from storm surges and flash floods to rogue waves and Medicanes (Dayan et al., 2023; Clementi et al., 2022; Garrabou et al., 2022; Milglietta and Rotunno, 2019; Wolff et al., 2018; Cavaleri et al., 2012). Served as examples, the 2018 Medicane Zorbas or the 2020 Storm Gloria caused several casualties and multi-million damages in susceptible coastal areas (Lorente et al., 2021; De Alfonso et al., 2021; Pérez-Gómez et al., 2021; Sotillo et al., 2021; Scicchitano et al., 2021).

Consequently, there is an increasing awareness not only about the potential anthropogenic influence on the intensity of these extreme weather episodes (Eyring et al., 2021) but also about the unavoidable need of gaining deeper insight into the underlying physical processes, already identified as one of the World Climate Research Program's Grand Challenges (WCRP website). The accurate monitoring of extreme events is crucial to implement adaptation policies and adopt prevention strategies that should eventually result in the enhancement of coastal communities' resilience (Linnenluecke et al., 2012). In response to this requisite, successive editions of the Copernicus Ocean State Report initiative have traditionally placed special emphasis on the multi-parameter analysis of severe sea states previously occurred in the Mediterranean basin (Álvarez-Fanjul et al., 2022; Clementi et al., 2022; Giesen et al., 2021; De Alfonso et al., 2020; Berta et al., 2020; Bensoussan et al., 2019; Notarstefano et al., 2019; Kokkini et al., 2018).

Recent initiatives like ECCLIPSE (assEssment of CLImate change in Ports of Southwest Europe) Interreg Sudoe project (ECCLIPSE website) have focused on analysing the impact of climate change on seaports. Although this topic has historically received less consideration than the corresponding impact for beach systems (Sánchez-Arcilla et al., 2016a), the central role of ports in countries' growth and globalised economy have motivated a plethora of newborn studies (Portillo Juan et al., 2022; Izaguirre et al., 2021), some of them devoted to the Mediterranean Sea (Sierra et al., 2015 and 2017, Sánchez-Arcilla et al., 2016b). In this sense, one of the main objectives of ECCLIPSE was to establish the fundamentals of a climate change observatory for Spanish ports, aiming at monitoring essential ocean variables and gaining an holistic understanding of violent weather from its physical drivers to its impact on port operability and infrastructure. Climate-driven extreme coastal hazards have been acknowledged to impose heavy socio-economic tolls as port downtime leads to reduction

of safety levels and wide trade losses due to the interruption of both the maritime transport and global supply-chain networks (Verschuur et al., 2022).

Following the footprints of ECCLIPSE, this work attempts to characterise the record-breaking storm that hit the Alborán Sea (SW Mediterranean Sea, Figure 1a) with wave heights above 7 m during the $4^{th}$-$5^{th}$ of April 2022 and evaluate the energetic response of Melilla harbour basins (Figure 1b) under the penetrating wave action. Port operations were precautionary disrupted due to the prevailing harsh weather conditions and the violent overtopping of breakwaters. While one ship was evacuated from its berth and later sheltered at the lee of Ras Taksefi Cape (Figure 1b), structural damages were reported in

the seawall tip and in several boats and marina pontoons.

A retrospective comparison of the present study case against extreme wave events previously registered at Melilla coastal buoy (Figure 1b) during 2011-2022 was conducted not only to put the former into a broader historical context but also to disentangle their common driving mechanisms (i.e., the predominant atmospheric conditions at synoptic scale). The return period associated with these extreme wave episodes, which is defined as the average time interval between two consecutive

events exceeding a specific wave height value, was also calculated. This concept is often used in marine engineering for the design of port facilities and the identification of dangerous events, providing a means for rational decision making and risk assessment (Salvadori et al., 2013). For instance, harbour breakwaters are commonly designed to withstand 100-year return period metocean conditions without significant damage, while having service lifetimes of similar durations (Todd et al., 2012; Gutierrez-Serret et al., 2009).

Additionally, following the approach of Pérez-Gómez et al. (2021) for 2020 Storm Gloria, high frequency (2 Hz) sea level data and agitation observations provided by Melilla tide-gauge (Figure 1b) during 2011-2022 were used to investigate the relationship between offshore energetic conditions and the sea state within the harbour. Precise estimations of agitation (i.e., oscillations within the port due to wind waves) are essential for downtime analysis and efficient port management (Romano-Moreno et al., 2022). Equally, infragravity (IG) waves with periods between 30 s and 300 s (Bellafont, 2019; Elgar et al.,

1992; Munk, 1950) were examined since their presence in semi-closed ports of small to intermediate size (where the surface water area and depth are about 1-10 $km^2$ and 5-10 m, respectively) may cause excessive ship motions at berth and unacceptable forces on mooring lines and fenders that could result in ship collisions and significant damage to vessels and port facilities (Costas et al., 2022; Bellotti and Franco, 2011). Under adverse circumstances, IG waves can be highly amplified by the basin geometry due to resonant processes (commonly referred to as seiches), resulting in large water level

fluctuations and strong horizontal currents that disturb port operations (unsafe and inefficient cargo activities) and negatively impact on cost-time efficiency (López and Iglesias, 2014; Okihiro et al., 1993).

Finally, a 30-year (1993-2022) regional wave reanalysis product developed in the frame of the Copernicus Marine Service for the Mediterranean Sea was analysed to characterise the spatio-temporal variability of the long-term extreme wave climate along the Alborán Sea. The intra-annual variability of the $99^{th}$ percentile (P99 hereinafter) for the significant wave height

was examined over this subregion at monthly timescale to identify potential trends, thereby complementing similar studies

previously focused on the intra-seasonal (Barbariol el al., 2021) or the inter-annual (Zacharioudaki et al., 2022b; Morales-Márquez et al., 2020) climate variability of extreme waves in the entire Mediterranean basin.

This work is structured as follows: Section 2 outlines the observational and modelled data sources. Section 3 describes the methodology adopted. Results are presented and discussed in Section 4. Finally, principal conclusions are drawn in Section 5.

## 2 Data

All the observational and modelled datasets used in this study are briefly described below. Complementary information about them is gathered in Table 1 and Table 2.

### 2.1 In situ observational data

Although the two in situ instruments used in this work were deployed before 2009, the time span for the observational datasets was standardized to 2011-2022 for consistency reasons as the collection of directional wave data started on April 2010 (Table 2).

#### 2.1.1 Melilla coastal buoy

A Datawell scalar buoy was moored at 15 m depth in April 2008, close to Melilla harbour (Figure 1b). It was replaced in April 2010 by a Triaxys buoy able to provide directional information. This in situ device, operated by Puertos del Estado, collects hourly-averaged estimations of diverse wave parameters (product ref. no. 1 in Table 1), encompassing the significant wave height ($SWH_o$), maximum wave height ($MWH_o$), mean period ($T_m$), peak period ($T_p$) and incoming mean wave direction ($MWD_o$). The quality control applied to data time series, defined by the Copernicus Marine In situ Team (Copernicus Marine In situ Team, 2017), consisted of a battery of automatic checks performed to flag and filter inconsistent values. For the Mediterranean Sea, the spike test was based on the difference between sequential measurements of $SWH_o$, $T_m$, and $T_p$ so they were discarded, respectively, when the difference exceeded 3 m, 4 s and 10 s. Occasional gaps (not larger than 6 h) were linearly interpolated to ensure the continuity of the records.

#### 2.1.2 Melilla port tide-gauge

A radar tide-gauge, manufactured by Miros and operated by Puertos del Estado as part of its REDMAR network (Pérez-Gómez et al., 2008 and 2014), was deployed inside of Melilla harbour in October 2007 (Figure 1b). Quality-controlled 2 Hz sea level data (product ref. no. 2 in Table 1) contain information of sea level oscillations with periods above 1 s, capturing all sea surface height variability including waves, high-frequency sea level oscillations (HFSLO) and tides. Sea level oscillations with periods over 1 h were extracted using a 10th-order Chebyshev low-pass filter with a cut frequency of 1/3600, whereas wave agitation (with periods below 30 s) was obtained using an 8th-order Butterworth high-pass digital filter

with a cut frequency of 1/30. HFSLO (with periods between 30 s and 1 h) were obtained by subtracting the two previous time series from the raw 2 Hz data signal. Then, a simplified four band energy spectrum was also calculated to facilitate the understanding of the energy distribution in the HFSLO band: i) period between 30 s and 5 min (IG waves); ii) period between 5 min and 15 min; iii) period between 15 min and 30 min; iv) period between 30 min and 1 hour. For further details about the frequency-domain analysis (used to describe how energy is distributed among all frequencies and to determine the

most energetic frequency in an hourly basis) and time-domain analysis (used to determine the hourly amplitudes of the HFSLO: maximum -$HFSLO_{max}$- and average of the highest third heights -$HFSLO_{13}$-), the reader is referred to as García-Valdecasas et al. (2021). Finally, 20-minute estimations of $HFSLO_{max}$, $HFSLO_{13}$, IG wave energy (IGE) and agitation were subsampled at hourly intervals (Table 2) and examined to assess the impact of extreme wave storms inside the harbour. Likewise, hourly estimations of total water fluctuations, astronomical tides and storm surge component were qualitatively

analysed to infer any potential sea level rise that could take place simultaneously (or in close sequence) to the extreme wave storms.

## 2.2 Modelled data

The time span for the modelled datasets was standardized to 1993-2022 for consistency reasons (Table 2).

### 2.2.1 ERA5 reanalysis

ERA5 reanalysis (product ref. no. 3 in Table 1), which is generated by the European Centre for Medium-Range Weather Forecast (ECMWF), provides hourly estimates from 1940 onwards for a large number of atmospheric and oceanic parameters which are regridded, respectively, to a 0.25° and 0.5° regular grid. In this work, hourly maps of modelled sea level pressure (SLP), wind at 10 m height (W10) and significant wave height ($SWH_m$) were analysed at synoptic scale (19ºW-5ºE, 26ºN-56ºN) for the 1993-2022 period (Table 2) in order to disentangle the common atmospheric configurations

that drove the most extreme wave events registered by Melilla buoy.

### 2.2.2 Multi-year wave product

The multi-year wave product of the Mediterranean Sea Waves forecasting system (product ref. no. 4 in Table 1) is based on a WAM model suite that predicts hourly wave parameters at 1/24° horizontal grid resolution. The atmospheric wind forcing used in WAM model consists of hourly 0.25° horizontal resolution ERA5 reanalysis from the ECMWF. The multi-year

product consists of a reanalysis dataset (MED reanalysis hereinafter), which spans from 1 January 1993 to 31 December 2022, and an interim dataset covering the period after the reanalysis until one month before present. In the present work, only the MED reanalysis was used: hourly $SWH_m$ estimations were examined for the selected 30-year period (Table 2) to characterise the spatio-temporal variability of the long-term extreme wave climate affecting the Alborán Sea, in general, and specifically Melilla harbour area. Equally, hourly maps of propagation direction ($MWD_m$) were depicted to assess the

prevalent wave directionality during the extreme events.

# 3 Methodology

As not all extreme metocean hazards necessarily have destructive impacts on coastal areas, there is not a worldwide consensus on the protocol for their categorization (Radovic and Iglesias, 2018). In this work, the 99.9[th] percentile (P99.9 hereinafter) of $SWH_o$ for the 12-year time series (2011-2022) provided by Melilla coastal buoy was used as threshold to select and tag chronologically a manageable number of extreme wave events previously occurred. Once shortlisted, these episodes were characterized in terms of intensity (magnitude of diverse wave parameters) and duration (hours above the P99 of $SWH_o$), placing the focus on the joint occurrence of interconnected extremes that might exacerbate the coastal impact compared to individual hazards occurring in isolation. Complementarily, hourly maps of $SWH_m$ were depicted to explore if the extreme wave events shared similar synoptic features in terms of severity and spatial distribution.

In order to elucidate the potential existence of common driving mechanism, the predominant atmospheric conditions (in terms of SLP and W10) at synoptic scale that led to the record-breaking storm were retrospectively compared to those giving rise to previous extreme wave events. Additionally, the temporal distribution of extreme episodes affecting Melilla area was derived from the 12-year observational time series of $SWH_o$ and $T_m$ to elucidate if they showed a relevant preference for a specific stage of the year. The annual cycle was split into six evenly spaced 50-day intervals and a longer 65-day summertime interval that did not negatively impact on the consistency of the percentages of occurrence obtained as extreme wave events during summer remained marginal regardless of the interval length selected.

The return period associated with these extreme wave episodes was derived from hourly time series of $SWH_o$ for two different periods: 2011-2021 (before the record-breaking storm) and 2011-2022 (including the storm). To this purpose, we assumed:

i)  an exceedance threshold based on the 95[th] percentile (P95) value of the dataset following the approach proposed by Harley (2017) and Fanti et al. (2023) for coastal storm analysis.

ii) 5-day distance between two independent storms. Although there is some subjectivity in how a time series is partitioned into separate storms, the broadly accepted criteria states that the independence between consecutive events is achieved by imposing that storm peaks must be separated by a time period longer than 3 days, which is the average lifetime of extra-tropical cyclones (Trigo et al. 1999). For instance, the most intense activity period of Storm Gloria in the western Mediterranean Sea ranged between 20 and 23 January 2020 (Amores et al., 2020; Lorente et al., 2021). Since adjacent peaks separated by 5 days will correspond to waves generated from different low-pressure systems, meteorologically independent events were identified by applying a moving time window of 5 days length between consecutive storms, in accordance to Mackay and Johanning (2018a and 2018b).

The long-term extreme sea state was characterised by using the Peak over Threshold (POT) method (Goda, 1988) with the fitting of a three-Parameter Weibull probability distribution to the $SWH_o$ observations. The POT method is based on extracting, from the recorded time series, those individual storms which surpass the aforementioned exceedance threshold of $SWH_o$ in the peak of the storm and are not dependant upon another one due to their proximity in time. The three-parameter

Weibull distribution was computed following the approach proposed by De Alfonso et al. (2021) to obtain the return period
for the maximum $SWH_o$ registered during the selected extreme wave events.

Furthermore, the relationship between offshore sea states and IG waves within Melilla harbour was investigated. Here we focused on its most common type: those induced by the non-linear interactions between incident wind short waves (Belloti and Franco, 2011). While IG waves tend to go unnoticed to human perception in deep waters (heights of the order of few cm), they can abruptly increase near the coastline and even exceed 1 m (Aucan and Ardhuin, 2013), contributing
significantly to nearshore processes (beach erosion) and affecting coastal structures (Okihiro et al. 1993). Significant efforts have been previously devoted to analysing the connection between offshore wave parameters and IGE, either at the shore (in the form of run-up) or in the nearshore area (surf zone). While Guza and Thornton (1982) found that the IG component of wave run-up increased linearly with increasing offshore $SWH_o$, Stockdon et al. (2006) concluded that the IG component scaled better with $SWH_o \cdot L$ (where L represents the deep-water wavelength) and was actually independent of the foreshore
slope. In the same line, Senechal et al. (2011) reported that IG wave run-up during extreme storm conditions was significantly less scatter when correlated with $SWH_o \cdot L$ than with $SWH_o$ only. By contrast, Inch et al. (2017) reported that nearshore IG waves were best predicted using an offshore forcing parameter that is proportional to $SWH_o^2 \cdot T_p$. These contradictory findings reveal that further research on the subject is required and suggest that nearshore IGE is unlikely a function of any single environmental factor (Lashley et al., 2020).

While the four aforementioned field studies focused on low-to-mild-sloping sandy beaches, the present work attempts to relate IGE measured within a harbour with offshore wave parameters. To this aim, a rough approximation approach (based on three simplifications) was adopted:

i) local slope effects were not included, similarity to Stockdon et al. (2006).

ii) IGE registered at Melilla tide gauge was scaled with $SWH_o^2$, $SWH_o \cdot L$ and $SWH_o^2 \cdot T_p$ despite the fact that IGE is affected
by wave–structure interaction processes (diffraction and reflection, to name the main ones) which are not so relevant in open sandy beaches.

iii) Although Melilla coastal buoy is moored at 15 m depth (d), the deep-water approximation is broadly accepted since the relative depth (defined as d/L) is above 0.5 the 78% of the time during 2011-2022 (not shown). Therefore, the wavelength can be defined as $L=(g \cdot T_m^2)/2\pi$, where the gravity acceleration g is 9.8 m·s$^{-2}$. As a consequence, we can
derive from point ii) that IGE was scaled with $SWH_o^2$, $SWH_o \cdot T_m^2$ and $SWH_o^2 \cdot T_p$.

Additionally, HFSLO (with periods between 30 s and 1 h) and harbour agitation (with periods below 30 s) data recorded by Melilla tide-gauge during 2011-2022 were thoroughly examined. On one hand, HFSLO heights observed during the selected extreme events were categorized based on specific IG wave thresholds which are universally common to all locations (McComb et al., 2020; McComb, 2011). This approach is valid since spectra of the 2 Hz data (not shown), generated to
identify energetic sea level variability inside the port, were dominated by energy in the IG band during these storms. On the other hand, total seawater levels were examined to disentangle if they exerted a relevant role in the sharp increase of harbour agitation during the extreme wave events and if astronomical tides were thereby enhanced by storm surge effects. In this

context, connected extremes are of particular concern for harbour operability, as their individual effects may interact synergistically and cause more damage in port structures than isolated extreme events (Velpuri et al., 2023).

Finally, potential long-term changes in the extreme sea state climate during the 30-year period analysed (1993-2022) were assessed over the Alborán Sea. As a preliminary step, the accuracy of MED reanalysis was evaluated at the grid point 2.916ºW-35.354ºN (denoted with a green rectangle in Figure 1b) closest to Melilla coastal buoy and located at a distance of 3450 m. Concurrent estimations of hourly $SWH_o$ and $SWH_m$ were compared for the period 2011-2022 and the best linear fit of scatter plot was computed. The statistical metrics used in the present study to compare two data sets included the mean,

the standard deviation, and the Pearson correlation coefficient (Emery and Thompson, 2001). Afterwards, maps of linear trend for the P99 of $SWH_m$ were obtained over the entire Alborán Sea at monthly timescale. The attention was particularly focused on the intra-annual variability in order to complement prior research dealing with intra-seasonal and inter-annual variability of extreme waves in the entire Mediterranean basin (Amarouche et al., 2022a; Barbariol el al., 2021; Zacharioudaki et al., 2022b; Morales-Márquez et al., 2020). The presence of temporal trends in the P99 of $SWH_m$ time series

was evaluated with two well-known non-parametric tests, which have been recently documented as the most used for trend detection in the Mediterranean Sea (De Leo et al., 2023):

i)    trends were calculated using the Sen´s slope estimator of P99 because it is not subject to the influence of extreme values (outliers) and therefore is more consistent than simple linear regression methods (Sen, 1968). Although P95 is also commonly used (Fanti et al., 2023), P99 was selected as reference percentile for the most extreme wave events affecting

Melilla area, in agreement with previous approaches reported in the literature (Zacharioudaki et al., 2022b, Barbariol et al., 2021).

ii)   the statistical significance at the 90% confidence interval was assessed at each grid point with the Mann-Kendall test (Mann, 1945; Kendall, 1962), in accordance with similar works previously published (Caloiero and Aristodemo, 2021; Barbariol et. al, 2021). Afterwards, a specific subdomain (2.70ºW-3.00ºW, 35.02ºN- 35.48ºN) in the vicinity of Melilla harbour was selected and the statistical significance was spatially-averaged to infer if this area is affected by meaningful

trends.

## 4 Results

### 4.1 Extreme events analysis

The P99.9 of $SWH_o$ (set to 4.45 m and derived from the 12-year time series provided by Melilla coastal buoy) was used as

threshold to detect the most extreme wave events (Figure 1c). Seven storms were identified and tagged chronologically from E1 to E7. They presented values ranging from 5.05 m (E3) to 7.32 m (E7), as shown in Table 3. The associated $T_m$ values, which ranged from 6.83 (E2) to 9.42 s (E7), surpassed the P99 (set to 6.25 s, Figure 1d). The seven episodes also showed concurrent high values of $MWH_o$ and $T_p$, emerging in the ranges 6.83-12.11 m and 9.13-10.75 s (Table 3). The storm that hit Melilla harbour during the 4th-5th of April 2022 (E7) exhibited unprecedented values for each wave parameter: the peak of

SWH$_o$ (7.32 m) was coincident with the greatest values of MWH$_o$ (12.11 m) and T$_m$ (9.42 s), jointly beating all previous historical records (Figure 1, c-d). In terms of storm duration (Table 3), defined as the number of consecutive hours above the P99 of SWH$_o$ (set to 3.01 m), E1 and E6 were significantly shorter (<20 h) than long-lasting E2 and E4 events (>50 h). The duration of E3 and E5 (27-31 h) events can be considered similar to E7 (37 h). From a directional perspective, the prevailing incoming wave directions during 2011-2022 were NE (41%) and NE-E (43%), with an overall associated mean value of 58º ± 37º (Figure 1e). These are the most common origins for waves recorded at Melilla coastal buoy due to its particular emplacement, sheltered to the east of Ras Taksefi Cape (Figure 1b). As a result, the shadow effect of this coastal promontory prevents the angular spreading of the storms coming from the westernmost sector. For extreme wave events with SWH$_o$ above P99 (3.01 m), the predominant incoming wave direction was NE-E with a 72% of occurrence, whereas the remaining 28% corresponded to the NE sector (Figure 1f).

Hourly maps of SWH$_m$ for E1-E6 events (Annex 1) and E7 (Figure 1a) shared common synoptic features such as the peak of SWH$_m$ (above 4.5 m) over the entire Alborán Sea. A secondary peak could be found over the Gulf of Cádiz for E1, E2, E4, E5 and E7 episodes, while E3 barely showed it. In the case of E6 event, the peak of SWH$_m$ over the easternmost part of the Alborán Sea was not so high (around 4 m) but affected broader areas of the SW Mediterranean Sea. The spatial patterns of SWH$_m$ and MWD$_m$, zoomed in the surrounding areas of Melilla harbour (small maps exposed in the lower right corner of each panel of Annex 1) revealed a similar visual resemblance for SWH$_m$ and a uniform MWD$_m$ field from the NE. The record-breaking E7 event stood out from the rest due to the severity of the storm, with SWH$_m$ above 5.5 m over the entire Alborán Sea (Figure 1a) but also in the vicinity of Melilla harbour (Figure 1b).

### 4.2 Return period analysis

For the period 2011-2021, the entire hourly time series of SWH$_o$ was fitted to a three-parameter Weibull distribution, leading to return periods of 3.25-4.51 years for the extreme wave events E1 to E6 (Table 4). Notwithstanding, the E7 event was associated with a 53-year return period which highlights the extraordinary magnitude of this twice-in-a-century high-impact episode. For the period 2011-2022, which already included the record-breaking E7 storm (April 2022), a new fitting of the three-parameter Weibull probability distribution to the SWH$_o$ observations was performed and the associated Weibull parameters (threshold, scale and shape) were updated (Table 4). Results revealed that the return period related to E1 to E6 events decreased by 17-22% to 2.69-3.51 years, while the updated E7 return period dropped by 53% from 53 years to 25 years. These relevant outcomes should be applicable in the design and construction of new facilities at Melilla harbour and also integrated into the port operations planning and day-to-day logistics activities.

### 4.3 Driving atmospheric conditions

The prevailing atmospheric conditions at synoptic scale during the seven extreme wave storms were inferred from the ERA5 reanalysis of SLP and W10. The SLP map for E7 event (Figure 2a) exhibited the so-called hybrid Rex block (Sousa et al., 2021; Lupo, 2021; Rex, 1950), a large-scale blocking pattern characterized by two adjacent (northwestern) high and

(southeastern) low pressure systems. This type of blocking is usual during the transition phase from an Omega block (midlatitude high-pressure centre surrounded by two low pressure systems on its western and eastern flanks) to a pure Rex shape (a north–south dipole pattern of SLP). Blocking episodes in Europe have been long acknowledged as persistent

atmospheric disturbances that can lead to weather extremes (Kautz et al., 2022). As a consequence, this dipole was visible for the whole investigation period, whereas it followed a clockwise rotation. The derived pressure gradient (above 2 Pa·km$^{-1}$) gave rise to very strong north-easterly winds (above 20 m·s$^{-1}$) that affected broad areas of the SW Mediterranean and Alborán Seas, while extremely intense easterlies were channelled through the Strait of Gibraltar due to its specific geometric configuration (Figure 2b). In the Gulf of Cádiz (denoted in Figure 1a), the wind field exhibited a counterclockwise rotation

around the low-pressure core.

The analyses of the six previous extreme events revealed that all of them shared very similar meteorological conditions: i) a northwestern-southeastern hybrid Rex pattern of SLP anomalies (Annex 2), in contrast to the climatological mean (Figure 2c) that shows two well-known semi-permanent pressure systems (i.e., the Azores High at middle latitudes and the Icelandic Low at subpolar latitudes); ii) a peak of wind speed (> 15 m·s$^{-1}$) over the entire Alborán Sea, where easterlies blew strongly

along both sides of the Strait of Gibraltar (Annex 3). Only the event E6 showed a slightly different structure (Annex 3f), with moderately strong winds (13-15 m·s$^{-1}$) blowing from the NE and massively affecting the entire western Mediterranean Sea. In terms of persistence, intense winds steadily affected the study area for 1-2.5 days, except in the case of E1 and E6 events where the duration was shorter (14-16 h), as derived indirectly from the time that the $SWH_o$ consecutively exceeded the P99 (Table 3).

The primary factors that jointly triggered the record-breaking E7 wave storm were the short distance (1400 km) between the two main pressure systems along with the relatively deep (below 1000 hPa) system of low pressures over the Gulf of Cádiz (Figure 2a). The resulting SLP gradient was anomalously powerful (above 2 Pa·km$^{-1}$), leading to very strong easterlies (up to 20 m·s$^{-1}$, as shown in Figure 2b) that ultimately induced high (~6 m) waves over the entire Alborán Sea (Figure 1a).

The previous six episodes also presented intense (albeit 25-50% weaker) SLP gradients, ranging from 1.01 Pa·km$^{-1}$ (E4,

Annex 2d) to 1.48 Pa·km$^{-1}$ (E6, Annex 2f), due to the usually longer distances (ranging from 1900 km to 3000 km) comprised between both pressure systems (Annex 2). Although the E1 event exhibited SLP cores with similar separation (1438 km, showed in Annex 2a), the low-pressure system was not so deep (1016 hPa), in contrast to the E7 event where minimum SLP values dropped below 1000 hPa (Figure 2a).

Finally, it should be noted that the seven extreme episodes took place during the same stage of the year, a 50-day period

between late February and early April (Figure 2d and Table 3). Therefore, it might be deduced that large-scale atmospheric blocks leading to severe sea states (above the P99 of $SWH_o$ and $T_m$) in Melilla tend to be more probable during the winter-to-spring transition period, in agreement with previous blocking climatologies for the eastern North Atlantic (Kautz et al., 2022; Barriopedro et al., 2006).

## 4.4 Sea state within the port

An accurate estimation of the historical harbour wave agitation is fundamental for many practical applications such as port downtime analysis (Romano-Moreno et al., 2022). The analysis of hourly time series of agitation provided by Melilla tide gauge revealed that there was a record-breaking value during E7 event (1.41 m, Figure 2e), while the six previous events also exceeded the P99.9 threshold (0.56 m, Figure 3a). The agitation response is usually determined by wave penetration into the harbour arising from the combination of diverse parameters: $SWH_o$, $T_m$, $MWD_o$, astronomical tide and storm surge

outside the port (Camus et al., 2018). As shown in Figure 2e and Annex 4, the impact of the last two elements on harbour agitation during the seven extreme events was negligible due to a number of factors, namely: i) Melilla harbour waters are characterized by a maximum tidal range of 0.40 m; ii) for each extreme event, the evolution of harbour agitation was independent from the tidal phase as the peak of agitation was not coincident with high tides; iii) during E7, the low-pressure core (~1000 hPa) was located in the Gulf of Cadiz (western side of the Strait of Gibraltar, Figure 2a) so the storm surge

affecting Melilla harbour was small (~5 cm, Figure 2e); iv) during the previous six extreme events (E1-E6), the meteorological residual was even negative (Annex 4), ranging from -2 cm (E3) to -14 cm (E2).

Hourly scatter plots evidenced the strong relationship between the agitation inside the port and the wave conditions outside the port registered by Melilla coastal buoy (Figure 3, b-d). The best linear fit of scatter plot between the agitation and $SWH_o$ revealed a significantly high correlation coefficient (0.87). During the 12-year period analysed (2011-2022), there were 967

hourly agitation values above the P99 threshold (0.36 m): the 89% of them were associated with waves coming from the predominant sector comprised between 50º and 70º (clockwise from true north), while 6% of them were related to incoming waves with angles emerging from 70º to 90º (Figure 3b). The remaining 5% was assigned to waves with an angular spread ranging from 30º to 50º. Therefore, the overall agitation is direction-dependent due to the harbour orientation (Figure 1b) and its inherent structural design (mouth width, port layout configuration, etc.). Additionally, harbour agitation was also

importantly modulated by offshore period, as shown in Figure 3 (c-d). Agitation values above the P99 were generally observed when $T_m$ and $T_p$ values were above 4 s and 6 s, respectively. Equally, the highest values of agitation (above 1 m height) were associated with $T_m$ and $T_p$ values above 7 s and 10 s, respectively. It seems reasonable to deduce that the record-breaking harbour agitation (1.41 m) registered during E7 event was caused by the combined effect of unprecedented values of $SWH_o$ (7.32 m), $MWH_o$ (12.11 m) and $T_m$ (9.42 s) in tandem with a very high value of $T_p$ (10.75 s) and a $MWD_o$

(55º) comprised within the predominant angular sector (50º-70º) previously mentioned.

Operational thresholds in the IG band, which are common to all locations, have been historically proposed for safe conditions during port operations (McComb et al., 2020; McComb, 2011). Since the spectra of 2 Hz sea level oscillations measured inside the harbour by Melilla tide gauge (not shown) revealed a high energy content in the IG band during the seven storms, $HFSLO_{13}$ values registered during the seven extreme events (which contained not only the predominant

contribution of oscillations in the IG band but also of oscillations with periods between 5 min-1 hour) were categorized according to this methodology (Figure 3e). The exploration of hourly timeseries of $HFSLO_{13}$ showed that E1 and E6 events

surpassed 0.15 m threshold (denoted as "extreme caution" in Figure 3e), while the remaining five events exceeded also the "danger" threshold (0.20 m), with an unprecedented value of 0.31 m during the E7 episode. Likewise, hourly values of $HFSLO_{max}$ went clearly beyond 0.35 m during the extreme episodes, reaching the record-breaking value of 0.58 m during E7 event. Furthermore, IGE was scaled with $SWH_o^2$, $SWH_o \cdot T_m^2$ and $SWH_o^2 \cdot T_p$ (Figure 3, f-h). The best linear fit of each scatter plot showed very high correlations: 0.94, 0.93 and 0.96, respectively. Therefore, IGE was best predicted using an offshore forcing parameter that is proportional to $SWH_o^2 \cdot T_p$, in accordance with Inch et al. (2017). As expected, the highest IGE values (above 1500 $m^2 \cdot s$) were observed for energetic swell waves with $SWH_o$ and $T_p$ above 5 m and 10 s, respectively.

## 4.5 Trends in extreme wave climate

The evolution on the extreme wave conditions over the Alborán Sea during the 30-year period analysed (1993-2022) was assessed. As a preliminary step, $SWH_m$ estimation from MED reanalysis were compared against hourly in situ $SWH_o$ observations provided by Melilla coastal buoy during the concurrent 12-year period (2011-2022). To this aim, the MED reanalysis grid point (2.916ºW, 35.354ºN) closest to the moored buoy (located at a distance of 3450 m) was selected and both time series were compared. A significantly high correlation coefficient (0.96) for a set of 77100 hourly data was derived from the best linear fit of scatter plot (Annex 5a). Equally, the slope and intercept values were close to 1 (0.85) and moderately low (0.15), respectively.

These results revealed that MED reanalysis, albeit accurate in Melilla region, seems to underestimate $SWH_o$ , especially for extreme waves. Such systematic underestimation has been previously reported for the entire domain (Fanti et al., 2023; Zacharioudaki et al., 2022b) since shallow water processes cannot be properly captured by global and regional reanalysis because: i) the coastline and the bottom topography are not well resolved as the grid mesh is too coarse; ii) fetch limitations; iii) inherent uncertainties in the wind field used to force the wave model. These limitations are even more pronounced in regions with complex coastal configurations (sheltered by islands, headlands, and reefs) and in port-approach areas where sharp topo-bathymetric gradients pose special difficulties for accurate local predictions (Sánchez-Arcilla et al., 2016a). Nevertheless, according to Zacharioudaki et al. (2022b), the reanalysis skill can be considered robust and good enough to conduct further investigations about the wave climate affecting Melilla area and the related intra-annual variability in the Alborán Sea.

Thus, the monthly P50 and P99 of $SWH_m$ were computed over the entire Alborán Sea for the 1993-2022 period (Annex 5, b-e). In particular, we selected only April and July as representative months of the stormy and calm seasons, respectively. According to homogeneous spatial patterns of P50, the mean wave climate is rather similar for April and July, only differing in the magnitude: while April is characterised by a P50 slightly above 1.1 m over Alborán open waters (Annex 5b), P50 is around 0.7-0.8 in July (Annex 5c). By contrast, significant differences can be found in the most energetic sea states (Annex 5, d-e). In April, the P99 values around Melilla are up to 3 m, while they reach 4 m offshore (Annex 5d). Peaks of 4.3 m are attained in the easternmost sub-basin, probably as a consequence of strong easterly winds. On the contrary, during July the largest P99 barely reaches 3 m in the central part of Alborán Sea, while the spatial distribution of P99 generally remains

uniformly below 2 m in the rest of the spatial domain, including littoral areas and nearby regions of Melilla harbour (Annex 5e).

The climate variability over the Alborán Sea was assessed by analysing the intra-annual variations in the extreme $SWH_m$ conditions (Figure 4). Monthly trend maps of P99 were calculated for the period 1993-2022, revealing statistically significant changes in the vicinity of Melilla harbour for few specific months: while an increase of 2 cm·year$^{-1}$ was observed for April (Figure 4a), a downward P99 trend of 1.5-2 cm·year$^{-1}$ was detected for June (Figure 4b) and October (Figure 4c). The temporal trends for each month (Figure 4, d-f), computed over the subdomain surrounding Melilla harbour (black box in Figure 4, a-c), visually supported the previous statement: the trends were statistically significant at the 90% confidence interval for April, June, and October. By contrast, during both the second part of summer (July-September) and the transitional season (November - February), monthly maps of P99 trends (not shown) did not exhibit statistically significant values over the entire Alborán Sea. The trend map of P99 for March and May (not shown) showed large areas with positive trends and negative trends, respectively, but delimited over the easternmost part (2ºW-1ºW) of the Alborán basin.

The long-term changes detected in the extreme wave climate over Melilla are, to a certain extent, comparable to those previously exposed by Barbariol et al. (2021). Although the wave reanalysis used and its associated temporal coverage (1980-2019) were different, this previous work reported both an upward trend for the P99 of $SWH_m$ (about 0.8-1.2 cm·year$^{-1}$) and a non-significant trend in the vicinity of Melilla harbour for the extended winter (defined as NDJFM) and for summer (defined as JJA), respectively. From a broader perspective focused on the entire western Mediterranean Sea, Barbariol et al. (2021) also documented a relevant positive trend (1.2 cm·year$^{-1}$) during winter in the Gulf of Lyon (denoted in Figure 1a) due to strong north-westerly Mistral winds. By contrast, Amarouche et al. (2022b) examined a 41-year (1979-2020) hindcast database and determined that the west coast of Gulf of Lyon was affected by a significant upward trend for all seasons, with a considerable annual increase (4 cm·year$^{-1}$) of maximum values of $SWH_m$. Complementarily, Amarouche et al. (2022a) demonstrated significant decadal increases in wave storm intensity and duration not only over the eastern part of the Alborán Sea but also in the Balearic basin. All these findings highlighted both the existence of an inter-seasonal variability of P99 of $SWH_m$ and the importance of multi-temporal scales analysis.

**5 Conclusions**

Gaining a deeper, holistic understanding of extreme weather events and the related driving mechanisms has been identified as one of the World Climate Research Program's Grand Challenges (WCRP website) due to its detrimental impact on ecosystems health and societal assets (Hochman et al., 2022). Concerning the latter, climate-driven extreme coastal hazards have been long recognized to impose heavy socio-economic tolls, particularly aggravated in vulnerable semi-enclosed regions like the Mediterranean Sea and in exposed sectors like harbour systems (Verschuur et al., 2023).

As port downtime leads to reduction of safety levels and wide trade losses through maritime transport and global supply-chain networks (Verschuur et al., 2022), the accurate monitoring of violent weather-related episodes is decisive to adopt

prevention strategies (i.e., wise design of safe port infrastructures) and mitigation measures that should eventually result in the enhancement of coastal communities' resilience.

In the present work, the attention is focused on the unprecedented storm that hit Melilla harbour (Alborán Sea, Figure 1a) during the 4th-5th of April 2022 with heavy rainfall and strong easterly winds, which induced extremely high waves (above 7 m) with associated long mean periods (above 9 s) that simultaneously beat previous historical records (Figure 1, c-d). The return period associated with this extreme wave event decreased from 53 years to 25 years. These outcomes are essential for the safe design of future facilities at Melilla port (Naseef et al., 2019). Conversely, it is worth pointing out that the port is also subjected to a constant geometric modification (in the docks, basins, bathymetry, breakwaters, etc.) which in turn can induce additional variations in the port response to extreme wave events that should be further assessed.

The analysis of hourly time series of $SWH_o$ (2011-2022) revealed that there were seven episodes that exceeded the P99.9 threshold (4.45 m), denoted chronologically from E1 to E7 in Figure 1c. The retrospective comparison of the record-breaking E7 event against six previous extreme wave episodes (E1 to E6) revealed that all of them were connected with similar large-scale atmospheric blocks: a dipole-like SLP pattern, characterised by two adjacent (northwestern) high and (southeastern) low pressure systems, induced strong easterly winds channelled over the entire Alborán Sea (Figure 2 a-b, Annex 2 and Annex 3). Furthermore, this common atmospheric configuration seems to predominantly feature during the same stage of the year, a 50-day period between late February and early April (Figure 2d). These findings contrast with other Spanish harbours (i.e., NW Iberian Peninsula) where the storm season typically spans from November to March (Ribeiro et al., 2023), highlighting the strong need of conducting a tailored assessment for each specific port and oceanographic region. Therefore, it might be deduced that large-scale atmospheric blocks leading to severe sea states in Melilla tend to be more probable during the winter-to-spring transition period. This outcome is in line with prior blocking climatologies for the eastern North Atlantic (Kautz et al., 2022; Barriopedro et al., 2006). In this context, previous works have also explored the dynamical links between blocking patterns and the Nort Atlantic Oscillation (NAO), which is the leading mode of atmospheric circulation variability over the Euro-Atlantic sector and is characterized by a seesaw of atmospheric mass between the Iceland Low and the Azores High (e.g., Hurrell and Deser 2009). The NAO appeared as the leading variability pattern during winter, accounting for the 45% of the blocking frequency variance (Barriopedro et al., 2006).

High frequency (2 Hz) sea level and agitation observations during the 2011-2022 period, provided by Melilla tide gauge, were used to investigate the relationship between offshore energetic waves and the sea state inside of the harbour (Figure 3). A record-breaking value of harbour agitation (1.41 m) was recorded during the E7 event (Figure 3a). The highest agitation records (above 1 m) were registered for incident high waves coming predominantly from the sector comprised between 50º and 70º (clockwise from true north) with $T_m$ and $T_p$ values above 7 s and 10 s, respectively (Figure 3, b-d). Extreme sea level oscillations (30 s -1 h), which also reached record heights (up to 0.58 m), were linked to the highest values in the IG energy band (Figure 3e). The seven extreme events in the Alborán Sea led to harsh sea conditions within the port: the energy in the IG band was significantly correlated (0.96) with an offshore parameter proportional to $SWH_o \cdot T_p^2$, with energetic swell being responsible for the highest energies (above 2000 $m^2 \cdot s$), as shown in Figure 3 (f-h). Therefore, the IG waves related to

energetic swell commonly observed in the NW Iberian coast, can also be present during extreme wave events in the Mediterranean coast, as previously reported for the 2020 Storm Gloria by Pérez-Gómez et al. (2021) and Álvarez-Fanjul et al. (2022).

Additionally, MED reanalysis was used to characterise the long-term mean (Annex 5) and extreme (Figure 4) wave climate
over the Alborán Sea for the period 1993-2022. The intra-annual variability of the P99 of $SWH_m$ was examined at monthly timescale to identify the existence of potential trends. Results seem to suggest that the intensity of extreme wave events impacting Melilla harbour has increased for April (Figure 4a and 4d), while observed trends indicate a significant decrease of P99 for the $SWH_m$ during June (Figure 4b and 4e) or October (Figure 4c and 4f). Such alterations of outer-harbour wave climate conditions might impact on in-port wave agitation response as the amount of energy penetrating into the harbour
would be different, as previously indicated by Sierra et al. (2015).

Still, it should be noted that the present work does not focus on the duration of extreme wave events over the SW Mediterranean Sea, so future endeavours should address this relevant aspect to complement the results here presented. Moreover, long-term historical changes in wave period and directionality are receiving increasing attention and should be further analysed to assess their specific impact on harbours operability (Erikson et al., 2022; Casas-Prat and Sierra, 2012).
Permanent modifications in the wave direction might result in enhanced wave penetration into the harbour and thereby larger agitation as port protective structures were originally designed to dampen wind and short waves coming from a predetermined sector (Casas-Prat and Sierra, 2012). Likewise, offshore wave period also plays a primary role in the modulation of harbour agitation, as derived from Figure 3 (c-d). As a consequence, any sharp increase in both wave period and $SWH_o$ could lead to severe sea states within the port. Regardless of the reported limitations of global and regional
reanalyses (inherent to their coarse spatial resolution) when used at coastal and port scales (Fanti et al., 2023; Zacharioudaki et al., 2022b), the MED reanalysis used in this work can be considered a robust first-guess estimator for the present intra-annual variability assessment of extreme waves in Melilla. This statement is supported not only by the comprehensive Quality Information Document (Zacharioudaki et al., 2022a) but also by the 12-year skill assessment conducted against in situ hourly observations from Melilla coastal buoy (Annex 5a). The comparison yielded a correlation coefficient of 0.96 and
revealed a slight underestimation of extreme $SWH_o$ values. To overcome such a drawback, future works should include the implementation of a dynamical downscaling methodology to improve wave reanalysis accuracy at finer coastal scales (Vannucchi et al., 2021). Of course, this would necessarily require finding the right trade-off between adequate spatial resolutions and the available in-house computational resources. Complementarily, additional efforts should be devoted to assessing the dominant modes of extreme waves variability and their relationship with the most important climatic indices
since this could enhance the prognostic skills of extreme wave events and benefit the adaptation plans in the entire Spanish harbour system.

Finally, it is worth mentioning that most of the outcomes derived from this work could not only feed the incoming climate change observatory for the Spanish ports (which should be fully operational by 2025) but also be integrated into tailored multi-hazard early warning systems. They would act as a key component of robust capacity analysis frameworks, covering a

wide range of dimensions, such as legislative, planning, infrastructure, technical, scientific and institutional partnerships (Haigh et al., 2018). Special attention should be focused on the thorough revision of security protocols and the implementation of mitigation plans within the harbour territory based on the updated return periods presented in this work. The design lifetime risk should be recalculated accordingly as coastal structures in the vicinity of the harbour must resist growing stresses during their lifespan and operations, such as wave overtopping, floodings or resonance, to name a few. While the current port layout configuration must be adapted to the increasing frequency and magnitude of these stressors, future maritime facilities at Melilla harbour should be wisely designed and constructed taking into account these outcomes in order to withstand extreme wave regimes imposed by the changing marine environment (Vanem et al., 2019). Albeit methodologically robust, the return periods exposed in this work are based on short (12-year) time series of quality-controlled in situ wave observations. Therefore, they should be further complemented with return periods computed by means of longer modelled time series from very high-resolution wave reanalysis.

## Data availability

The model and observation products used in this study from both the Copernicus Marine Service and other sources are listed in Table 1.

## Author contributions

PL, MA, FM, BPG and SPR conducted the pilot study through fruitful discussions in the framework of working team meetings. PL: designed the experiment, analysed the long-term wave trends, created the figures, and prepared successive versions of the draft with inputs from several co-authors. MA: conducted a bibliographic revision of extreme metocean events previously occurred in the Mediterranean Sea. PG: computed the return period before and after the record-breaking event. FM: extracted time series from Puertos del Estado internal database and prepared diverse in situ sensors datasets. BPG: proposed the agitation and infragravity band study in the port, and analysed the corresponding tide-gauge records. SPR: provided a tailored coastline for Melilla harbour and analysed the atmospheric driving mechanisms during the event. MIR: applied a quality control for historical time series of wave parameters from Melilla coastal buoy. Finally, several authors participated in successive iterations, the drafting and revision of the manuscript.

## Competing interests

The contact author has declared that none of the authors has any competing interests. Disclaimer. Publisher's note: Copernicus Publications remains neutral with regard to jurisdictional claims in published maps and institutional affiliations.

## Acknowledgments

The authors are grateful to the Copernicus Marine Service for the data provision.

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

**Tables**


| Product ref. no. | Product ID & type | Data access | Documentation |
|---|---|---|---|
| 1 | INSITU_IBI_PHYBGCWAV_DISCRETE_MYNRT_013_033, in situ observations | EU Copernicus Marine Service Product (2022a) | PUM: In situ TAC partners (2022); QUID: Wehde et al. (2022) |
| 2 | 2 Hz data, high frequency sea level oscillations and agitation parameters from Melilla tide-gauge, in situ observations | Puertos del Estado websites: https://portus.puertos.es https://portuscopia.puertos.es/Catalog http://opendap.puertos.es/thredds/catalog/tidegauge_meli/catalog.html | Product description: García Valdecasas et al. (2021) https://bancodatos.puertos.es/BD/informes/INT_3.pdf |
| 3 | ERA5 global reanalysis, numerical models | Copernicus Climate Data Store: https://cds.climate.copernicus.eu/cdsapp#!/dataset/reanalysis-era5-single-levels?tab=form | Product description: https://confluence.ecmwf.int/display/CKB/ERA5%3A+data+documentation |
| 4 | MEDSEA_MULTIYEAR_WAV_006_012, numerical models | EU Copernicus Marine Service Product (2022b) | PUM: Denaxa et al. (2022); QUID: Zacharioudaki et al. (2022a) |

**Table 1. Products from the Copernicus Marine Service and other complementary datasets used in this study, including the Product User Manual (PUM) and QUality Information Document (QUID). For complementary datasets, the link to the product description, data access and scientific references are provided. Last access for all web pages cited in this table: 11 January 2024.**


| Source (Product ref) | Type | Location (coverage) | Variable (unit) | Temporal resolution | Time span used | Spatial resolution |
|---|---|---|---|---|---|---|
| In situ sensor (1) | Buoy | Coastal location (2.94ºW – 35.33ºN) | $SWH_o$ (m) | Hourly | 2011-2022 | Point-wise location |
|  |  |  | $MWH_o$ (m) |  |  |  |
|  |  |  | $T_m$ (s) |  |  |  |
|  |  |  | $T_p$ (s) |  |  |  |
|  |  |  | $MWD_o$ (º) |  |  |  |
| In situ sensor (2) | Tide-gauge | Port location (2.93ºW – 35.29ºN) | Agitation (m) | Hourly | 2011-2022 | Point-wise location |
|  |  |  | $HFSLO_{13}$ (m) |  |  |  |
|  |  |  | $HFSLO_{max}$ (m) |  |  |  |
|  |  |  | IGE ($m^2 \cdot s$) |  |  |  |
| Numerical Model (3) | ERA5 reanalysis | Regional domain (19ºW - 5ºE 26ºN - 56ºN) | SLP (Pa) | Hourly | 1993-2022 | 0.25º |
|  |  |  | W10 ($m \cdot s^{-1}$) |  |  | 0.25º |
|  |  |  | $SWH_m$ (m) |  |  | 0.5º |
| Numerical Model (4) | MED reanalysis | Subregional domain (6ºW - 1ºW 35ºN - 37ºN) | $SWH_m$ (m) | Hourly | 1993-2022 | 1/24º |
|  |  |  | $MWD_m$ (º) |  |  |  |

**Table 2. Complementary information about the data sources used in this study.**



| Event | Date (hour) | $SWH_o$ (m) | Time above P99 (h) * | $MWH_o$ (m) | Mean period (s) | Peak period (s) | Mean direction (°) |
|-------|-------------|-------------|----------------------|-------------|-----------------|-----------------|--------------------|
| E1 | 2016-02-21 (00) | 5.25 | 16 | 9.46 | 7.15 | 9.13 | 63 |
| E2 | 2017-02-21 (01) | 5.21 | 57 | 7.22 | 6.83 | 9.25 | 66 |
| E3 | 2017-03-15 (01) | 5.05 | 27 | 7.79 | 6.99 | 9.98 | 51 |
| E4 | 2017-04-21 (15) | 5.36 | 58 | 6.97 | 7.03 | 9.34 | 69 |
| E5 | 2019-03-27 (00) | 5.21 | 31 | 8.03 | 6.88 | 9.91 | 69 |
| E6 | 2021-03-20 (21) | 5.09 | 14 | 6.83 | 6.91 | 9.69 | 55 |
| E7 | 2022-04-04 (21) | 7.32 | 37 | 12.11 | 9.42 | 10.75 | 55 |

Table 3. Characterization of the seven most extreme waves event registered by Melilla coastal buoy during the 12-year period analysed (2011-2022). *Consecutive hours above the 99th percentile of $SWH_o$.

| Parameter | 2011-2021 | 2011-2022 | Decrease |
|-----------|-----------|-----------|----------|
| Weibull parameter: threshold (or location) | 1.82 | 1.88 | ------ |
| Weibull parameter: scale | 1.19 | 1.10 | ------ |
| Weibull parameter: shape (or Weibull slope) | 1.20 | 1.07 | ------ |
| Return period for events with $SWH_o$ = 3 m | 1.02 years | 1.02 years | 0.00 % |
| Return period for events with $SWH_o$ = 4 m | 1.38 years | 1.34 years | 2.89 % |
| Return period for events with $SWH_o$ = 5 m | 3.09 years | 2.59 years | 16.18 % |
| Return period for E1 extreme event ($SWH_o$ = 5.25 m) | 4.00 years | 3.19 years | 20.25 % |
| Return period for E2 extreme event ($SWH_o$ = 5.21 m) | 3.83 years | 3.08 years | 19.58 % |
| Return period for E3 extreme event ($SWH_o$ = 5.05 m) | 3.25 years | 2.69 years | 17.23 % |
| Return period for E4 extreme event ($SWH_o$ = 5.36 m) | 4.51 years | 3.51 years | 22.17 % |
| Return period for E5 extreme event ($SWH_o$ = 5.21 m) | 3.83 years | 3.08 years | 19.58% |
| Return period for E6 extreme event ($SWH_o$ = 5.09 m) | 3.38 years | 2.78 years | 17.75 % |
| Return period for E7 extreme event ($SWH_o$ = 7.32 m) | 53.06 years | 24.91 years | 53.23 % |

Table 4. Return period computed for two different periods, as derived from hourly in situ observations from Melilla coastal buoy. The long-term extreme sea state was characterised by using the Peak Over Threshold method with the fitting of a three-Parameter Weibull probability distribution to the observed significant wave height ($SWH_o$).

**Figures**

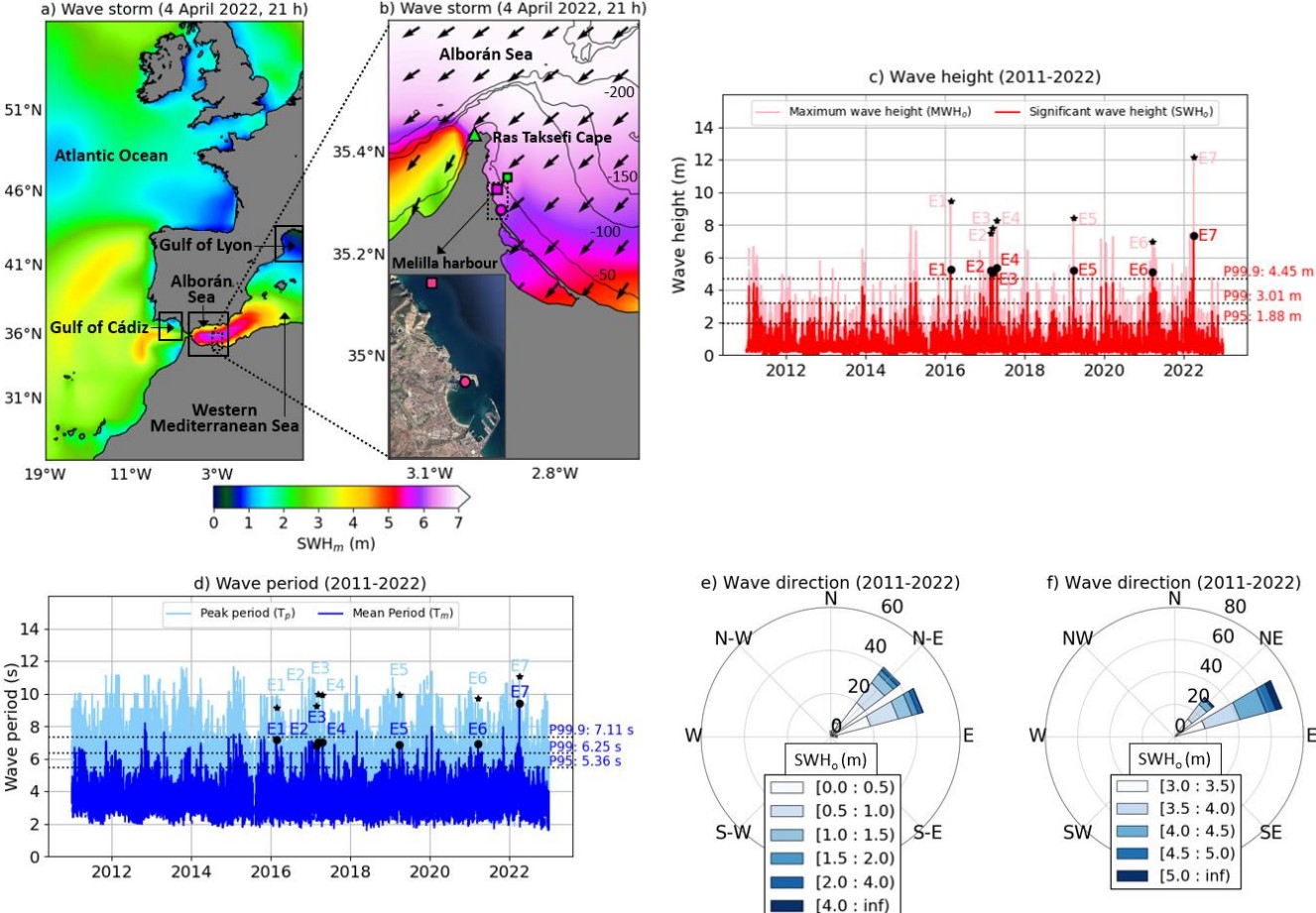


**Figure 1. Wave storm in the SW Mediterranean Sea; a) hourly map (4 of April 2022, 21 h local time) of SWH$_m$ at synoptic scale during the peak storm as derived from ERA5 reanalysis -product ref. no. 3 (Table 1); b) hourly map of SWH$_m$ and MWD$_m$ at coastal scale during the peak storm as derived from MED reanalysis -product ref. no. 4 (Table 1). Isobath depths are labelled every 50 m. Magenta dot and square represent Melilla tide-gauge and coastal buoy location, respectively. Green triangle and square indicate the location of Ras Taksefi Cape and the grid point of MED reanalysis closest to Melilla coastal buoy, respectively; c) Hourly time series of SWH$_o$ and MWH$_o$ recorded at Melilla coastal buoy for 2011-2022 -product ref. no. 1 (Table 1)-. Black dots and stars indicate the seven extreme events examined, labelled from E1 to E7. The 99.9th (P99.9), 99th (P99) and 95th (P95) percentiles are represented by horizontal black dotted lines; d) Hourly time series of T$_m$ and T$_p$ for 2011-2022 -product ref. no. 1 (Table 1)-; e) Wave**

 rose illustrating the main incoming directions (MWD$_o$) during 2011-2022 -product ref. no. 1 (Table 1); f) Wave rose showing the MWD$_o$ associated with SWH$_o$ values above P99 (3.01 m) during 2011-2022.

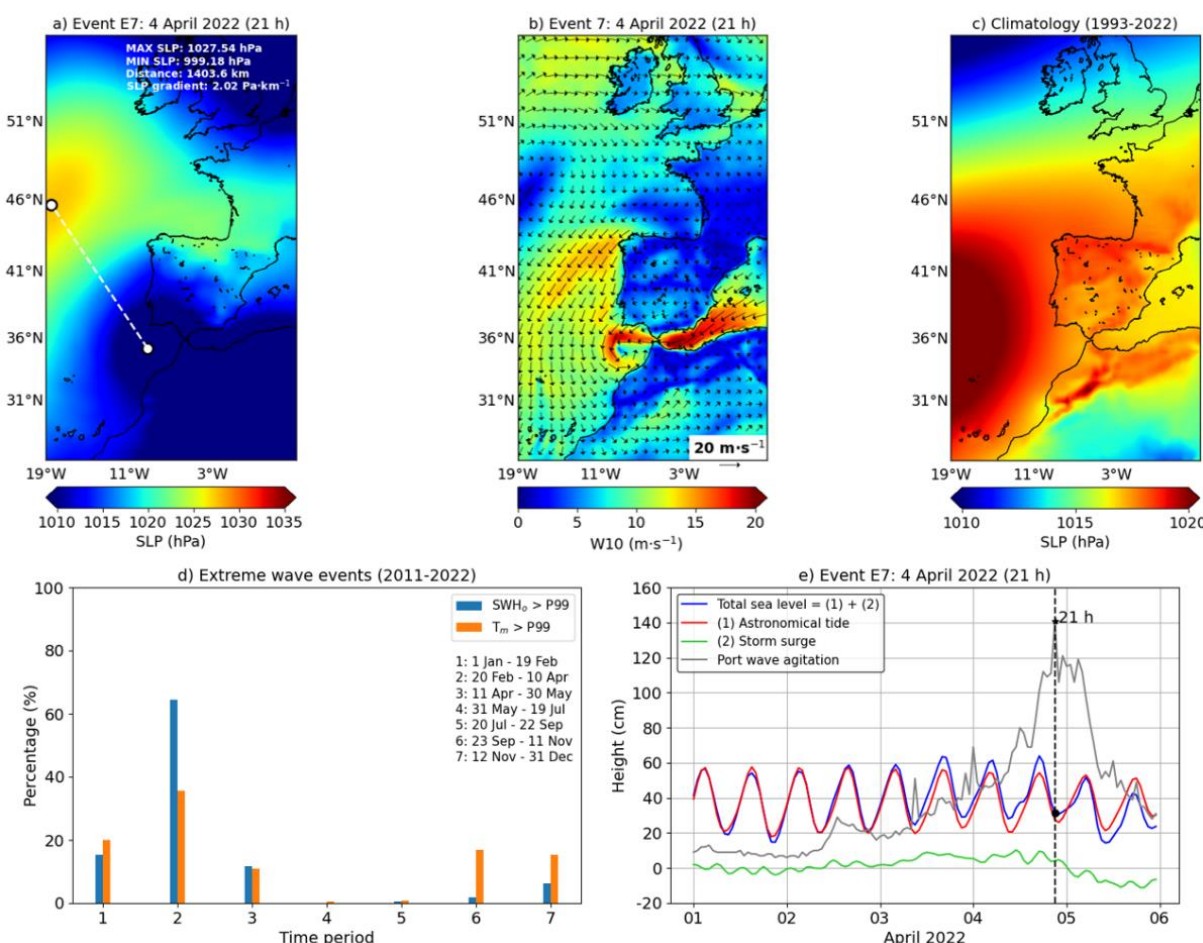

Figure 2. (a-b) Hourly synoptic patterns of sea level pressure (SLP) and wind at 10 m height (W10) during the extreme event E7; c) Climatology (1993-2022) of SLP. Maps derived from ERA5 reanalysis -product ref. no. 3 (Table 1); d) Bar diagram with the temporal distribution of events above the 99$^{th}$ percentile (P99) of significant wave height (SWH$_o$) and mean wave period T$_m$ derived from the 12-year time series (2011-2022) provided by Melilla coastal buoy (product ref. no. 1 in Table 1). The annual time span was divided into seven 50-day periods, except period 5 (20 July-22 September) which is composed by 65 days; e) Time series of total sea level height (blue line) and port agitation (black line) observations during E7 extreme event as provided by Melilla tide-gauge (product ref. no. 2 in Table 1). Astronomical tides and storm surge component (meteorological residuals) are represented by the red and green lines, respectively. The vertical dashed black line indicated the peak of E7 wave storm.

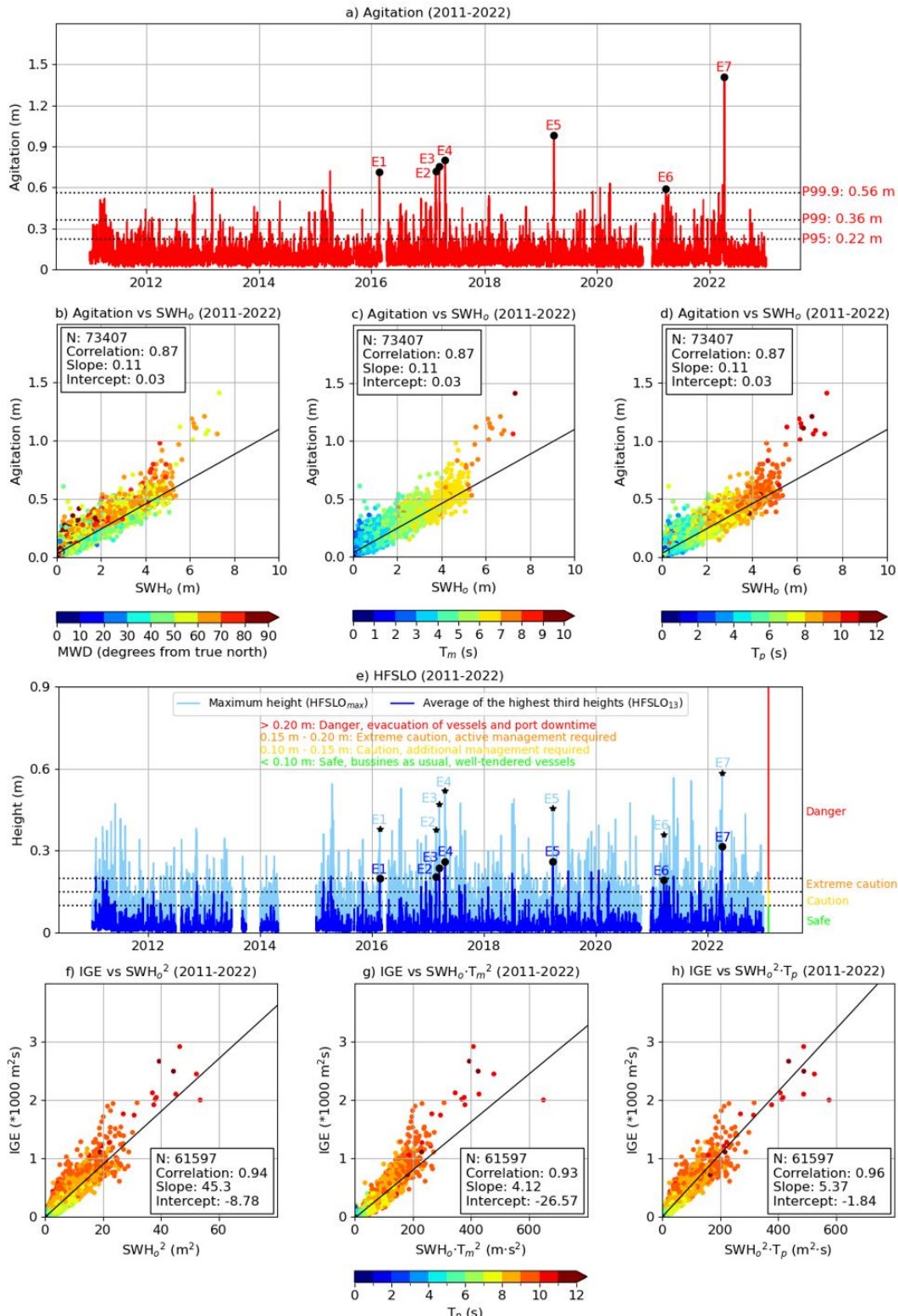

**Figure 3. a) Hourly time series of agitation inside the harbour for the period 2011-2022 (product ref. no. 2 in Table 1) as provided by Melilla tide-gauge; b-d) Best linear fit (solid black line) of scatter plots of the harbour agitation against $SWH_o$ observations provided by Melilla coastal buoy. Statistical metrics are adhered in the white box, where N represents the number of hourly observations; e) Hourly time series of High Frequency Sea Level Oscillations (HFSLO) with periods between 30 s and 1 h: maximum height (cyan line) and average of the highest third heights**

**(blue line) for the period 2011-2022 (product ref. no. 2 in Table 1), as registered by Melilla tide gauge. The seven extreme events analysed in this work are denoted by black stars and dots. Thresholds for port management, which are universally common to all locations (McComb et al., 2020; McComb, 2011) are indicated with horizontal dotted lines; f-h) Best linear fit (solid black line) of scatter plots of the energy in the IG band (IGE) against offshore wave observations from Melilla coastal buoy.**


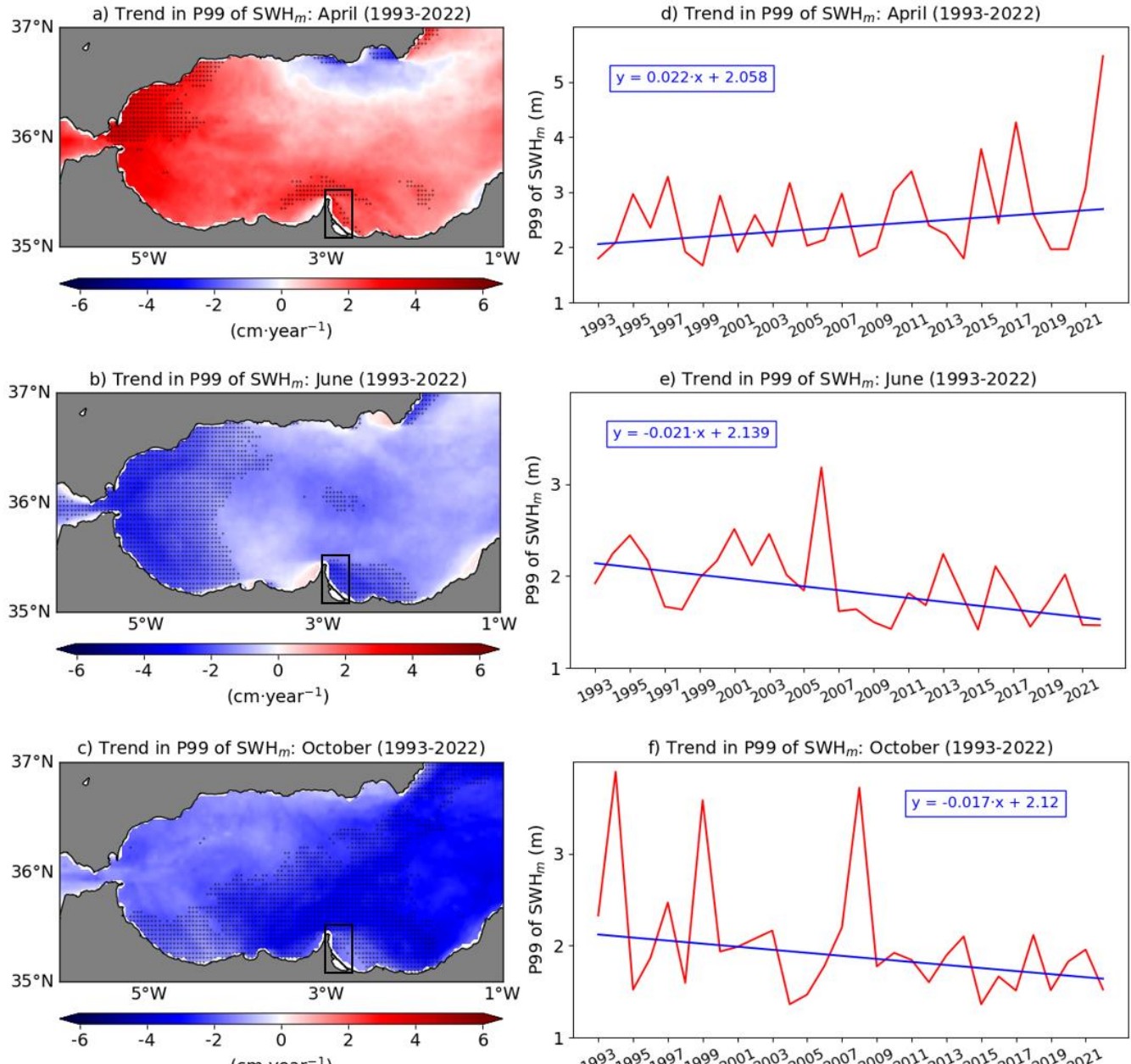

**Figure 4. Left column: monthly trend maps of the 99th percentile (P99) of significant wave height (SWHm) over the Alborán Sea for the 1993-2022 period as derived from MED reanalysis -product ref. no. 4 (Table 1)-. Areas with statistically significant trends at the 90% confidence intervals are denoted by black dots. Right column: temporal trends, computed over the Melilla subdomain (represented by a black box in the associated maps).**

**Annex**

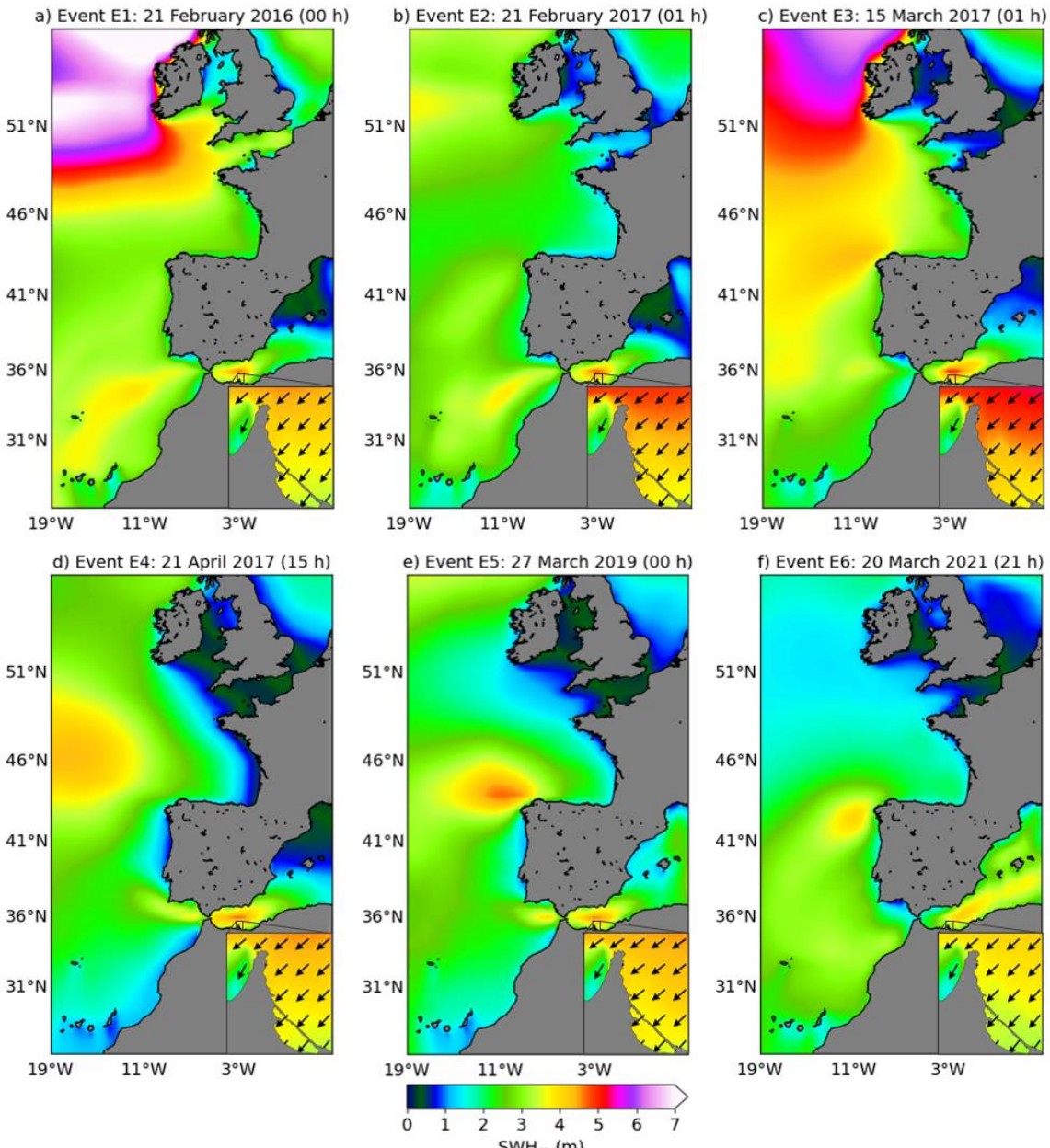

**Annex 1. Hourly maps of significant wave height (SWH$_m$), derived from ERA5 reanalysis -product ref. no. 3 (Table 1)-, corresponding to six extreme wave events (E1-E6) affecting Melilla area. Small maps in the right bottom corner of each panel represent the hourly SWH$_m$ and wave propagation direction in the vicinity of Melilla harbour as derived from MED reanalysis -product ref. no. 4 (Table 1)-. The hour represents local time.**


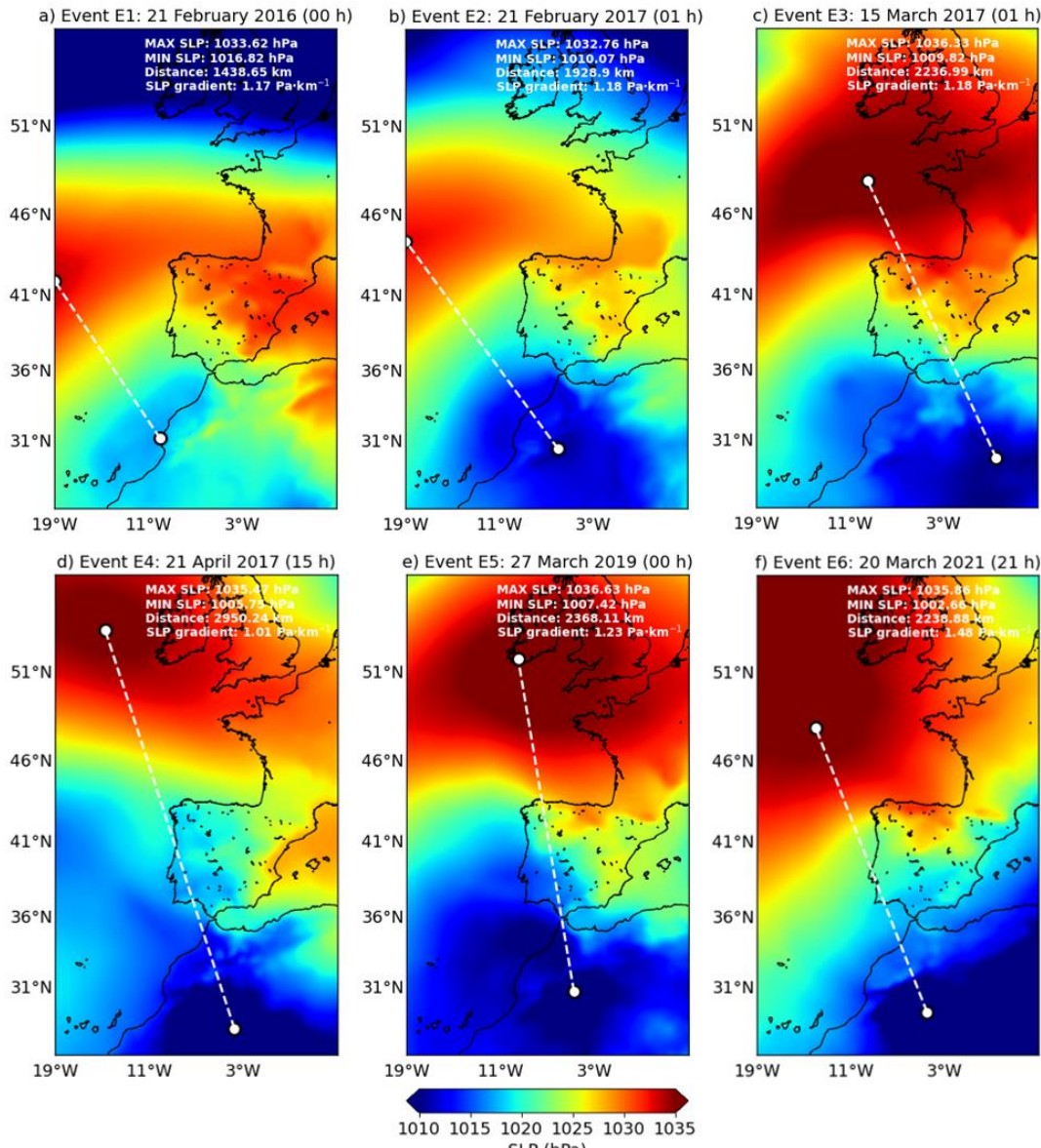

**Annex 2. Hourly maps of sea level pressure (SLP), derived from ERA5 reanalysis -product ref. no. 3 (Table 1)-, corresponding to six extreme wave events (E1-E6) affecting Melilla area. Maximum and minimum values of SLP are marked with white dots and linked with a dashed white line. The distance between both pressure centres and the 855 related SLP gradient are indicated in the upper right corner. The hour represents local time.**

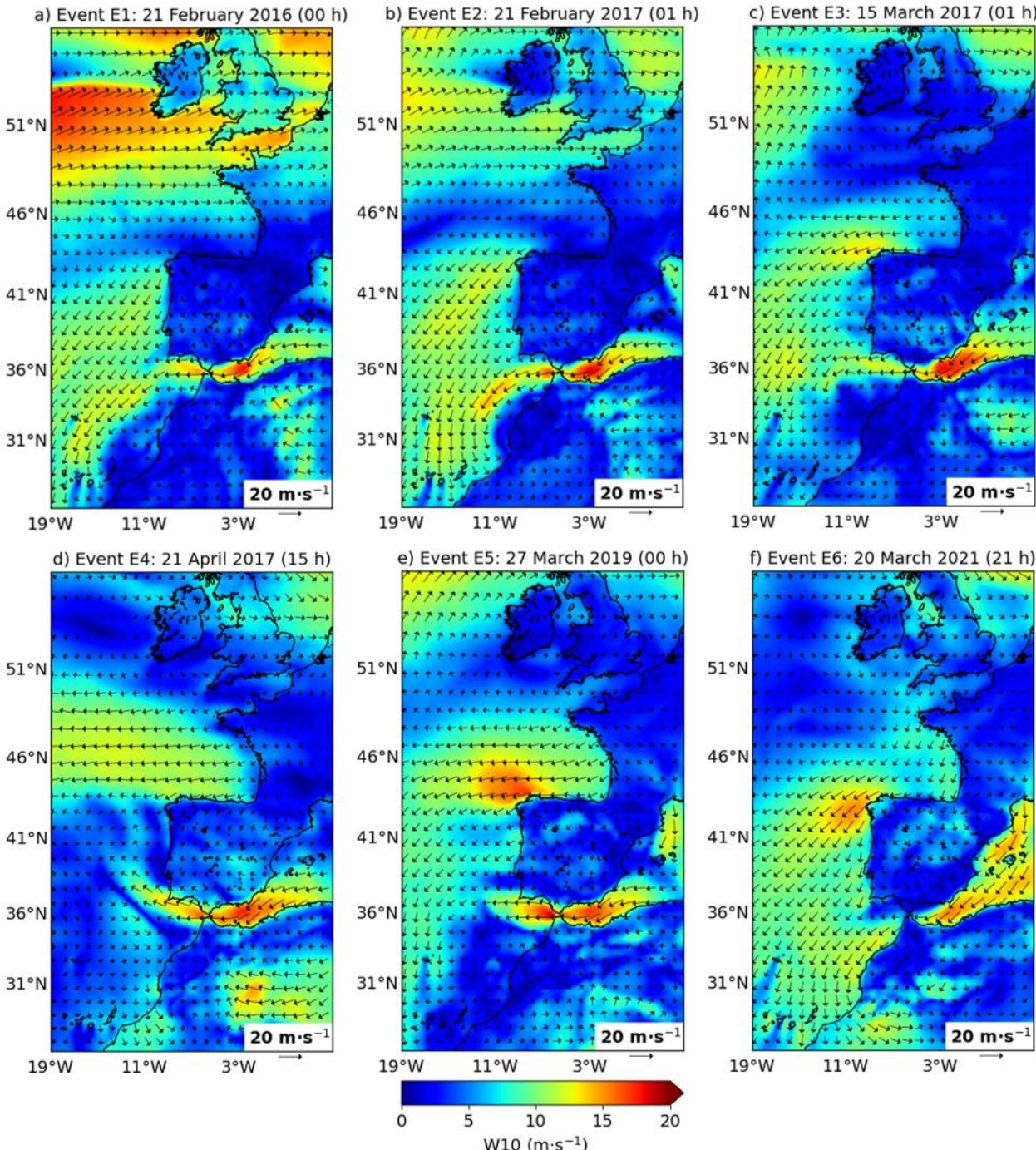

**Annex 3. Hourly maps of wind at 10 m height (W10), derived from ERA5 reanalysis -product ref. no. 3 (Table 1)-, corresponding to six extreme wave events (E1-E6) affecting Melilla area. The hour represents local time.**

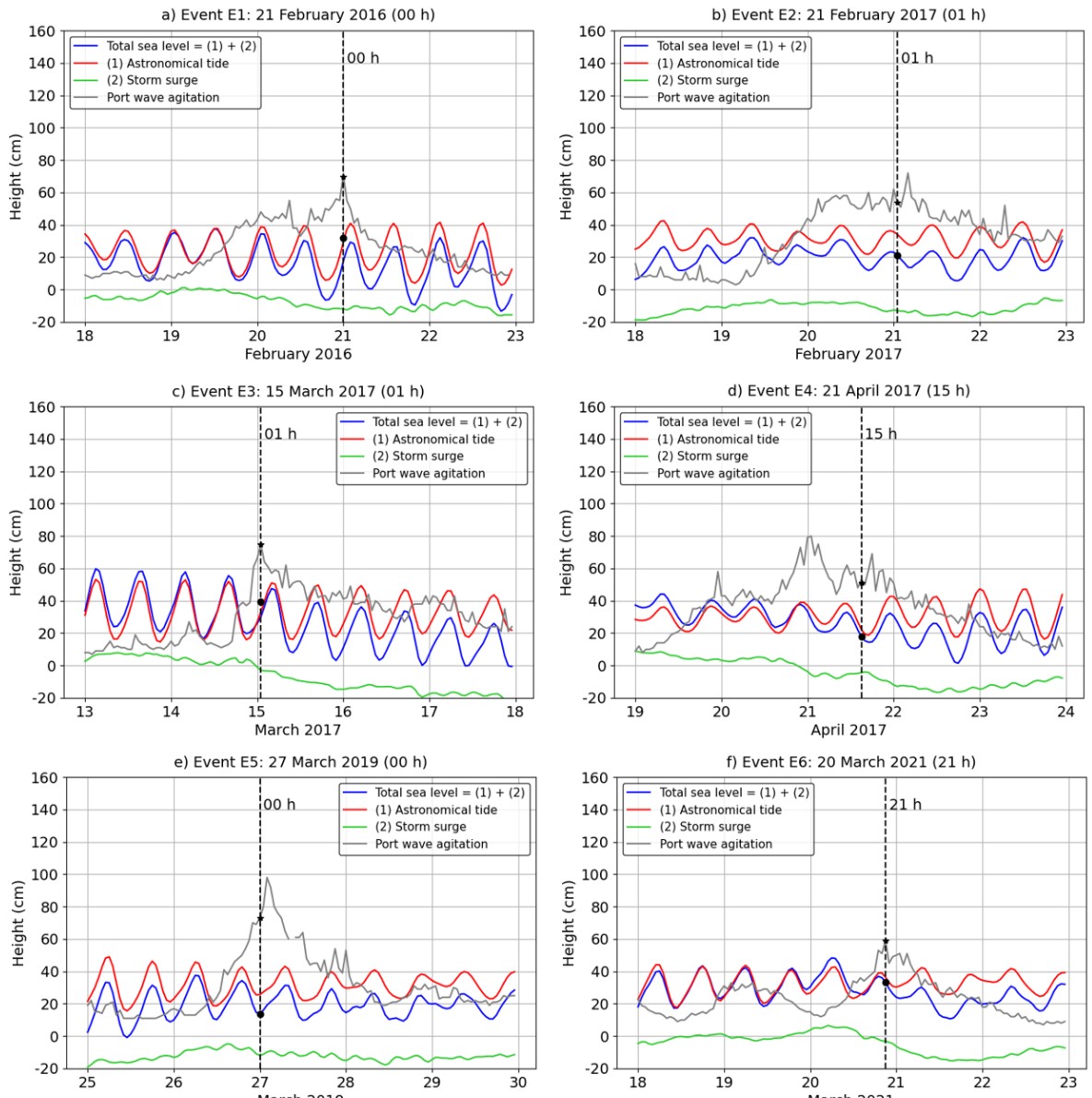

**Annex 4.** Hourly times series of sea level height (blue line) and port agitation (grey line) observations corresponding to the six extreme wave events detected before the study case and labelled in Figure 1d. Observations provided by Melilla tide-gauge (product ref. no. 2 in Table 1). Astronomical tides and meteorological residuals are represented by the red and green lines, respectively. The vertical dashed black line indicated the peak of the wave storm for each of the six events analysed.

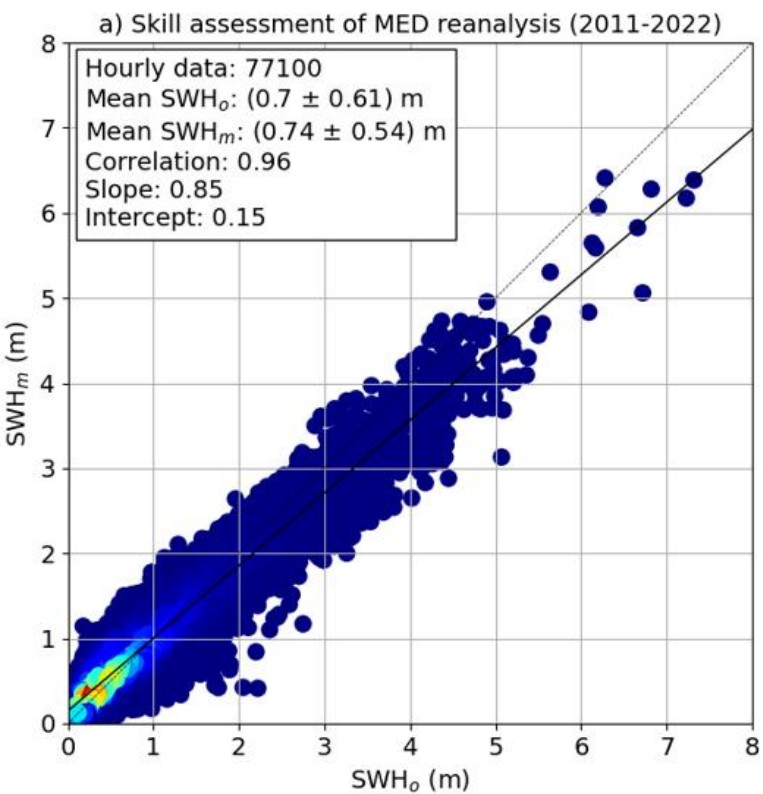

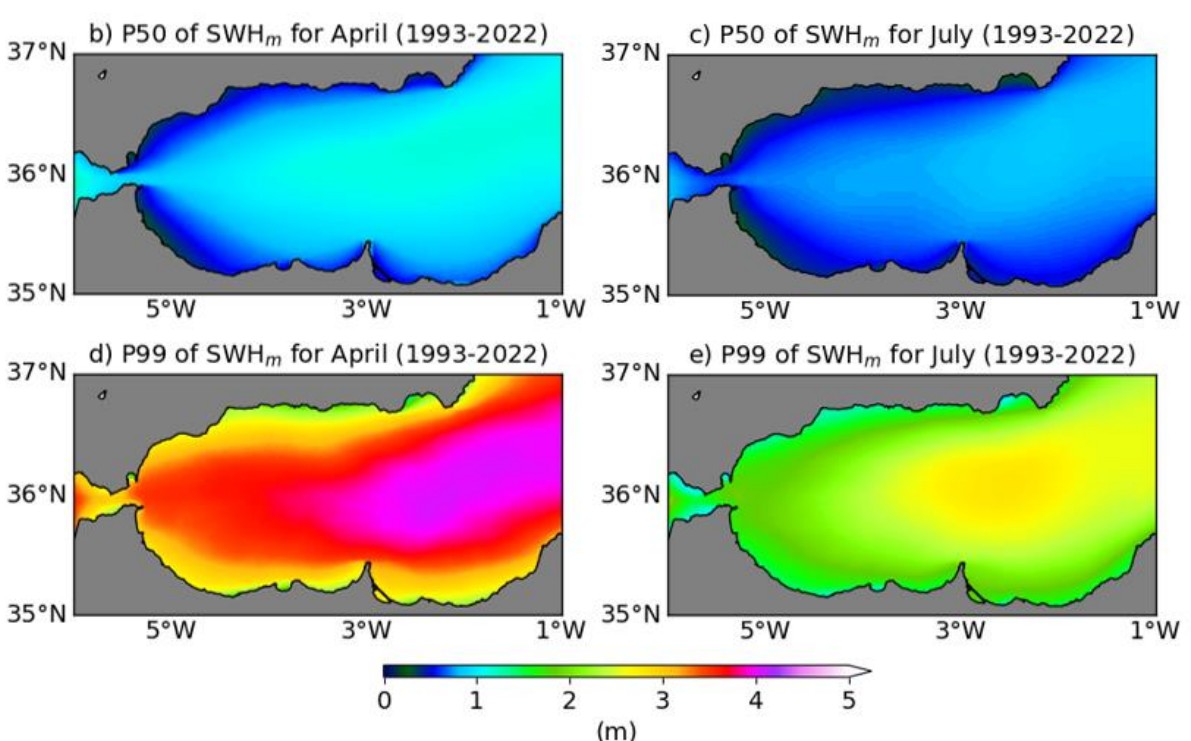

**Annex 5. a) Skill assessment of MED reanalysis -product ref. no. 4 (Table 1)- at the grid point closest to Melilla coastal buoy -product ref. no. 1 (Table 1): best linear fit (solid black line) of scatter plot between hourly estimations of modelled ($SWH_m$) and observed ($SWH_o$) significant wave heigh for the concurrent 12-year period (2011-2022). The dotted black line represents the result of perfect agreement with slope 1.0 and intercept 0. Statistical metrics are adhered in the white box; b) Spatial distribution of the 50$^{th}$ -P50- (b, c) and 99$^{th}$ -P99- (d, e) percentiles of $SWH_m$ over the Alborán Sea for April (b, d) and July (c , e), as derived from MED reanalysis for the 1993-2022 period.**

875

880