# Peer review of "Monitoring the record-breaking wave event in Melilla harbour (SW Mediterranean Sea)"

_State of the Planet, 2023_

## Referee Comment (RC1)

**Review of the article titled "Monitoring the record-breaking wave event in Melilla harbour (SW Mediterranean Sea)" by Lorente, P., et al. 2023**

The manuscript "Monitoring the record-breaking wave event in Melilla harbour (SW Mediterranean Sea)" by Lorente, P., et al. 2023 uses different database such as reanalysis, forecasting model, radar tide-gauge and *in situ* coastal buoys, to describe an oceanic extreme event that occurred in the Melilla port during April 4th and 5th, 2022. It also analyzes the extreme regime in the Alboran Sea. The impacts of extreme wave events on harbors and the need to revise the level of security within them regarding the new climatic conditions are interesting points to study. However, the reviewer considers that the article needs crucial improvements throughout the manuscript before being considered for publication in the journal State of Planet.

**OVERALL COMMENTS**

- The abstract should be rewritten to provide a more comprehensive explanation of all the values presented by the authors.

- One of the main shortcomings of the manuscript is the explanation of the different datasets used. To consider the article for publication, a comprehensive restructuring of the data section is necessary to address the following issues:

    1. What is the source of the data?
    2. What is the period during which they were used?
    3. What are the temporal and spatial resolutions?
    4. When and why were these data used?

    All this information can be included in Table 1. I suggest including the following columns in Table 1: Variables (SWH, wave period, wave direction, etc.), temporal resolution, spatial resolution, and time span.

- The time span for the different datasets should be standardized. Sometimes the time period is from 1993 to 2022, while other times it is from 2010 to 2022, or from 2008 to 2022, or even from 2015 or 2011 to 2022. This inconsistency extends throughout the article, including the methods section and various figures. If standardization is not possible due to the different scales analyzed, it must be specified why and reference the database being used.

- Why is the "wave forecsast model" of Puertos del Estado used? Would not it be more consistent to use the same database for atmospheric and oceanic variables (such as ERA5)?

- Another deficiency of the manuscript is the lack of consistency in calculating the $99^{\text{th}}$ percentile. The authors use both the annual and monthly $99^{\text{th}}$ percentile, as well as climatology (the average of each of the months, e.g., January, February, etc.), interchangeably, even though these values are statistically different.

- The methods section should be rewritten and restructured, as the method described as "the percentile method" is essentially the peak over threshold (POT) method. Why was the $99^{\text{th}}$ percentile threshold chosen as a reference instead of other values?

- The use of tables is excessive in the manuscript, making it challenging for the reader to follow the narrative. Tables 2 and 3 should be integrated into the introduction section to improve readability. Additionally, Table 4 should be removed, as the results presented there are better visualized in Figures 1 and 3.

- The figures should be renumbered according to their order in the manuscript.

- A climatic analysis is recommended, including an examination of correlations with different climatic indices influencing the area and an analysis of temporal variability using for example, wavelet-type tools.

- The manuscript neglects the value of tides, even though the tidal range in the Mediterranean can reach up to 1 meter. However, it has been proven that the $99^{\text{th}}$ percentile of the IG is 0.28 m, and of the agitation range is 0.38 m, which is within the order of magnitude of tides in the Mediterranean. Therefore, a sensitivity study of the tidal value in the port should be conducted before neglecting this factor.

- The third major deficiency in the work is the study of extreme event trends in Melilla port. In Figure 4, it can be seen that for the area marked with a black rectangle, most of the pixels do not show a significant trend for April or July (the two months selected for a comparison between P99 and P50 in Annex 4). In my opinion, it cannot be concluded that the regression line is significant based on the time series shown in Figure 4 of the manuscript; the series exhibit too much variability.

- In this work, the analysis of wave height is detailed, while the analysis of wave period is given less attention, even though, for agitation activity, the period is more relevant than the wave height (Eq. 4). This is why in event E7, the agitation is so high compared to the time series, as the period at that time is significantly higher than in the rest of the time series. This fact should be given more emphasis, and the atmospheric conditions that could have caused this remarkable event should be explored.

- The conclusion section could focus more on how ports need to revise their security protocols based on studies of extremes in the surrounding area, taking into account the analysis of return periods.

**SPECIFIC COMMENTS**

**Introduction**

**L41.** Modify the order of the tables according to when they appear in the text.

**L44.** Provide the link to the ECCLIPSE website.

**L55.** Infragravity waves have a period ranging from 25 seconds to 5 minutes, as indicated by [Munk, 1950].

**L59.** Table 4 could be omitted as it is redundant with figures 1 and 3.

**L60.** In the study area, significant wave heights (SWH) exceed 7m, the same order of magnitude than in the Gulf of Lion.

**Data**

**L110.** When does the multi-year wave product reanalysis end and the interim dataset begin?

**Methodology**

**L129.** Why if there are buoy data from 2008, do the authors choose to use them only from 2010?

**L137.** Which spiking method did you use? Were the gaps small enough to ensure that the time series was not totally distorted after processing?

**L140.** Pearson correlation coefficient.

**Eq 2 and 3.** Why do you use the sample variance instead of the population variance?

**L155.** The correct reference was Stockdon et al. (2006), not Inch et al. (2017).

**L160.** Specify the data that were used.

**Results**

**L173.** Specify the time span.

**L180.** Why do you consider data for wave directions only for the period between 2011 and 2022?

**L186.** How do you calculate the exceedance threshold and the time between two independent storms?

**L233.** Could you provide spectra to demonstrate how the infragravity waves dominate the energy during the analyzed events?

**L235.** It is not possible to see all these results in Table 6. Could you display them graphically?

**L243.** Would you mean "20 minute time-series"?

**L253.** Instead of "the 655 hourly", it would be clearer to mention the time span.

**L268.** How do you calculated the "monthly P99"? Is it the P99 of all the January data (February, March, etc.)? Or is it the mean value of all the P99 from all the January, February, etc. months?

**Conclusions**

**L313-321.** These points should be included within the introduction section.

**L336.** It is not the "percentile's method", it is the peak over threshold.

**Bibliography**

**L421.** Berta, et al. (2020) should appear after Bensoussan, et al. (2019).

**Annexes**

**Annex 3.** Adjust all the colorbars, as P99 seems smaller than P50.

**Annex 5.** Consider removing this annex because the most of the pixels show non-significant trend values.

**References**

Walter H Munk. On the wind-driven ocean circulation. *Journal of meteorology*, 7(2):80–93, 1950.

---

## Author Comment (AC1)

Funded by
the European Union

**Copernicus**

**Ocean State Report 8**

**Call of contribution**

**COPERNICUS MARINE SERVICE**

**Mercator Ocean International**
**12 December 2022**

MERCATOR OCEAN International, 2 avenue de l'aérodrome de Montaudran - 31400 Toulouse, France
Société civile de droit français au capital de 2 000 000 € - 522 911 577 RCS Toulouse

marine.copernicus.eu
mercator-ocean.eu

**OVERVIEW**

**Call of contribution for the 8th cycle of the Copernicus Ocean State Report (Copernicus OSR): Q&A**

**Where can I find information on the call of contribution?**

➢ All documents for the OSR8 call of contribution are stored at (incl. templates) the OSR8 atlas link: https://atlas.mercator-ocean.fr/s/ycWec5N62ZjjygS

➢ Information calls are proposed on:
  o Thursday, 22.12.2022, 10 am, (Join Zoom meeting: https://us02web.zoom.us/j/87129737226 ; Meeting ID: 871 2973 7226, Passcode: 563558)
  o Thursday, 12.01.2023, 11am (Join Zoom Meeting: https://us02web.zoom.us/j/89626171412; Meeting ID: 896 2617 1412; Passcode: 468252)
  o Friday, 20.01.2023, 10am (Join Zoom Meeting: https://us02web.zoom.us/j/82506994903; Meeting ID: 825 0699 4903; Passcode: 889805)

If any of these proposed time slots is not suitable for you, please contact the OSR coordination team via osr8@mercator-ocean.fr

**How do I submit my proposal for Copernicus OSR8?**

➢ The propositions can be submitted through upload in the respective chapter folders: https://atlas.mercator-ocean.fr/s/ycWec5N62ZjjygS

➢ More details on each chapter are provided below (section 2), and further questions can be tackled during the Q&A zoom call (see details above).

**How do I propose my contributions?**

➢ A template is provided at the end of this document and on the link above, the OSR8 atlas link (see question 1). Once the template is completed, please upload it on the already mentioned atlas link. The use of at least one CMEMS product is mandatory, and up to the target year, 2022.

**When does the call of contribution for the OSR8 cycle close?**

➢ The call of contribution will close on Wednesday, 1st of February 2023.

**Which is the major point of contact?**

➢ Correspondence to: osr8@mercator-ocean.fr

**How is the 8th CMEMS OSR cycle organized?**

> ➢ The roadmap for the 8th cycle of the CMEMS OSR is provided in Figure 3, and further described in section 1.2 below.

**1    ORGANISATION**

**1. Organization of the 8th cycle of the Copernicus Ocean State Report**

The overall organization of the Copernicus Ocean State Report (Fig. 1) is implemented across different levels, i.e.:

- the Ocean State Report (OSR) coordination is assured by Mercator Ocean international and covers the overall coordination, strategy, and provides support and assistance to the author team for the draft development throughout each cycle.
- the OSR editorial board: The OSR editorial board manages the independent (from the coordination) peer-review process in collaboration with the State of the Planet Journal.
- the OSR author team is established during each Report cycle through the call of contribution process and is fully supported by the OSR coordination. Copernicus Marine producers are solicited for the contributions for each call and is also open for contributions from external experts. The author teams are additionally supported by OSR guidelines, which will be provided after the call of contribution phase.
- The OSR content follows the goals of the OSR strategy, and is organized in 4 principal chapters, which are further detailed in section 2.2.

[Figure]

*Figure 1: Overview on the organization of the Copernicus Ocean State Report (OSR).*

A roadmap for the development of the Copernicus OSR draft is presented in Figure 2. Each cycle is coordinated within several steps that include the following milestones:

[Figure]

***Figure 2:*** *Roadmap of the Copernicus OSR draft developments. Each cycle is coordinated within different steps for chapter 1 – chapter 4 (section 2.2), including several milestones such as the call of contribution, the launch of each OSR cycle development, the OSR draft development and the first submission to the Journal for peer-review.*

- **MS 1- Establishment of call of contribution guideline**: The OSR coordination develops each year in October-November a guideline for the call of contribution, which includes information on the criteria for contributions for each OSR chapter, as well as details on the organisation for this call.

- **MS 2- OSR call of contribution:** The call of contribution for each OSR Year runs from December to January (year+1) and is launched through notification via mail to the TAC and MFC leaders and the MYP experts. A mid-term remote call is usually organized by the OSR coordination to allow for preliminary exchanges with applicants, and clarifications. The call of contribution usually closes during the last week of January/early February. For Chapter 1 and according to the refined OSR content (see section 2.2), the call of contribution will include a structuration of chapter 1 on the indicator topics, allowing for the nomination of authors by the TACs and MFCs for contribution.

- **MS 3 - Launch of OSR cycle:** During January/early February, the OSR coordination evaluates the proposed contributions. In some cases, further exchanges with some individual author teams will be proposed to seek for clarifications and to decide whether a section, from chapter 2 and 3, will run through a 1-year or a 2-year development phase. The outcomes of the call of contribution are then communicated usually mid-February during a remote call with the author teams. During the second part of February, the OSR coordination establishes the organisation (e.g., google doc launch), and prepares and disseminates the OSR guideline, including details on section developments, and organisational elements for author assistance. A remote call is then organized by the OSR coordination during which each author team introduces their proposed sections. For chapter 1, a specific call will be organized with the nominated experts, and additional experts will be potentially solicited to complement expertise.

- MS 4 - OSR draft development: The period from February to July is dedicated to the section development. This period is organized within several steps, such as deadlines for several stages of the section development, regular remote calls with the entire author team, ad-hoc individual calls for specific clarifications and draft development support, internal review steps, etc.

- MS 5- OSR first submission to the journal for peer-review: During June to July, intensive exchanges between the OSR coordination and the author teams are established through final internal review steps, and intensive draft development support. The request for the submission of the final section packages (section draft, author information, figures in low resolution) is usually launched by mid of June. Experience has shown that the period of one month needs to be usually anticipated to finalize the complete deposit of all sections to get ready for the submission of chapter 2 - chapter 4 by mid of July.

**1.1 Document and information exchange:**

- O**ne main OSR8 google** doc will be used for the organization:
  https://docs.google.com/document/d/1HSFpoM7wHfyWHLm664XjmVjpmsrZxiENea
  og3ED-uUY/edit?usp=sharing Moreover, document exchange will run through the
  **OSR8 atlas link**:
  https://atlas.mercator-ocean.fr/s/ycWec5N62ZjjygS
- There are several folders provided under this link, particularly one folder called 'call_of_contribution'. In this folder you will find 4 additional folders, i.e., one for each chapter, and you are invited to **upload your proposal in the respective chapter folder** for the call of contribution.
- To facilitate the attribution of your topic to one of the 4 chapters, a **description of each chapter** is provided in section 2 of this document.
- You are also invited to access **earlier OSR reports**, more information and links can be found at: https://marine.copernicus.eu/science-learning/ocean-state-report/
- Information on the call of contribution has been sent to the CMEMS TAC & MFC leaders, the Science, OSR, OMI experts, and all Copernicus OSR7 authors.
- Once the call of contribution is closed, notifications will run through the google doc, and a specific mailing list will be established based on the contact points provided in the call of contribution documents. **Please assure to clearly indicate all emails in the call of contribution documents for those who wish to be included in the regular information exchange.**
- **A OSR8 guideline will be provided on the 09. February 2023**, i.e., when the outcome of the call of contribution is finalized and communicated via the google doc, and the second OSR8 call on the same day, 10-12am (Toulouse time).
- Propositions for the "call of contributions" will be reviewed by MOI under the criteria given below for each chapter. The outcome of the call of contribution will be communicated on the 09th of February 2023.

- Organization of call of contribution: submission of section propositions should be done by chapter. Only section propositions following the corresponding templates will be accepted by MOI, and they can be found in the appendix of this guideline and have to be uploaded on the atlas. Please copy/paste and fill in.
- The completed template should be uploaded on the atlas no later than by the 1st of Feburary 2023. The link to upload the document is given at the first page of this document. **Submission via email will be not taken into account.**

**1.2 Roadmap of OSR8: Regular remote calls, milestones and internal review process**

[Figure]

*Figure 3:* *Roadmap for the Copernicus OSR#8 development, status 12. December 2022.*

**ZOOM meetings** (all Toulouse local time): Meeting details are provided below, and will be also sent out in the morning of the meeting date to the OSR8 email list, and provided in the google-doc

- Meeting 1: Thursday, 22.12.2022, 10-12 am: Objective: Info on OSR8 cycle and new strategy, Q&A for call of contribution; (Join Zoom meeting: https://us02web.zoom.us/j/87129737226; Meeting ID: 871 2973 7226, Passcode: 563558)

- Meeting 2: Thursday, 12.01.2023, 11-12:30 am: Objective: Info on OSR8 cycle and new strategy, Q&A for call of contribution; (Join Zoom Meeting:

https://us02web.zoom.us/j/89626171412; Meeting ID: 896 2617 1412; Passcode: 468252)

- Meeting 3: 20. January 2023, 10-12am: Objective: Info on OSR8 cycle and new strategy, Q&A for call of contribution; (Join Zoom Meeting: https://us02web.zoom.us/j/82506994903; Meeting ID: 825 0699 4903, Passcode: 889805)

- Meeting 4: 9. February 2023, 10-12 am: Objective: Outcome of call of contribution (Join Zoom meeting: https://us02web.zoom.us/j/87125438987; Meeting ID: 871 2543 8987, Passcode: 907853)

- Meeting 5: 9. March 2023, 10-12 am: Objective: Overview on section organization (2h, each section will present): (Join Zoom meeting: https://us02web.zoom.us/j/87568200624; Meeting ID: 875 6820 0624, Passcode: 502258)

- Meeting 6: 3. April 2023, 10-12 am: Objective: Overview on section draft progress (2h, each section will present); (Join Zoom meeting: https://us02web.zoom.us/j/89307093323; Meeting ID: 893 0709 3323, Passcode: 110756)

- Meeting 7: 16. May 2023, 10-12 am: Objective: Overview on section draft progress (2h, each section will present); (Join Zoom meeting: https://us02web.zoom.us/j/88343850040, Meeting ID: 883 4385 0040, Passcode: 515666)

- Meeting 8: 06. June 2023, 10-12 am: Objective: Retour on internal review: major issues (Join Zoom meeting: https://us02web.zoom.us/j/88939535431, Meeting ID: 889 3953 5431, Passcode: 060452)

- Meeting 9: 22. June 2023, 10-12am: Objective: Overview on section draft progress (2h, each section will present) (Join Zoom meeting: https://us02web.zoom.us/j/81380072655, Meeting ID: 813 8007 2655, Passcode: 632969)

**Milestones:**
- Call of contributions: 12.12.-01.02.2023
- OSR8 draft development: 09.02.-22.06.2023
- Final preparation phase: 22.06.-14.07.2023
- Submission to Journal: 14. July 2023

**Organization of internal review:**

- Phase 1 internal review: 16. May – 06. June 2023: Feedback to all sections, and an overview on internal review will be provided on the OSR8 call 8.

- Phase 2 internal review: 06. June- 14. July 2023: Interactive exchanges to further support section development, including individually scheduled calls.

**2. Topical organization of the 8ᵗʰ CMEMS Ocean State Report:**

**2.1 Specifications for the sections to be developed**

Note that more details will be provided in the OSR8 guideline, and in the OSR strategy document (both distributed on the 9ᵗʰ of February 2023). Major specifications include:

- Organization of email list for notifications: the initial list of emails should be established via the call of contribution documents. All section leads have the responsibility to assure that all co-authors are listed on this notification email list if they want to. A table example for authors is provided in the templates (see appendix). Note that first / corresponding authors will keep the responsibility to regularly inform and update the entire author team.

- For OSR8 chapters 2-4, the length of the section is limited to: max. of 4 figures / section, max. of 3000 words / section. Excluded from the 3000 words: figure captions, references, table of products, statement of main outcome. Length specifications for chapter 1 are provided in section 2.2 below.

- A product table will be added into each section, and contains a list of all products used in each section. The use of at least one CMEMS product is mandatory, however, all products available (including non-CMEMS products) can be listed in this table. This holds also for chapter 1, and a product table for each indicator will be included of sections 1.1. State of the global ocean and 1.2. State of the ocean in Europe (see section 2).

- You are invited to download previous OSR issues to further understand the format of the sections, and all links are provided here: https://marine.copernicus.eu/science-learning/ocean-state-report/

- Core-period to be covered 1993-2022 (depending on product availability and product limitations (in this case, exchange with MOI), the time series can be started earlier, or later; but the inclusion of data during the **year 2022 is mandatory**. For anomaly evaluations, the climatology averaged over the longest period available should be used.

- From OSR7 onwards, all sections will be handled as single submissions as part of a special issue, and thus **each section will be linked to its own DOI for publication**. A different DOI will be established for the entire report.

**2.2 Chapter specifications**

As previous issues of the CMEMS OSR ([https://marine.copernicus.eu/science-learning/ocean-state-report/](https://marine.copernicus.eu/science-learning/ocean-state-report/)), the 8[th] issue will be organized through 4 specific chapters (Fig. 1). The major difference to previous reports is chapter 1, which is now dedicated to a synopsis of the ocean state & the marine environment over the past decades (see below).

**Chapter 1: The state of the ocean**

Objective: To provide a regular synopsis of the state of the ocean and marine environment over the past decades, and up to the target year.

This chapter will be updated annually and will contain a synopsis of results from the Copernicus Ocean Monitoring Indicators[1] for reporting, together with a scientific context and major outcomes. The starting point for OSR8 is to focus on two major regions, i.e., the global ocean and the European regional seas (Fig. 4. This chapter will be organized into two main sections (section 1.1: State of the global ocean; section 1.2: State of the Ocean in Europe), both organized across indicator topics. The indicator topics included for OSR8 are aligned to ocean-related global climate indicators from the GCOS framework[2], together with other indicators from the Copernicus Ocean Monitoring Indicator framework. Additional indicators can be proposed by experts during the call of contribution (Fig. 4). Each indicator topic should provide a state-of-the-art scientific context, together with a discussion of the results. A team of experts should be established for each indicator topic during the call of contribution, with respective responsibilities. Author teams will be listed per topic at the end of the document. Support will be provided by the Copernicus Ocean State Report coordination team.

Specifications for these two sections include:
- Science-driven description of each indicator topic should be limited to about 500 words.
- Each indicator topic should include relevant visualization, with a maximum of 2 figures per indicator topic.
- The use of at least one Copernicus Marine Service product is mandatory. A multi-product approach should be considered whenever possible to allow for collaborations, and the use of different types of products.
- Each indicator should provide information up to the year 2022.
- A robust uncertainty evaluation and associated error bar should be included for each indicator topic.

Three (for multiple options / meet availabilities) calls are proposed for the OSR experts from all TACs and MFCs to organize the contribution for chapter 1 in collaboration, and to further provide clarifications. If interested, additional experts are more than welcome to join this effort. Proposed calls for further clarifications are provided at the beginning of this document.

The contribution to this new Chapter 1 is encouraged through three different scenarios, described below.
* * *
[1] https://marine.copernicus.eu/access-data/ocean-monitoring-indicators

[2] https://gcos.wmo.int/en/global-climate-indicators#:~:text=The%20Global%20Climate%20Indicators%20are,as%20well%20as%20the%20cryosphere.

**Case 1:** An expert from the TACs and MFCs is proposed to help in the development of an indicator topic (either global or Europe, or both). The expected contribution will include the following:

- Product guidance, and if required pre-preparation of datasets, and figure development support
- Support the development of the draft, including a synthesis of state-of-the-art knowledge based on recent per-reviewed literature, and the development of a robust uncertainty framework
- Participate in internal review process of the entire section (Global, Europe, or both)

**Case 2:** An expert proposes themself to lead a specific indicator topic, either for the global section, or the Europe section, or both. In addition to the tasks proposed under Case 1, the expected contribution will include the following:

- Coordinate – with support from the OSR coordination team – the strategy and development of the section and figure development
- Assure the scientific quality of the section

**Case 3:** Proposition of an additional indicator. In addition to the tasks described under Case 1 and independently of taking the lead in the Case 2, the expected contribution will include the following:

- Assure integration of the additional indicator in the proposed structure, and assure overall coordination
- Note that the indicator should be already integrated in the Copernicus Marine Service Ocean Reporting activity (e.g., indicator published in previous OSR issues, and/or integrated in the OMI framework)

Please use the proposed template for the call of contribution process (CALL_OF_CONTRIBUTION -Templates).

**State of the global Ocean**

One section including info on:
- SST
- OHC
- Sea Level
- Ocean acidification & CO2 flux
- CHl-a
- Sea ice
- Others ➜ call of contribution, from the implemented indicator framework

**State of the Ocean in Europe**

One section including info on:
- SST
- OHC
- Sea Level
- Ocean acidification & CO2 flux
- CHl-a
- Sea ice
- Others ➜ call of contribution, from the implemented indicator framework

*Figure 4: Overview on organization, and indicator topics for chapter 1 of OSR8.*

**Chapter 2: Updated and new pathways in ocean science**

Objective: To provide regular scientific reviews of the suite of indicators that inform the OSR, and to provide Copernicus Marine internal and other scientific and technical stakeholders a high-quality peer-reviewed pathway, and incremental updates for disseminating and accessing Copernicus Marine information on the state of the ocean

This chapter will contain scientific articles updated at either annual, or bi-annual frequency. Both, Essential Ocean Variables and Ocean Monitoring Indicators will build the baseline for the scientific studies. Results for the global ocean and for European regional seas will be included, and will cover information over the past decades, and up to the target year of each Report cycle. The use of at least one Copernicus Marine product is mandatory and can be complemented by non-CMEMS products. Following either an annual or bi-annual preparation phase will be agreed between the proposal expert and the Ocean State Report lead at the kick-off meeting of each Report cycle.

**Chapter 3: Ocean state and change for relevance to society**

Objective: To provide scientific articles that contribute to improved understanding of the ocean state for relevance to society

This chapter will contain scientific articles updated at either annual, or bi-annual frequency. Both, Essential Ocean Variables and Ocean Monitoring Indicators will build the baseline for the scientific studies. Results for the global ocean and for European regional seas will be included and will cover societal-relevant information over the past decades, and up to the target

year of each Report cycle. The use of at least one Copernicus Marine product is mandatory and can be complemented by non-CMEMS products. Following either an annual or bi-annual preparation phase will be agreed between the proposal expert and the Ocean State Report lead at the kick-off meeting of each Report cycle.

**Chapter 4: Specific events in the ocean in 2022**

Objective: To provide on an annual basis scientific article on interesting events of the target year

This chapter will contain scientific articles updated at annual frequency. Both, Essential Ocean Variables and Ocean Monitoring Indicators will build the baseline for the scientific studies. Results for the global ocean and for European regional seas will be included and will cover information for the target year of each Report cycle. The use of at least one Copernicus Marine product is mandatory and can be complemented by non-CMEMS products.

**Appendix – Template**

**Template proposition for chapter 1**

Please, provide all the required information for every proposed contribution, individual cases and sections.

**Contribution to which case and section? More than one option can be selected:**

- ☐Case 1: Expert proposed to help in the assessment
    - o ☐State of the global Ocean
    - o ☐State of the Ocean in Europe
- ☐Case 2: Expert proposed to lead a specific section
    - o ☐State of the global Ocean
    - o ☐State of the Ocean in Europe
- ☐Case 3: Proposition of additional indicator
    - o ☐State of the global Ocean
    - o ☐State of the Ocean in Europe

**Which indicator topic?** Please precise.

Main contact point (name, email):

List of authors (name, affiliation):

*Example for author information, please indicate in bold those authors which wish to be included in the OSR8 email list from the 09. of February onwards:*

| Name of expert | Affiliation | Full address | Email |
|---|---|---|---|
| Karina von Schuckmann | Mercator Ocean International, France | 2 Av. de l'Aérodrome de Montaudran, 31400 Toulouse, France | Karina.von.schuckmann@mercator-ocean.fr |

**For Case 1 only:** Summary of the candidate's expertise and expectation for the section, and own contribution (max. 200 words):

**For Case 2 only:** Title of the proposed section and abstract of proposed rationale for this indicator topic (max. 500 words):

**For Case 3 only:** Short abstract (max. 500 words)**:** include in the abstract how the proposed indicator will be integrated in the planned Chapter 1 structure. Note that the indicator should be already integrated in the Copernicus Marine Service Ocean Reporting activity (e.g., indicator published in previous OSR issues, and/or integrated in the OMI framework). If this is not the case, please, submit your section to Chapter 2.

**Data use** (at least one Copernicus Marine product is mandatory):

- CMEMS product (long-name, CMEMS product name):
- Non-CMEMS product (description (max. 100 words), scientific reference and or web link to data information; link to data source/download)

Time period to be covered (evaluation should go up to 2022):

**The propositions can be submitted through upload in the respective chapter folders: https://atlas.mercator-ocean.fr/s/ycWec5N62ZjjygS**

**Appendix – Template**

**Template proposition for chapter 2-4**

Contribution for which chapter?:

Main contact point (name, email):

Title of the proposed section:

List of authors (name, affiliation):

*Example for author information, please indicate in bold those authors which wish to be included in the OSR8 email list from the 09. of February onwards:*

| Name of expert | Affiliation | Full address | Email |
|---|---|---|---|
| Karina von Schuckmann | Mercator Ocean International, France | 2 Av. de l'Aérodrome de Montaudran, 31400 Toulouse, France | Karina.von.schuckmann@mercator-ocean.fr |

Abstract (max. 500 words):

Data use:

- CMEMS product (long-name, CMEMS product name):
- Non-CMEMS product (description (max. 100 words), scientific reference and or web link to data information; link to data source/download)

Time period to be covered:

Why and for what is this topic important? (max. 150 words):

Stakeholders for the proposed topic (max. 100 words):

**For Chapter 2 & 3 only:** Will this section be developed for OSR8 or OSR9? If this section is proposed for OSR9 (i.e. 2-year development cycle), please provide a rationale & clear explanation why more time is needed (500 words):

**The propositions can be submitted through upload in the respective chapter folders: https://atlas.mercator-ocean.fr/s/ycWec5N62ZjjygS**

---

## Author Response (AR1)

**RESPONSE TO REVIEWER 1**

**Review of the article titled "Monitoring the record-breaking wave event in Melilla harbour (SW Mediterranean Sea)" by Lorente, P., et al. 2023**

The manuscript "Monitoring the record-breaking wave event in Melilla harbour (SW Mediterranean Sea)" by Lorente, P., et al. 2023 uses different database such as reanalysis, forecasting model, radar tide-gauge and in situ coastal buoys, to describe an oceanic extreme event that occurred in the Melilla port during April 4$^{th}$ and 5$^{th}$, 2022. It also analyses the extreme regime in the Alborán Sea. The impacts of extreme wave events on harbours and the need to revise the level of security within them regarding the new climatic conditions are interesting points to study. However, the reviewer considers that the article needs crucial improvements throughout the manuscript before being considered for publication in the journal State of Planet.

Many thanks to the anonymous Reviewer-1 for the detailed review and the number of useful tips provided. Please find below a thorough point-by-point response with the hope of improving the quality of the document to make it acceptable for final publication.

It is worthwhile mentioning that we have successfully accomplished 37 from the 39 suggestions provided by the Reviewer-1, which constitutes the 95%.

Just in case the Reviewer-1 is not familiarized with the 8$^{th}$ Ocean State Report initiative, we would like to clarify that it is characterized by some specific limitations in terms of length (up to 3000 words) and maximum number of figures (4). Therefore, we have tried hard to fulfil all the Reviewer-1´s requirements but always adhering to the journal´s premises.

**OVERALL COMMENTS**

[Comment 1] The abstract should be rewritten to provide a more comprehensive explanation of all the values presented by the authors.

**OK, 100% accomplished**. The abstract has been completely rewritten to better clarify the main results derived from the present study.

[Comment 2] One of the main shortcomings of the manuscript is the explanation of the different datasets used. To consider the article for publication, a comprehensive restructuring of the data section is necessary to address the following issues:

a. What is the source of the data?

b. What is the period during which they were used?

c. What are the temporal and spatial resolutions?

d. When and why were these data used? All this information can be included in Table 1.

I suggest including the following columns in Table 1: Variables (SWH, wave period, wave direction, etc.), temporal resolution, spatial resolution, and time span.

**OK, 100% accomplished**. We fully agree with this comment: data section has been completely restructured. The suggested columns have been inserted in a new Table 2 in order to provide a thorough answer to the questions above shortlisted by the Reviewer-1. We could not use Table 1 for this purpose as Table 1 has a mandatory format (compliant with the Ocean State Report guidelines) that must be fully respected for final acceptance and publication.

[Comment 3] The time span for the different datasets should be standardized. Sometimes the time period is from 1993 to 2022, while other times it is from 2010 to 2022, or from 2008 to 2022, or even from 2015 or 2011 to 2022. This inconsistency extends throughout the article, including the methods section and various figures. If standardization is not possible due to the different scales analysed, it must be specified why and reference the database being used.

**OK, 100% accomplished**. We fully understand the Reviewer-1´s confusion at first sight. We have standardized instrumental datasets as shown in the **new Table 2** with the aim of clarifying the situation:

i)  **1993-2022** is the time span for both the **MED wave reanalysis and ERA5 reanalysis**.
ii) **2011-2022** is now the time span for **all in situ observational data** (from both the tide-gauge and the coastal buoy). Data section, Methodology section and diverse figures have been updated accordingly. Please, also accept our apologies for the typo "2010-2022" along the entire manuscript, which was meant "2008-2022" in the first version of the manuscript. In the new revised version of the manuscript, the time span employed and cited is always 2011-2022.

[Comment 4] Why is the "wave forecast model" of Puertos del Estado used? Would not it be more consistent to use the same database for atmospheric and oceanic variables (such as ERA5)?

**OK, 100% accomplished**. For consistency reasons, all those maps (of sea level pressure, wind at 10 m height and significant wave height) covering the regional domain (from Canary Islands to Ireland) are now based on ERA-5 reanalysis. Accordingly, the wave forecast model of Puertos del Estado has been deleted from the table of products used (Table 1 and Table 2).

Notwithstanding, Figure 1b is nowadays based on MED reanalysis outcomes because: i) Reviewer-2 has requested to plot the wave direction in the vicinity of Melilla harbour; ii) MED has higher horizontal resolution than ERA-5 reanalysis (as shown in Table 2) so a larger amount of wave vectors can be plotted; iii) Furthermore, MED reanalysis provides finer details of the SWH field over Melilla harbour area, including the shadow effects at the lee of Ras Taksefi Cape; iv) since the MED wave reanalysis is forced with ERA-5 atmospheric fields (as stated in section 2.2.2), the consistency of this approach is ensured.

[Comment 5] Another deficiency of the manuscript is the lack of consistency in calculating the 99th percentile. The authors use both the annual and monthly 99th percentile, as well as climatology (the average of each of the months, e.g., January, February, etc.), interchangeably, even though these values are statistically different.

**OK, 100% accomplished**. In order to avoid any confusion, Annex 3b and 3c have been removed so annual P99 values are no longer shown in the document.

Nowadays, in the new version of the manuscript the monthly P99 was only calculated for MED reanalysis (modelled product) spanning the period 1993-2022 in:

i)      Figure 4: Trends of P99 of $SWH_m$
ii)     Annex 5 (b-e): maps of $50^{th}$ and $99^{th}$ percentiles of $SWH_m$ for April and July.

The P99 of $SWH_o$ (Figure 1c), $T_m$ (Figure 1d) and harbour agitation (Figure 3a) were computed for the entire 12-year (2011-2022) in situ timeseries (observational product) to characterize the seven extreme events selected.

[Comment 6] The methods section should be rewritten and restructured, as the method described as "the percentile method" is essentially the peak over threshold (POT) method. Why was the 99th percentile threshold chosen as a reference instead of other values?

**OK, 100% accomplished**. The methods section has been totally reformulated. The 99[th] percentile was selected as reference percentile for the most extreme wave events affecting Melilla area, in agreement with previous approaches reported in the literature (Zacharioudaki et al., 2022b, Barbariol et al., 2021).

References:

Zacharioudaki, A., Ravdas, M. and Korres, G.: Wave climate extremes in the Mediterranean Sea obtained from a wave reanalysis for the period 1993-2000, Ocean state Report, Issue 6, Journal of Operational Oceanography, 2022b.

Barbariol, F., Davison, S., Falcieri, F.M., Ferretti, R., Ricchi, A., Sclavo, M. and Benetazzo, A.: Wind Waves in the Mediterranean Sea: An ERA5 Reanalysis Wind-Based Climatology, Front. Mar. Sci., 8, 760614, doi: 10.3389/fmars.2021.760614, 2021.

[Comment 7] The use of tables is excessive in the manuscript, making it challenging for the reader to follow the narrative. Tables 2 and 3 should be integrated into the introduction section to improve readability. Additionally, Table 4 should be removed, as the results presented there are better visualized in Figures 1 and 3.

**OK, 100% accomplished**. We fully agree, as the manuscript must be shortened and synthesized to make it compliant with the Special Issue guidelines, this convenient suggestion is more than welcome. The number of Tables has been shortened from 6 to 4. We have integrated all the information gathered in Table 2 and Table 3 into the manuscript body. Table 4 has been rewritten and renamed as Table 2. The new Table 3 has been inserted following the Reviewer-2´s suggestion ([Comment 21]: "*This table (in Figure 1) could be moved to the "Table" section, allowing that more information could be added*"). Finally, all the references to these tables have been updated accordingly.

[Comment 8] The figures should be renumbered according to their order in the manuscript.

**OK, 100% accomplished**. The figures were revised and corrected.

[Comment 9] A climatic analysis is recommended, including an examination of correlations with different climatic indices influencing the area and an analysis of temporal variability using for example, wavelet-type tools.

**Albeit not accomplished, mentioned in the conclusions as future work**. We agree with the Reviewer-1 that a climatic analysis along with the exploration of those climatic indices affecting the study would definitely provide added value to the present investigation.

However, this ongoing Special Issue of the Ocean State Report is rather restrictive in terms of length (only four figures and 3000 words are allowed) and we are afraid we do not have enough space to compute the suggested analysis that certainly deserves a detailed exploration in the

context of a future complementary paper. Since it is important to underline the necessity to conduct this future investigation, a paragraph has been introduced in section 5 (Conclusions):

*"Complementarily, additional efforts should be devoted to assessing the dominant modes of extreme waves variability and their relationship with the most important climatic indices since this could enhance the prognostic skills of extreme wave events and benefit the adaptation plans in the entire Spanish harbour system."*

[Comment 10] The manuscript neglects the value of tides, even though the tidal range in the Mediterranean can reach up to 1 meter. However, it has been proven that the 99th percentile of the IG is 0.28 m, and of the agitation range is 0.38 m, which is within the order of magnitude of tides in the Mediterranean. Therefore, a sensitivity study of the tidal value in the port should be conducted before neglecting this factor.

**OK, 100% accomplished**. In order to better explain why the impact of both astronomical tides and storm surges on harbour agitation was not considered, we have computed the figure shown below (which is the new Annex 4 in the manuscript). Timeseries of sea level height (blue line) and port agitation (grey line) observations corresponding to the 6 extreme wave events detected before the study case. Observations were provided by Melilla tide-gauge. Astronomical tides and meteorological residuals are represented by the red and green lines, respectively. The vertical dashed black line indicated the peak of the wave storm for each of the 6 events analysed.

[Figure]

Equally, the figure for the E7 event (which is the new Figure 2d in the manuscript):

[Figure]

d) Event E7: 4 April 2022 (21 h)

For the 7 extreme events E1-E7, the following conclusions can be derived:

1) The maximum tidal range observed in Melilla harbour (blue line) is around 40 cm.
2) The surge (green line) due to the storm is negligible for E7 event (below 10 cm), with the meteorological residual being even negative during the six previous episodes (E1-E6).
3) The meteorological residual tends to decrease during the 5-day tome window selected for each event.
4) The evolution of harbour agitation is independent from the tidal phase as the peak of agitation is not coincident with high tides.

Therefore, the paragraph in the manuscript:

"The impact of the last two elements on harbour agitation was not taken into account since: i) the Mediterranean Sea is a microtidal environment with tidal ranges below 1 m (Samper et al., 2022); and ii) the low-pressure core was located in the western side of the Strait of Gibraltar so the storm surge affecting Melilla harbour was negligible (Figure 2, a)."

…has been replaced by:

"As shown in Figure 2d and Annex 3, the impact of the last two elements on harbour agitation during the seven extreme events was negligible due to a number of factors, namely: i) Melilla harbour waters were characterized by a maximum tidal range of 0.40 m; ii) The evolution of harbour agitation was independent from the tidal phase as the peak of agitation was not coincident with high tides; iii) the low-pressure core was located in the Gulf of Cadiz (western side of the Strait of Gibraltar, Figure 2a) so the storm surge affecting Melilla harbour was small (below 10 cm) for E7 event (Figure 2d); iv) the meteorological residual was even negative in the rest of previous extreme events analysed (Annex 3)."

[Comment 11] The third major deficiency in the work is the study of extreme event trends in Melilla port. In Figure 4, it can be seen that for the area marked with a black rectangle, most of the pixels do not show a significant trend for April or July (the two months selected for a comparison between P99 and P50 in Annex 4). In my opinion, it cannot be concluded that the

regression line is significant based on the time series shown in Figure 4 of the manuscript; the series exhibit too much variability.

**OK, 100% accomplished**. The statistical significance at the 90% confidence interval was assessed with the Mann-Kendall test (Mann, 1945; Kendall, 1962), in accordance with similar works previously published (Caloiero and Aristodemo, 2021; Barbariol et. al, 2021). While the trends were statistically significant at the 90% confidence interval for April, June, and October, **in the case of July the observed downward trend was only significant at the 80% confidence interval, as already mentioned in the manuscript (section 4.5, line 304)**. In order to clarify this issue and avoid any misunderstanding, we have removed those panels associated with the month of July. Equally, we have removed Annex 4 from the manuscript (following the Reviewer-1´s suggestion) as the maps do not show any significant trend in the vicinity of Melilla harbour.

With regards to the panels associated with the month of April, the Reviewer-1 claims that "*most of the pixels within the black rectangle do not show a significant trend and therefore, it cannot be concluded that the regression line is significant*". As there are indeed 43 grid point (or pixels) inside the selected black rectangle, we show below the confidence interval associated with each grid point:

[Figure]

Significance of P99 trend: April (1993-2022)

STUDY AREA: 43 GRID POINTS

● 92% - 94% (14 grid points)
● 90% - 92% (3 grid points)
● 88% - 90% (10 grid points)
● 86% - 88% (16 grid points)

More specifically, with numbers:

| | | | | | |
|---|---|---|---|---|---|
| 93.7 % | 93.6 % | 93.4 % | 93.7 % | 92.3 % | 91.3 % |
| 91.7 % | 87.8 % | 87.9 % | 87.9 % | 87.9 % | 89.6 % |
| 87.3 % | 92.6 % | 87.8 % | 87.7 % | 87.9 % | 88.0 % |
| 87.5 % | 88.4 % | 92.8 % | 88.8 % | 87.8 % | 87.5 % |
| 89.2 % | 89.6 % | 93.1 % | 92.3 % | 87.7 % | 87.9 % |
| NaN | 88.4 % | 89.9 % | 93.1 % | 93.9 % | 88.4 % |
| NaN | NaN | 87.5 % | 91.0 % | 93.6 % | 92.3 % |
| NaN | NaN | NaN | 87.9 % | 89.9 % | 92.9 % |
| NaN | NaN | NaN | NaN | NaN | 89.2 % |

Although it is true that there are 26 grid points with a significance interval **below** the 90% and only 17 grid points with a significance interval **above** the 90%, the spatially-averaged confidence interval for the 43 grid point selected is 90.6% and hence the upward trend in the study area can be considered, on average, statistically significant.

With regards to the statement "***the series exhibit too much variability***", we must highlight that, as already specified in the manuscript, trends were calculated using the Sen´s slope estimator of 99$^{th}$ percentile as it was unequivocally proved to be not subject to the influence of extreme values (outliers) and therefore is more consistent than simple linear regression methods (Sen, 1968), as already indicated in the first version of the manuscript.

We guess that perhaps the Reviewer-1 expressed just a subjective opinion based on his/her personal perception after a merely visual inspection. Finally, we would like to highlight that the Reviewer-2 found this approach and the results derived absolutely consistent.

References:

Barbariol, F., Davison, S., Falcieri, F.M.., Ferretti, R., Ricchi, A., Sclavo, M. and Benetazzo, A.: Wind Waves in the Mediterranean Sea: An ERA5 Reanalysis Wind-Based Climatology. Front. Mar. Sci., 8:760614, doi: 10.3389/fmars.2021.760614, 2021.

Caloiero, T. and Aristodemo, F.: Trend Detection of Wave Parameters along the Italian Seas. Water. 13(12):1634. doi:10.3390/w13121634, 2021.

Sen, P.K. Estimates of the regression coefficient based on Kendall's tau. J. Am. Stat. Assoc., 63, 1379–1389, 1968.

[Comment 12] In this work, the analysis of wave height is detailed, while the analysis of wave period is given less attention, even though, for agitation activity, the period is more relevant than the wave height (Eq. 4). This is why in event E7, the agitation is so high compared to the time series, as the period at that time is significantly higher than in the rest of the time series. This fact should be given more emphasis, and the atmospheric conditions that could have caused this remarkable event should be explored.

**OK, 100% accomplished**. Although we absolutely agree with the Reviewer-1, we also must confess that the specific format of this Special Issue did not provide room for deeper analysis of wave period. In this line of thought, we already described in Section 5 ('Conclusions'):

"long-term historical changes in wave period and directionality are receiving increasing attention and should be further analysed to assess their specific impact on harbours operability (Erikson et al., 2022; Casas-Prat and Sierra, 2012). Permanent modifications in the wave direction might result in enhanced wave penetration into the harbour and thereby larger agitation as port protective structures were originally designed to dampen wind and short waves coming from a predetermined sector (Casas-Prat and Sierra, 2012). Likewise, offshore wave period also plays a primary role in the modulation of harbour agitation, as derived from Figure 3 (c-d). As a consequence, any sharp increase in both wave period and SWH$_o$ could lead to the so-called compound extreme events, which are considered to be a major risk of climate change since they can cause more significant damage in port structures than individual extreme events (Velpuri et al., 2023)."

In order to follow the Reviewer-1's suggestion and mitigate this shortcoming, several sentences have been introduced along the manuscript with the aim of emphasising the relevant role played by the wave period in the harbour agitation, especially during E7 extreme event:

Section 4.4 ('Sea state within the port'):

"Additionally, harbour agitation was also importantly modulated by offshore period, as shown in Figure 3 (c-d). Agitation values above the P99 were generally observed when $T_m$ and $T_p$ values were above 4 s and 6 s, respectively. Equally, the highest values of agitation (above 1 m height) were associated with $T_m$ and $T_p$ values above 7 s and 10 s, respectively. It seems reasonable to deduce that the record-breaking harbour agitation (1.41 m) registered during E7 event was caused by the combined effect of unprecedented values of $SWH_o$ (7.32 m) and $T_m$ (9.42 s) in tandem with a very high value of $T_p$ (10.75 s) and a $MWD_o$ (55°) comprised within the aforementioned predominant sector (50°-70°). This clearly shows that compound events (i.e., multiple extreme events that occur simultaneously or in close sequence) are of particular concern for harbour operability, as their individual effects may interact synergistically."

Finally, with regards to the atmospheric conditions that induced this remarkable event, they were explored Figure 2 (a-b), Annexes 2 and 3 along with the Section 4.3 entitled "Driving atmospheric conditions". Furthermore, find below a comprehensive explanation:

Some weather extremes occurring in Europe have been connected with a particular atmospheric flow pattern, known as atmospheric blocking. Blocking episodes have been long acknowledged as a large-scale disturbance (high-pressure systems at the surface) that persist for a long period of time in the middle and high latitude flows (Lupo, 2021). Depending on their location, long-lasting blocking systems also may lead to a shift in the storm track, which influences the occurrence of wind and precipitation anomalies in Europe. Due to these multifaceted linkages, compound events are often observed in conjunction with blocking conditions (Kautz et al., 2021).

There are 3 main types of atmospheric blocks affecting western Europe, as shown in the figure below (extracted from Sousa et al., 2021):
i) The Omega block: a relatively widespread high-pressure area is observed between the midlatitudes and the subtropics, with low pressure systems on its western and eastern flanks (Figure 1a).
ii) The "hybrid" Rex block: during the transition from an Omega to a Rex (pure) shape, the blocking can temporarily exhibit "mixed" patterns (Figure 1b).
iii) The "canonical" Rex block: a north–south dipole pattern characterized by two adjacent (northern) high and (southern) low pressure systems in upper atmospheric levels (Rex, 1950).

[Figure]

In general, blockings in the Northern Hemisphere tend to occur more frequently in winter and spring than in the other seasons. Particularly, eastern North Atlantic blocks are more common in the period from winter to spring (Kautz et al., 2022), in line with previous blocking climatologies

(Barriopedro et al., 2006). Previous works have explored the dynamical links between blocking and the Nort Atlantic Oscillation (NAO), which is the leading mode of atmospheric circulation variability over the Euro-Atlantic sector and is characterized by a seesaw of atmospheric mass between the Iceland Low and the Azores High (e.g., Hurrell and Deser 2009). The NAO appeared as the leading variability pattern during winter, accounting for the 45% of the blocking frequency variance (Barriopedro et al., 2006).

Within this context, we have added the following paragraph to Section 5 ('Conclusions'):

"Therefore, it might be deduced that large-scale atmospheric blocks leading to severe sea states in Melilla tend to be more probable during the winter-to-spring transition period. This outcome is in line with previous blocking climatologies for the eastern North Atlantic (Kautz et al., 2022; Barriopedro et al., 2006). In this context, previous works have also explored the dynamical links between blocking patterns and the Nort Atlantic Oscillation (NAO), which is the leading mode of atmospheric circulation variability over the Euro-Atlantic sector and is characterized by a seesaw of atmospheric mass between the Iceland Low and the Azores High (e.g., Hurrell and Deser 2009). The NAO appeared as the leading variability pattern during winter, accounting for the 45% of the blocking frequency variance (Barriopedro et al., 2006)."

References:

Barriopedro, D., R. García-Herrera, A. R. Lupo, and E. Hernández, 2006: A climatology of Northern Hemisphere blocking. J. Climate, 19, 1042–1063, https://doi.org/10.1175/JCLI3678.1

Hurrell, J.W. and Deser, C. North Atlantic climate variability: The role of the North Atlantic Oscillation. J. Mar. Syst. 78: 28–41. doi: 10.1016/j.jmarsys.2009.11.002, 2009.

Kautz, L.A., Martius, O., Pfahl, S., Pinto, J.G., Ramos, A.M., Sousa, P.M. and Woollings, T. Atmospheric blocking and weather extremes over the Euro-Atlantic sector – a review. Weather Clim. Dynam., 3, 305–336, doi:10.5194/wcd-3-305-2022, 2022.

Lupo AR. Atmospheric blocking events: a review. Ann N Y Acad Sci. 2021 Nov;1504(1):5-24. doi: 10.1111/nyas.14557. Epub 2020 Dec 31. PMID: 33382135.

Rex, D. F. (1950). Blocking action in the middle troposphere and its effect upon regional climate. Tellus 2, 275–301. doi: 10.1111/j.2153-3490.1950.tb00339.x

Sousa, P. M., D. Barriopedro, R. García-Herrera, T. Woollings, and R. M. Trigo: A New Combined Detection Algorithm for Blocking and Subtropical Ridges. J. Climate, 34, 7735–7758, doi:10.1175/JCLI-D-20-0658.1, 2021.

[Comment 13] The conclusion section could focus more on how ports need to revise their security protocols based on studies of extremes in the surrounding area, taking into account the analysis of return periods.

**OK, 100% accomplished**. Although this issue was partially addressed in the last paragraph of the Conclusions section, additional sentences have been added to better underline the need to revise security protocols taking into account the updated return periods:

"Special attention should be focused on the thorough revision of security protocols and the implementation of mitigation plans within the harbour territory based on the updated return periods presented in this work. The design lifetime risk should be recalculated accordingly as coastal structures in the vicinity of the harbour must resist growing stresses during their lifespan and operations, such as wave overtopping, floodings or resonance, to name a few. While the current port layout configuration must be adapted to the increasing frequency and magnitude

of these stressors, future maritime facilities at Melilla harbour should be wisely designed and constructed taking into account these outcomes in order to withstand extreme wave regimes imposed by the changing marine environment (Vanem et al., 2019)."

**2 SPECIFIC COMMENTS**

[Comment 14] L41. Modify the order of the tables according to when they appear in the text.

**OK, 100% accomplished**. Corrected in the manuscript. Table 1 is nowadays mentioned in the text before Table 2.

[Comment 15] L44. Provide the link to the ECCLIPSE website.

**OK, 100% accomplished**. The link to ECCLIPSE website was already added in the reference list (line 456), following the journal guidelines. Further details can be found at: https://www.state-of-the-planet.net/submission.html#references, where it is clearly stated that:

- Webpages
    - Title
    - URL
    - Access date
    - Year (if not the same as access date)

    Example: Copernicus Publications: https://publications.copernicus.org/, last access: 25 October 2018.

[Comment 16] L55. Infragravity waves have a period ranging from 25 seconds to 5 minutes, as indicated by [Munk, 1950].

**OK, 100% accomplished**. Corrected in the text. Reference added to the list.

[Comment 17] L59. Table 4 could be omitted as it is redundant with figures 1 and 3.

**OK, 100% accomplished**.

[Comment 18] L60. In the study area, significant wave heights (SWH) exceed 7m, the same order of magnitude than in the Gulf of Lion.

**OK, 100% accomplished**. The paragraph has been reformulated.

[Comment 19] L110. When does the multi-year wave product reanalysis end and the interim dataset begin?

**OK, 100% accomplished**. The multi-year wave product of the Mediterranean Sea Waves forecasting system contains a reanalysis dataset (from 1 January 1993 to 31 December 2022) and an interim dataset covering the period after the reanalysis until one month before present (i.e, from 1 January 2023 to 1 October 2023). In the present work, only the reanalysis dataset was used. This information is now included in Section 2.2.2 ('Multi-year wave product').

[Comment 20] L129. Why if there are buoy data from 2008, do the authors choose to use them only from 2010?

**OK, 100% accomplished**. Sorry, this was a typo. The data used covered from April 2008 to December 2022. Now the time span has been standardized for all the instrumental datasets: 2011-2022.The motivation for this decision was the following: the Datawell scalar buoy was replaced in April 2010 by a Triaxys buoy able to provide directional information, so we selected entire annual cycles from 2011 to 2022.

[Comment 21] L137. Which spiking method did you use? Were the gaps small enough to ensure that the time series was not totally distorted after processing?

**OK, 100% accomplished**. Only small gaps (not larger than 6 h) in observational dataset were linearly interpolated. The quality control, defined by the CMEMS in situ team (Copernicus Marine In situ Team 2017), was based on a battery of automatic checks performed in real time to flag and subsequently filter inconsistent values. Some of the tests are listed in the table exposed below (and extracted from Lorente et al., 2019), where the spiking test is succinctly described:

**Table 2.** Automatic quality-control checks defined by the Copernicus Marine In situ Team and performed in real time to in situ wave measurements.

| Check | Description |
|---|---|
| Global range test | Gross filter based on observed values for waves. It needs to accommodate all of the expected extremes encountered in the study region. |
| Spike test | Based on the difference between sequential measurements. For the significant wave height, wave period and wave peak period, a value is flagged when the difference exceeds 3 m, 4 s and 15 s, respectively, for the Atlantic ocean. |
| Stuck value test | A wave parameter should not remain in the same value for more than 12 hours with more than 50% of data not null and valid. |
| Rate of change in time | Based on the difference between the current value with the previous and next ones. |

Obviously, the thresholds used in the spike test for the Mediterranean partially differ from those above exposed for the Atlantic Ocean: for the significant wave height, wave period and wave peak period, a value is flagged when the difference exceeds 3 m, 4 s and 10 s, respectively, for the Mediterranean Sea. An additional sentence has been added to Section 2.1.2 to better clarify this.

References:

Copernicus Marine In situ Team. 2017. Copernicus in Situ TAC, Real Time Quality Control for WAVES. Toulouse, France: Copernicus in situ TAC, 1–19. doi:10.13155/46607.

Lorente, P.; Basañez Mercader, A.; Piedracoba, S.; Pérez-Muñuzuri, V.; Montero, P.; Sotillo, M.G.; Álvarez-Fanjul, E. Long-term skill assessment of SeaSonde radar-derived wave parameters in the Galician coast (NW Spain). Int. J. Remote Sens. 2019, 10, 9208–9236

[Comment 22] L140. Pearson correlation coefficient.

**OK, 100% accomplished**. Specified in the manuscript.

[Comment 23] Eq 2 and 3. Why do you use the sample variance instead of the population variance?

**OK, 100% accomplished**. Apologies, this was a typo. We obviously used the population variance. The equation has been modified accordingly.

$$\bar{x} = \frac{1}{N}\sum_{i=1}^{N} x_i \tag{1}$$

$$\sigma = \sqrt{\frac{1}{N}\sum_{i=1}^{N}(x_i - \bar{x})^2} \qquad (2)$$

$$\text{Correlation} = \frac{1}{N}\sum_{i=1}^{N}\left(\frac{x_i - \bar{x}}{\sigma_x}\right)\left(\frac{y_i - \bar{y}}{\sigma_y}\right) \qquad (3)$$

**Update:** Reviewer-2 has suggested replacing these three equations by a reference:

"[Comment 11] *Lines 141-143, equations (1),(2),(3) are the well know definitions of mean, standard deviation and correlation. Is it really necessary to introduce them here? Or could you just give a reference of a statistical or methods book/paper*".

Therefore, we have replaced the equations by a reference in the manuscript.

[Comment 24] L155. The correct reference was Stockdon et al. (2006), not Inch et al. (2017).

**OK, 100% accomplished**. Replaced in the text.

[Comment 25] L160. Specify the data that were used.

**OK, 100% accomplished**.

[Comment 26] L173. Specify the time span.

**OK, 100% accomplished**.

[Comment 27] L180. Why do you consider data for wave directions only for the period between 2011 and 2022?

**OK, 100% accomplished**. As already stated in section 2.4: the Datawell scalar buoy was replaced by a Triaxis buoy able to provide directional information in 2011. Text amended to better clarify it.

[Comment 28] L186. How do you calculate the exceedance threshold and the time between two independent storms?

**OK, 100% accomplished**. That paragraph has been expanded to better clarify the approach adopted.

• With regards to the exceedance threshold, we followed the approach proposed by Harley (2017) and Fanti et al. (2023) for coastal storm analysis: the most pragmatic approach is to simply set the threshold according to the 95[th] percentile of the significant wave height dataset.

**References:**

Harley, M. Coastal storm definition. In Coastal storms: processes and impacts 1–21 (John Wiley & Sons, 2017).

Fanti, V., Ferreira, Ó., Kümmerer, V. and Loureiro, C.: Improved estimates of extreme wave conditions in coastal areas from calibrated global reanalyses. Commun Earth Environ 4, 151 (2023). https://doi.org/10.1038/s43247-023-00819-0

• With regards to the time between two independent storms, there is some subjectivity in how a time series is partitioned into separate storms. The broadly accepted criteria used to define independent storms typically state that the time between the wave height peak of two adjacent storms must be larger than some minimum value. Such minimum value in the North Atlantic is usually chosen considering that the average lifetime of extra-tropical cyclones is 3 days (Trigo et

al. 1999). For instance, the most intense activity period of Storm Gloria ranged between 20 and 23 January 2020 (Amores et al., 2020). Within this context, Mackay and Johanning (2018a and 2018b) showed that values of storm peak separated by 5 days were effectively independent: "Given these observations, defining storms as local maxima in SWH in a 5-day window appears to be sufficient to ensure independence. In this context, changing the minimum separation affects the isolation of lower peaks which have little influence on the extremes. A separation time of 5 days is also reasonable based on physical arguments, since peaks separated by 5 days will correspond to waves generated from separate low-pressure systems". Therefore, in the present work, storms were defined using a minimum temporal separation of 5 days between adjacent peaks, as suggested by Mackay and Johanning (2018a and 2018b).

**References:**

Amores, A., Marcos, M., Carrió, D. S., and Gómez-Pujol, L. Coastal impacts of storm gloria over the northwestern mediterranean. Nat. Hazards Earth Syst. Sci. 20, 1955–1968. doi: 10.5194/nhess-20-1955-2020, 2020.

Mackay, E. and Johanning, L. Long-term distributions of individual wave and crest heights, Ocean Eng., 165, 164-183, 10.1016/j.oceaneng.2018.07.047, 2018a.

Mackay, E. and Johanning, L. A generalised equivalent storm model for long-term statistics of ocean waves, Coastal Engineering, 140, 411-428, doi: 10.1016/j.coastaleng.2018.06.001, 2018b.

Trigo, I.F., Davies, T.D. and Bigg, G.R. Objective climatology of cyclones in the Mediterranean region. J Clim 12(6):1685–1696. doi:10.1175/1520-0442(1999)0122.0.CO;2, 1999.

[Comment 29] L233. Could you provide spectra to demonstrate how the infragravity waves dominate the energy during the analysed events?

**Not accomplished.** We might provide spectra in the next iteration with both reviewers. At the present stage, we have not provided them since i) the total number of Figures (4 + 5 additional annexes) is already significantly high; ii) Reviewer-2 has not required that ancillary information.

[Comment 30] L235. It is not possible to see all these results in Table 6. Could you display them graphically?

**OK, 100% accomplished**. All these results are now exposed in Figure 3e. Furthermore, since the thresholds for port management are also indicated in this panel, Table 6 has been removed from the document to make easier for the reader to follow the narrative.

[Comment 31] L243. Would you mean "20 minute time-series"?

**OK, 100% accomplished**. Corrected in the manuscript.

[Comment 32] L253. Instead of "the 655 hourly", it would be clearer to mention the time span.

**OK, 100% accomplished.** Clarified in the text.

[Comment 33] L268. How do you calculated the "monthly P99"? Is it the P99 of all the January data (February, March, etc.)? Or is it the mean value of all the P99 from all the January, February, etc. months?

Yes, the first option: the 99$^{th}$ percentile (P99) value for January was computed considering all January hourly data comprised between 1993 and 2022 (green line in old Annex 3c) and

comprised between 2009 and 2022 (blue line in old Annex 3c). Notwithstanding, Annex 3b and 3c have been removed, in line with [Comment 5] from Reviewer-1.

Nowadays, in the new version of the manuscript the monthly P99 was only calculated for MED reanalysis in:

iii)     Figure 4: Trends of P99 of $SWH_m$
iv)     Annex 5 (b-e): maps of 50th and 99th percentiles of $SWH_m$ for April and July.

[Comment 34] L313-321. These points should be included within the introduction section.

**OK, 100% accomplished**. The key points outlined in L313-321 have been also inserted in specific parts of the introduction.

[Comment 35] L336. It is not the "percentile's method", it is the peak over threshold.

**OK, 100% accomplished**. "Percentile´s method" was replaced by "POT method". The methodology has been reformulated accordingly.

[Comment 36] L421. Berta, et al. (2020) should appear after Bensoussan, et al. (2019).

**OK, 100% accomplished.**

[Comment 37] Annex 3. Adjust all the colorbars, as P99 seems smaller than P50.

**OK, 100% accomplished**. We guess Reviewer-1 meant Annex 4. The colorbar has been modified to solve this issue.

[Comment 38] Annex 5. Consider removing this annex because the most of the pixels show non-significant trend values.

**OK, 100% accomplished. R**emoved from the manuscript. The references to this Figure have been also deleted.

[Comment 39] Walter H Munk. On the wind-driven ocean circulation. Journal of meteorology, 7(2):80–93, 1950.

**OK, 100% accomplished**. Added to the references list.

**RESPONSE TO REVIEWER 2**

1. General comments.

The manuscript discusses the extreme wave event that affected the Melilla coast and harbour between the 4th and 5th April 2022, with unprecedented impact. The observations collected by an offshore wave buoy and by a tide gauge installed inside the Melilla port are used to characterize the extreme wave events that affected the area from 2008 to 2022, to evaluate the impact of these events on the conditions inside the port of Melilla (separating IG wave motions and agitation due to wind waves) and to show the distinctive extreme nature of the April2022 event when compared with the previous extreme events observed. Estimates of the return period for these events show the importance of retaining the April 2022 event in evaluation of return periods for the planning of coastal structures or harbour operations. A regional wave model (WAM based) for the Iberian area is used to detail the wave conditions at affecting the Melilla area on the 4thApril2022, at the peak of the storm. ERA5 reanalysis are used to characterize the meteorological conditions associated with the extreme wave events observed in the Melilla area from 2008 to 2022. A reanalysis dataset from a regional model (WAM-based) for the Mediterranean Sea, covering the period 1993-2022, is used to provide the long-term perspective of evolutions of wave conditions (in terms of significant wave height) in the Alborán Sea.

The central subject of the paper, the extreme events affecting the SW Mediterranean coast and its impacts on the harbour conditions, addresses relevant scientific and societal questions.

The manuscript title reflects the overall content of the paper, highlighting the focus on the extreme wave event that affected the Melilla harbour in April 2022.

The abstract does provide a concise and complete summary of the key questions focused, of the work developed in the paper and main results.

Given the scientific and societal relevance, the new data that is explored, the important integration between observations and modelling results, it is the opinion of the referee that the paper should be considered for publication after some improvements on the structuration, presentation of matters and discussion, as indicated in the detailed analysis below.

Many thanks to Reviewer-2 for the detailed review and the number of useful tips provided. Please find below a thorough point-by-point response with the hope of improving the quality of the document to make it acceptable for final publication.

It is worthwhile mentioning that we have fully (partially) accomplished 65 (1) from the 67 suggestions provided by Reviewer-2, which constitutes the 97%.

Just in case the Reviewer-2 is not familiarized with the 8th Ocean State Report initiative, we would like to clarify that it is characterized by some specific limitations in terms of length (up to 3000 words) and maximum number of figures (4).

Finally, we would like to inform Reviewer-2 that the abstract has been rewritten to better clarify the main results derived from the study, following Reviewer-1 request ([comment 1] "*The abstract should be rewritten to provide a more comprehensive explanation of all the values presented by the authors*").

1. Specific Comments

2.1 Text

[Comment 1] Lines 90-95: The regional WAM-based wave forecast system described in section 2.2 is used in figure1.a to characterize the wave conditions offshore the Iberian Atlantic and Mediterranean areas and in figure 1.b to characterize the wave conditions that affected the Melilla area at 21:00 of 4 April 2022, during the peak of the storm. A more extensive use of the results from this model could have been conducted. The wave directions could be included in figure 1b, in this way illustrating eventual refraction effects that could be associated with the shelf topography in the Melilla area and that could, in some way, be relevant to interpret the offshore buoy measurements or the effects inside the harbour.

**100% accomplished**. A more extensive use of the model results has been conducted to characterize the wave conditions during the record-breaking storm. Wave directions have been included in Figure 1b as requested, along with the bathymetric contours.

[Figure]

We must point out that now Figure 1a is nowadays based on ERA-5 reanalysis outcomes (not on our WAM wave forecast model) due to [Comment 4] from Reviewer-1: "*Why is the "wave forecast model" of Puertos del Estado used? Would not it be more consistent to use the same database for atmospheric and oceanic variables (such as ERA5)?*" Therefore, for consistency reasons, all those maps (of sea level pressure, wind at 10 m height and significant wave height) covering the regional domain (from Canary Islands to Ireland) are now based on ERA-5 reanalysis. Accordingly, our wave forecast model has been deleted from the table of products used (Table 1).

Figure 1b is nowadays based on MED reanalysis outcomes because: i) it has higher horizontal resolution than ERA-5 reanalysis (as shown in Table 2) so a larger amount of wave vectors can be plotted; ii) this reanalysis is forced with ERA-5 atmospheric fields, so the consistency is ensured.

Finally, with regards to the wave direction, we must say that the MED regional reanalysis have a 1/24° horizontal resolution which, albeit significantly high, is still too coarse to accurately capture fine details at coastal/littoral scales such as refraction effects related to the shelf topography in the Melilla area.

Notwithstanding, the MED reanalysis (Figure 1b) agree in terms of wave direction with the information already provided by Melilla coastal buoy in Table 3 (mean incoming direction: 55° or in other words, propagation direction: 235°).

Within this context, in Conclusions section we already indicated the following: *"future works should include the implementation of a dynamical downscaling methodology to improve wave reanalysis accuracy at finer coastal scales (Vannucchi et al., 2021). Of course, this would necessarily require finding the right trade-off between adequate spatial resolutions and the available in-house computational resources"*.

[Comment 2] Also, if results from this model are available for the global period 2008-2022, they could be used to characterize the wave field in the area during the different extreme events that were observed, perhaps providing some relevant information regarding different impacts of these events on the area of the harbour.

**OK, 100% accomplished**. As previously indicated, we have replaced the wave maps from our model by outcomes from ERA-5 reanalysis (at regional scale) and MED reanalysis (at coastal scale for the area around Melilla harbour).

[Figure]

The ERA-5 reanalysis dataset covers from 1940 so we plotted the significant wave height field for the 6 previous extreme events (new Annex 1). As shown above, the visual resemblance between all these events is noticeable and share some common features, namely: the peak of wave height over the entire Alborán Sea. A secondary peak can be found at the westernmost part of the Strait of Gibraltar (over the Gulf of Cádiz) for E1, E2, E4 and E5 episodes, while E3 barely shows such a peak. In the case of E6 event, the peak over the Alborán Sea is not so high (around 4 m) and affected broader areas of the SW Mediterranean Sea.

Equally, the maps of SWH and wave propagation direction in the vicinity of Melilla harbour for the 6 extreme events (small maps exposed in the lower right corner of each panel) reveal that all of them shared pretty similar directional features.

[Comment 3] Line 94: Although readers can access the product documentation indicated in Table 1 it would facilitate the reading of the paper if indication of the geographical area covered by the model and of the spatial resolution of model outputs is provided.

**OK, 100% accomplished**. A new Table 2 has been introduced in the manuscript where columns 3 and 7 provide information about the models' spatial coverage and horizontal resolution, respectively.

[Comment 4] Line 101: The indication of the geographical position of the wave buoy would facilitate the reader.

**OK, 100% accomplished**. The geographical positions of both the coastal buoy and the tide-gauge are gathered in the new Table 2 (column 3). The position was not indicated explicitly in the main body of the manuscript (section 2) to avoid redundancies with Table 2.

[Comment 5] Lines 101-102: Radar tide gauge data was, apparently, available since October 2007. Could you justify why you have not extended the analysis of these measurements to the longer period 2008-2022, that could be consistent with the analysis of the offshore wave buoy?

**OK, 100% accomplished**.

Following also Reviewer-1´s request ([Comment 3] "*The time span for the different datasets should be standardized*"), we have standardized instrumental datasets as shown in the new Table 2 with the aim of clarifying the situation:

iii) **1993-2022** is the time span for both the **regional MED wave reanalysis and ERA5 reanalysis**.
iv) **2011-2022** is now the time span for **all in situ observational data** (from both the tide-gauge and the coastal buoy). Data section, Methodology section and diverse figures have been updated accordingly. In the new revised version of the manuscript, the time span employed and cited is always 2011-2022. The motivation for this decision was the following: the Datawell scalar buoy was replaced in April 2010 by a Triaxys buoy able to provide directional information, so we selected entire annual cycles from 2011 to 2022.

[Comment 6] Lines 97-102, Section 2.3 Tide Gauges: could you better describe here how the IG and agitation bands were defined from the 2-Hz data?

**OK, 100% accomplished**. A brief paragraph has been inserted into the new section 2.1.2 ("Melilla port tide-gauge") of the manuscript to better clarify this point. Raw 2-Hz data contain information of sea level oscillations with periods above 1 s, essentially capturing all sea surface height variability including waves, high-frequency sea level oscillations (HFSLO) and tides. HFSLO with periods between 30 s and 1 h were extracted from raw 2-Hz data by means of fit-for-purpose high- and low-pass digital filters, always trying to: (1) fix the width of the transition band from pass to stop as narrow as possible and (2) maximize the filter stability.

First, a subsampled time series with a cadence time of five minutes was created. This subsampled time series was used to obtain sea level oscillations with periods over 1 h using a 10th-order Chebyshev IIR low-pass filter with 1/3600 cut frequency (periods above 1 h). Then, wave agitation was obtained using an 8th-order Butterworth high-pass digital filter over the 2-Hz raw data with a cut frequency of 1/30 (periods below 30 s). Finally, the signal of HFSLO (with periods between 30 s and 1 h) was obtained by subtracting the two previous time series from the 2-Hz raw data signal. Further details can be found in the figure below:

[Figure]

Figure extracted from García-Valdecasas et al (2021): a) Transfer function of the low-pass filter used to filter periods below 1 h; b) Transfer function of the high-pass filter used to obtain wave agitation; c) Comparison of raw data and filtered HFSLO energy spectrums.

Finally, a simplified four band energy spectrum was also calculated to facilitate the understanding of the energy distribution in the HFSLO band:

1) 30 s < T < 5 min (infragravity waves)
2) 5 min < T < 15 min
3) 15 min < T < 30 min
4) 30 min < T < 1 hour

[Comment 7] Line 129: "…for two different periods (a) 2010-2020 (before the record-breaking storm and (b) 2010-2022…". Why didn't you use the complete dataset that was available 2008-2022?

**OK, 100% accomplished**. Sorry, this was a typo. In the first version of the manuscript, the dataset covered from April 2008 to December 2022. As above indicated, we have standardized instrumental datasets: in the new version of the manuscript, 2011-2022 is the time span for all in situ observational data (from both the tide-gauge and the coastal buoy). Accordingly, the return periods were recomputed for 2011-2021 (before the record-breaking storm) and for 2011-2022 (after the record-breaking storm).

[Comment 8] Line 131: "…Three-Parameter Weibull…", assure consistency with nomenclature introduced below (here Three)

**OK, 100% accomplished**. "3-parameter Weibull distribution" has been replaced by "three-parameter Weibull distribution" along the entire document.

[Comment 9] Lines 137-138: Gaps linearly interpolated no matter the gap extension or did you used a limit time gap?

**OK, 100% accomplished**. Only small gaps (not larger than 6 h) in observational dataset were linearly interpolated. Already clarified in the manuscript.

[Comment 10] Lines 137-138: Should this paragraph be moved to section "2. Data", given that there you also provided some information regarding the use of tide gauge data to infer wave conditions inside the port?

**OK, 100% accomplished**. The paragraph has been moved to section 2.1.1

[Comment 11] Lines 141-143, equations (1),(2),(3) are the well know definitions of mean, standard deviation and correlation. Is it really necessary to introduce them here? Or could you just give a reference of a statistical or methods book/paper?

**OK, 100% accomplished**. The three equations have been replaced by a reference to Emery and Thompson (2001) in section 3 ("Methodology"):

"The statistical metrics used in the present study to compare two data sets included the mean, the standard deviation, and the Pearson correlation coefficient (Emery and Thompson, 2001)."

Reference:

Emery, W.J. and Thompson, R.E. Data Analysis Methods in Physical Oceanography, Elsevier Science, ISBN 9780080477008, 654 pages, Amsterdam, 2001.

[Comment 12] Line 147:

(a) "...the spectra of 2 Hz Sea level oscillations measured by the tide gauge revealed…" should become "...the spectra of 2 Hz sea level oscillations measured by the tide gauge (not shown) revealed…".

**OK, 100% accomplished**. That sentence has been rephrased as follows:

"Since the spectra of 2 Hz sea level oscillations measured inside the harbour by Melilla tide gauge (not shown) revealed…"

(b) Also, this phrase will fit better in section "4. Results". In the present section ("3. Methodology") it would be more relevant to mention that spectra of the 2 Hz data was built to identify energetic sea level variability inside the port, and in the following section ("4. Results", particularly on "4.4 Sea state inside the port") then to refer that the spectra revealed a high energy content in the IG band.

**OK, 100% accomplished**. The sentence was moved to section 4.4 ('Sea state within the port').

Additionally, the following phrase was added to Section 3 ('Methodology'):

"spectra of the 2 Hz data (not shown), generated to identify energetic sea level variability inside the port, were dominated by energy in the IG band during these storms."

[Comment 13] Line 148-149: "As the IG energy in the nearshore has been documented to be positively correlated with offshore SWH… ". The phrase seems to suggest that in the two references indicated the IG energy measured by the Melilla port tide gauge was found to be positively correlated with the SWH measured by the offshore Melilla buoy, which is not the case. The two references indicated in the text focus on the nearshore (surf zone) areas of beaches with a large variety of conditions but are not including harbour areas.

**OK, 100% accomplished**. The paragraph has been reformulated in Section 3 with the aim of avoiding any confusion.

[Comment 14] Line 150: "…. a scatter plot was computed…". A scatter plot between what and what?

**OK, 100% accomplished**. Since the entire paragraph has been reformulated in Section 3, that specific sentence is no longer in the document.

[Comment 15] Line 154: SWH was already defined as the significant wave height, but we are using this notation both for the SWH calculated from wave measurements and for the SWH obtained from models. Perhaps it could help the reader if this is distinguished. For example, in this line we are referring to SWH calculated from wave measurements while a few lines below (e.g., line 161) we are referring to SWH obtained from models. It could help readers if a distinction is made between significant wave heights obtained from observations collected by the wave buoy (noted, for example, $SWH_o$) and significant wave heights obtained from models (noted, for example, $SWH_m$)

**OK, 100% accomplished**. The proposed notation for the significant wave height has been used along the entire manuscript (both text and figures): $SWH_o$ for in situ observations and $SWH_m$ for model estimations. Further details can be found in the new Table 2.

[Comment 16] Line 156:

(a) Equation (4), suggesting that IG height as runup is proportional to $(SWH\ L)^{1/2}$ was proposed by Stockdon et al. (2006) (their equation 18). Inch et al. (2017) instead suggested that IG wave height could be better predicted from offshore wells using the relation $SWH^2*T_p$ , instead of the previous one.

**OK, 100% accomplished**. Corrected in section 3 ('Methodology'), where a new paragraph has been inserted:

"Significant efforts have been previously devoted to analysing the connexion between offshore wave parameters and IGE, either at the shore (in the form of run-up) or in the nearshore area (surf zone). While Guza and Thornton (1982) found that the IG component of wave run-up increased linearly with increasing offshore $SWH_o$, Stockdon et al. (2006) concluded that the IG

component scaled better with $SWH_o \cdot L$ (where L represents the deep-water wavelength) and was actually independent of the foreshore slope. In the same line, Senechal et al. (2011) reported that IG wave run-up during extreme storm conditions was significantly less scatter when correlated with $SWH_o \cdot L$ than with $SWH_o$ only. By contrast, Inch et al. (2017) reported that nearshore IG waves were best predicted using an offshore forcing parameter that is proportional to $SWH_o^2 \cdot T_p$. These contradictory findings reveal that further research on the subject is required and suggest that nearshore IGE is unlikely a function of any single environmental factor (Lashley et al., 2020)."

Finally, new outcomes derived from both approaches have been included In section 4 ('Results').

(b) The studies of Stockdon et al. (2006) and Inch et al. (2017) focus on IG motions near sloping beaches. Can you comment about the validity of using their results in an area such as the harbour of Melilla?

**OK, 100% accomplished**. Two paragraphs have been added to the manuscript:

"Significant efforts have been previously devoted to analysing the relationship between offshore wave parameters and IG wave energy, either at the shore (in the form of run-up) or in the nearshore area (surf zone). While Guza and Thornton (1982) found that the IG component of wave run-up increased linearly with increasing offshore $SWH_o$, Stockdon et al. (2006) concluded that the IG component scaled better with $SWH_o \cdot T_m^2$ and was actually independent of the foreshore slope. In the same line, Senechal et al. (2011) reported that IG wave run-up during extreme storm conditions was significantly less scatter when correlated with $SWH_o \cdot T_m^2$ than with $SWH_o$ only. By contrast, Inch et al. (2017) reported that nearshore IG waves were best predicted using an offshore forcing parameter that is proportional to $SWH_o^2 \cdot T_p$. These contradictory findings reveal that further research on the subject is required and suggest that nearshore IG wave energy is unlikely a function of any single environmental factor (Lashley et al., 2020).

While the four aforementioned field studies focused on low-to-mild-sloping sandy beaches, in the present work (focused on harbours), a rough approximation approach based on two simplifications was adopted: i) local slope effects were not included, in line with Stockdon et al. (2006); ii) IG wave energy registered at Melilla tide gauge was scaled with $SWH_o$, $SWH_o \cdot T_m^2$ and $SWH_o \cdot T_p^2$ although the former dataset is affected by wave–structure interaction processes (diffraction and reflection, to name the main ones) which are not so relevant in sandy beaches."

References:

Guza, R. T., and Thornton, E. B.: Swash oscillations on a natural beach, Journal of Geophysical Research: Oceans, 87(C1), 483-491, 1982.

Inch, K., Davidson, M., Masselink, G. and Russell, P.: Observations of nearshore infragravity wave dynamics under high energy swell and wind-wave conditions, Continental Shelf Research, 138, 19-31, doi:10.1016/j.csr.2017.02.010, 2017.

Lashley C.H., Bricker J.D., Van der Meer, J., Altomare, C. and Suzuki, T.: Relative Magnitude of Infragravity Waves at Coastal Dikes with Shallow Foreshores: A Prediction Tool. Journal of Waterway, Port, Coastal, and Ocean Engineering, 146 (5), doi:10.1061/(ASCE)WW.1943-5460.00005, 2020.

Senechal, N., G. Coco, K. R. Bryan, and R. A. Holman: Wave runup during extreme storm conditions, J. Geophys. Res., 116, C07032, doi:10.1029/2010JC006819, 2011.

Stockdon, H.F., Holman, R.A., Howd, P.A. and Sallenger Jr, A.H.: Empirical parameterization of setup, swash and runup. Coast. Eng., 53, 573–588, 2006.

[Comment 17] Line 158: In the work of Stockdon et al. (2006) the experimental conditions presented waves not exceeding more than 3.5m. In that case, the measurements from an offshore buoy located over the bathymetric of 20m could be used as characterizing the waves in deep water. In the present work, the focus is on extreme events, with waves with heights exceeding 6 or 7m. In this case, can you justify that measurements from a buoy deployed a 14m depth can indeed be used as representative of waves propagating in deep water and, particularly, that the deep-water limit of the dispersion relation that is used in equation (5) can indeed be used to calculate the wavelength in equation (4)?

**OK, 100% accomplished**. Yes, we can justify the use of equation [5] which represents the wave length (L) in deep waters. Firstly, we would like to clarify that:

- Melilla buoy is moored at d=15 m depth (not 14 m).
- In the work of Stockdon et al. (2006), Hs and T were recorded hourly at a Waverider buoy located in ~18 m of water (Section 3.3 "Field experiments" in page 577). It is the depth of the different sites selected (to conduct the comparisons) which varied between 7 and 20 m (section 3.2 "Environmental parameters" in page 576).
- Respect the wave climate at Melilla buoy, the 98th percentile of significant wave height for 2011-2022 is set to 2.6 m.

There are 3 different ways of computing L depending on the type of wave, which in turn is defined by the value of the relative depth d/L :

| Relative Depth d/L | wave type | wave celerity | wave length |
|---|---|---|---|
| $d/L < .05$ | shallow water wave | $\sqrt{gd}$ | $\sqrt{gd}\,T$ |
| $.05 < d/L < .50$ | intermediate depth wave | $\sqrt{\dfrac{gL}{2\pi}\tanh\left(2\pi\dfrac{d}{L}\right)}$ | $\dfrac{gT^2}{2\pi}\tanh\left(2\pi\dfrac{d}{L}\right)$ |
| $d/L > .50$ | deep water wave | $\sqrt{\dfrac{gL}{2\pi}}$ | $\dfrac{gT^2}{2\pi}$ |

We have computed the relative depth d/L for shallow waters and deep waters using the mean wave period (T) provided by Melilla coastal buoy:

[Figure]

As it can be observed, assuming the shallow water equation for the wave length we get a time series for the relative depth (top) characterized by a mean of 0.34 and a standard deviation of 0.08. The premise d/L < 0.05 is never fulfilled for the 12-year period here analysed (2011-2022). Conversely, if we assume the deep-water equation for the wave length, we get a d/L time series where the mean value is 0.78, the standard deviation is 0.37 and the premise d/L > 0.05 is fulfilled the 78% of the time for the period 2011-2022. The remaining 22% corresponds to d/L values below 0.5 and above 0.05 (i.e., intermediate depth water). Therefore, the deep-water wavelength approximation is broadly valid. A brief sentence has been added in Section 3 (Methodology) to shed light on this point:

"Although Melilla coastal buoy is moored at 15 m depth (d), the deep-water approximation is broadly accepted since the relative depth (defined as d/L) is above 0.5 the 78% of the time during 2011-2022 (not shown). Therefore, the wavelength can be defined as $L=(g \cdot T_m^2)/2\pi$, where the gravity acceleration g is 9.8 m·s$^{-2}$. As a consequence, we can derive from point ii) that IGE was scaled with $SWH_o^2$, $SWH_o \cdot T_m^2$ and $SWH_o^2 \cdot T_p$."

[Comment 18] Line 172-179: The extreme events analysis is developed here in terms of SWH but can you also indicate the values of maximum wave height measured by the wave buoy offshore during the extreme storms?

**OK, 100% accomplished**. Values of maximum wave height during the extreme events have been added to the new Table 3. Furthermore, time series of maximum wave height has been also added to Figure 1c.

[Comment 19] Line 173: "… and derived from a long-term time series…", perhaps change to "… and derived from a long-term (14 years) time series…" to become clearer when the term "long-term mean and extreme wave climate" is latter use to refer model results for the 1993 to 2022 period.

**OK, 100% accomplished**. The paragraph has been reformulated. Now the sentence is:

"The P99.9 of SWH$_o$ (set to 4.45 m and derived from the 12-year time series provided by Melilla coastal buoy) was used as threshold to detect the most extreme wave events."

[Comment 20] Lines 173-174: "and derived from a long-term time series provided by a Melilla coastal buoy (Figure 1, b-c)…". The long-term time series that is referred in the text is presented in figure 1d so this should be indicated in the text.

**OK, 100% accomplished**. Replaced by "(Figure 1c)" as Figure 1 has been restructured.

[Comment 21] Line 174: "… was abruptly exceed during 42 hours." The duration of the extreme conditions above P99 is an important aspect but the reader cannot derive it from the presented tables or pictures. This could be included as an additional column in the table that reports the main characteristics of the 7 extreme events presented in the offshore buoy time series (table inserted in figure 1). This table could be moved to the "Table" section, allowing that more information could be added.

**OK, 100% accomplished.** A new Table 3 was inserted in the manuscript with additional information to better characterize the extreme events. In particular, a column defined as "Time above P99 (h)" has been created to inform about the duration of the extreme conditions above the 99$^{th}$ percentile.

[Comment 22] Lines 175: "…. Coincident with a maximum value of the mean wave period (9.42s) …" these are results that are presented in figures 1d,e and so the reference to these figures should be included here.

**OK, 100% accomplished**. The reference "(Figure 1, d-e)" was included at the end of the sentence, which has been reformulated:

"The storm that hit Melilla harbour during the 4$^{th}$-5$^{th}$ of April 2022 (E7) exhibited unprecedented values for each wave parameter: the peak of SWH$_o$ (7.32 m) was coincident with the greatest values of MWH$_o$ (12.11 m) and T$_m$ (9.42 s), jointly beating all previous historical records (Figure 1, c-d)."

[Comment 23] Line 180: "… were NE … ". To be consistent, please include the angular sector for NE as you have done for NE-E

**OK, 100% accomplished**.  Indeed, in the sentence "the prevailing incoming wave directions were NE and NE-E (58 ± 37)°", the angular sector (58 ± 37)° represents the mean direction + standard deviation for the entire dataset, not the angular sector associated with NE-E. In order to avoid any misunderstanding, the sentence has been rephrased as follows:

"From a directional perspective, the prevailing incoming wave directions during the 12-year period analysed were NE (41%) and NE-E (43%), with an overall associated mean value of 58° ± 37° (Figure 1e)."

[Comment 24] Line 183: It would be interesting if you could include some information regarding how the direction of the extreme events compares with the prevailing incoming wave directions for the period 2011-2022.

**OK, 100% accomplished**. In addition to the wave rose for the entire range of wave heights (Figure 1e), a new wave rose has been plotted (Figure 1f) where only SWH values above 3 m (99$^{th}$ percentile) have been taken into account (see below). As it can be deduced, the prevailing incoming sector for the most extreme SWH events is NE-E (72%), while the remaining 28% correspond to the NE sector. A brief sentence has been included in the manuscript:

"For extreme wave events with SWH$_o$ above P99 (3.01 m), the predominant incoming wave direction was NE-E with a 72% of occurrence, whereas the remaining 28% corresponded to the NE sector (Figure 1f)."

[Figure]

[Comment 25] Lines 185-188: The description of how the return periods was calculated would be better located in section 3 (Methodology), in the text block dedicated to the return period associated with the extreme episodes (starting on line 123). Here

**OK, 100% accomplished**. That piece of text has been moved to section 3 ('Methodology')

[Comment 26] Line 192: "…3-parameter Weibull probability…" assure consistency with nomenclature introduced above (e.g., in line 131)

**OK, 100% accomplished**. "3-parameter Weibull distribution" has been replaced by "three-parameter Weibull distribution" along the entire document.

[Comment 27] Lines 188-194: This text include the results for the return periods that are associated to each one of the 7 extreme events indicated in the table that is inserted in figure1. These results are not presented in Table 5, which presents the return periods for different classes of SWH (3m, 4m …, 8m). It would be useful for the reader that the return periods calculated for each of the 7 extreme events could appear as a separated column in the table inserted in figure 1.

**OK, 100% accomplished**. The return period for each of the seven extreme events are shown now in Table 4. We tried to gather all this information in a column of the new Table 3, as suggested by Reviewer-2, but there is not enough space. Please accept our apologies.

[Comment 28] Line 201: Should you use SI units consistently along the text (in this case, 2 x 10$^{-3}$ Pa/m)? In some parts of the text the atmospheric pressure is reported in hPa, here differences in atmospheric pressure appear in Pa.

**OK, 100% accomplished**. We have checked the validity of the units used in the manuscript.

We would like to clarify that, although the Pascal is the standard SI unit for pressure, the Hectopascal is broadly accepted **as SI-derived unit** for measuring atmospheric or barometric pressure. In particular, we could find the following paragraph in "State of the Planet" journal website:

"Units: for units of physical quantities, the metric system is mandatory and, wherever possible, SI units should be used. **Hereby, we differentiate between SI base units, SI-accepted units, and SI-derived units**. Regarding the abbreviation of such units, SI base units and SI-accepted units must be abbreviated in conjunction with numbers (e.g. the velocity is 10 km h$^{-1}$) and must be written out without numbers (e.g. the velocity is given in kilometres per hour). SI-

derived units must also be written out when they do not contain a number**. If they contain numbers, the abbreviation is preferred where possible (e.g. the average atmospheric pressure is 1013 hPa**), but authors can decide not to abbreviate them if no abbreviation is commonly used (e.g. the distance is 237 nautical miles). Regarding the notation, if units of physical quantities are in the denominator, contain numbers, and are abbreviated, they must be formatted with negative exponents (e.g. 10 km h$^{-1}$ instead of 10 km/h)."

Further details can be found at: https://www.state-of-the-planet.net/submission.html#math

[Comment 29] Lines 208-209: "… where easterlies blew persistently…", could you in some way quantify what you mean persistently (how many days?) and if this persistence is similar for the different events or not?

**OK, 100% accomplished**. Since Melilla coastal buoy and Melilla tide-gauge do not have wind sensor, we could not quantitatively quantify the persistence. However, the persistence of strong winds can be indirectly estimated from the number of hours that $SWH_o$ registered at Melilla coastal buoy was above the 99$^{th}$ percentile. Since this information is now available in the new Table 3 (following the Reviewer-2´s suggestion), we have rephrased the sentence in the following way:

"ii) a peak of wind speed (> 15 m·s$^{-1}$) over the entire Alborán Sea, where easterlies blew strongly along both sides of the Strait of Gibraltar (Annex 2). Only the event E6 showed a slightly different structure (Annex 2f), with moderately strong winds (13-15 m·s$^{-1}$) blowing from the NE and massively affecting the entire western Mediterranean Sea. In terms of persistence, intense winds steadily affected the study area for 1-2.5 days, except in the case of E1 and E6 events where the duration was shorter (14-16 h), as derived indirectly from the time that the SWHo consecutively exceeded the P99 (Table 3)."

[Comment 30] Lines 211-212: "… a 6-week period between late February and early April (Figure 1, d)." This is only evident in the table inserted in figure 1 and not in figure 1d.

**OK, 100% accomplished**. A new panel has been added to Figure 2 (Figure 2d) where the temporal distribution of extreme wave events for 7 different time periods is exhibited. All the time periods selected were evenly distributed (50-day length) except period 6 that is 65-day length. The following sentence was added to section 3 ("Methodology"):

"The annual cycle was split into six evenly spaced 50-day intervals and a longer 65-day summertime interval that did not negatively impact on the consistency of the percentages of occurrence obtained as extreme wave events during summer remained marginal regardless of the interval length selected."

As shown below, it seems quite clear that the vast majority of wave episodes with SWH and mean period above P99 took place during Period 2, comprised between 20 February and 10 April.

[Figure]

The sentence has been rephrased: "Furthermore, this common atmospheric configuration seems to predominantly feature during the same stage of the year, a 50-day period between late February and early April (Figure 2d)".

[Comment 31] Line 214: And why is that? What are the conditions leading to the development of the low-pressure system over N Africa and the high latitude position of the Azores High? Relation with winter regimes (e.g., North Atlantic Oscillation winter)? – This could be brought to the discussion/conclusion section

**OK, 100% accomplished**. A paragraph has been added to Section 4 ('Results') and also in Section 5 ('Conclusions') related to [Comment 52] from Reviewer-2. Please, find below a comprehensive explanation:

Some weather extremes occurring in Europe have been connected with a particular atmospheric flow pattern, known as atmospheric blocking. Blocking episodes have been long acknowledged as a large-scale disturbance (high-pressure systems at the surface) that persist for a long period of time in the middle and high latitude flows (Lupo, 2021). Depending on their location, long-lasting blocking systems also may lead to a shift in the storm track, which influences the occurrence of wind and precipitation anomalies in Europe. Due to these multifaceted linkages, compound events are often observed in conjunction with blocking conditions (Kautz et al., 2021).

There are 3 main types of atmospheric blocks affecting western Europe, as shown in the figure below (extracted from Sousa et al., 2021):

iv) The Omega block: a relatively widespread high-pressure area is observed between the midlatitudes and the subtropics, with low pressure systems on its western and eastern flanks (Figure 1a).

v) The "hybrid" Rex block: during the transition from an Omega to a Rex (pure) shape, the blocking can temporarily exhibit "mixed" patterns (Figure 1b).

vi) The "canonical" Rex block: a north–south dipole pattern characterized by two adjacent (northern) high and (southern) low pressure systems in upper atmospheric levels (Rex, 1950).

[Figure]

Served as example, we have plotted again the sea level pressure maps for some specific extreme events but expanding the spatial coverage westwards. As it can be observed, E1, E2 and E7 correspond to hybrid REX patterns, while E5 event seems to be closer to the pure Rex shape with a clear north–south dipole. E3, E4 and E6 (not shown) appear to resemblance, to some extent, an omega block, whereas E4 and E6 (not shown).

[Figure]

In general, blockings in the Northern Hemisphere tend to occur more frequently in winter and spring than in the other seasons. Particularly, eastern North Atlantic blocks are more common in the period from winter to spring (Kautz et al., 2022), in line with previous blocking climatologies (Barriopedro et al., 2006). Previous works have explored the dynamical links between blocking and the Nort Atlantic Oscillation (NAO), which is the leading mode of atmospheric circulation variability over the Euro-Atlantic sector and is characterized by a seesaw of atmospheric mass between the Iceland Low and the Azores High (e.g., Hurrell and Deser 2009). The NAO appeared as the leading variability pattern during winter, accounting for the 45% of the blocking frequency variance (Barriopedro et al., 2006).

References:

Barriopedro, D., R. García-Herrera, A. R. Lupo, and E. Hernández, 2006: A climatology of Northern Hemisphere blocking. J. Climate, 19, 1042–1063, https://doi.org/10.1175/JCLI3678.1

Hurrell, J.W. and Deser, C. North Atlantic climate variability: The role of the North Atlantic Oscillation. J. Mar. Syst. 78: 28–41. doi: 10.1016/j.jmarsys.2009.11.002, 2009.

Kautz, L.A., Martius, O., Pfahl, S., Pinto, J.G., Ramos, A.M., Sousa, P.M. and Woollings, T. Atmospheric blocking and weather extremes over the Euro-Atlantic sector − a review. Weather Clim. Dynam., 3, 305–336, doi:10.5194/wcd-3-305-2022, 2022.

Lupo AR. Atmospheric blocking events: a review. Ann N Y Acad Sci. 2021 Nov;1504(1):5-24. doi: 10.1111/nyas.14557. Epub 2020 Dec 31. PMID: 33382135.

Rex, D. F. (1950). Blocking action in the middle troposphere and its effect upon regional climate. Tellus 2, 275–301. doi: 10.1111/j.2153-3490.1950.tb00339.x

Sousa, P. M., D. Barriopedro, R. García-Herrera, T. Woollings, and R. M. Trigo: A New Combined Detection Algorithm for Blocking and Subtropical Ridges. J. Climate, 34, 7735–7758, doi:10.1175/JCLI-D-20-0658.1, 2021.

[Comment 32] Lines 215-216: This statement is not correct as can readily be seem comparing figure 2-a (event 7) with Annex1- a (events E1). Both figures show a comparable distance between the atmospheric pressure centres (perhaps event shorter for event E1) but very different SLP values associated with these centres. It is the (atmospheric) pressure gradient, or SLP gradient, that was the relevant aspect in having a much strong wind fields (and wave generation mechanism) in E7 by regard to E1. Note that in lines 222-224, when you describe the E1 event, you are stating that the relevant factor is the SLP gradient and not the separation between the two main pressure systems.

**OK, 100% accomplished**. The sentence has been rephrased to accurately describe what is shown in Figure 2a:

"The primary factors that jointly triggered the record-breaking E7 wave storm were the short distance (1400 km) between the two main pressure systems along with the relatively deep (below 1000 hPa) system of low pressures over the Gulf of Cádiz (Figure 2a)."

[Comment 33] Lines 217: "…leading to very strong, persistent easterlies…", how long is "persistent" here? Can you provide a quantification (how many days?)

**OK, 100% accomplished**. We can provide an indirect quantification: almost two days, as the $99^{th}$ percentile of $SWH_o$ was exceeded during 37 consecutive hours. Indeed, this could only happen if strong winds blow persistently. Since there was no wind sensor in the study area, we could not quantify the persistence directly.

In order to better clarify this point, the following sentence was added:

"In terms of persistence, intense winds steadily affected the study area for 1-2.5 days, except in the case of E1 and E6 events where the duration was shorter (14-16 h), as derived indirectly from the time that the $SWH_o$ consecutively exceeded the P99 (Table 3)."

[Comment 34] Line 219: "… was abruptly exceeded during 42 consecutive hours (Figure 1, d).". We cannot infer from Figure 1, d, how long did SWH was above P99. Perhaps better: "… was abruptly exceeded (Figure 1, d) during 42 consecutive hours.".

**OK, 100% accomplished**. The sentence has been reformulated and moved to section 4.1 ('Extreme wave analysis'):

"In terms of storm duration (Table 3), defined as the number of consecutive hours above the P99 of $SWH_o$ (set to 3.01 m), E1 and E6 were significantly shorter (<20 h) than long-lasting E2 and E5

events (>50 h). The duration of E3 and E6 (27-31 h) events can be considered similar to E7 (37 h).”

[Comment 35] Lines 220-221: SLP gradient units and consistency with SI units along the text, if required.

**OK, 100% accomplished**. According to the journal´s guidelines, it is correct.

[Comment 36] Line 232: “According to the spectra content of 2Hz data …” change to “According to the spectra content of 2Hz data (not shown) …”

**OK, 100% accomplished**. The sentence has been reformulated and moved to Section 3 (‘Methodology’):

“[...] spectra of the 2 Hz data (not shown), generated to identify energetic sea level variability inside the port, were dominated by energy in the IG band during these storms.”

[Comment 37] Lines 232-233: As is mentioned in the caption and label of Figure 3, this figure presents the hourly time series of maximum sea level height in the IG band. So what is the sense of referring that “these oscillations are highly dominated by the IG band energy during the events.”? I guess that what you are saying is that the spectra of 2Hz data show that port variability, during those extreme events, is dominated by the IG band? Rephrase this paragraph so that this becomes clearer.

**OK, 100% accomplished**. The sentence has been reformulated and moved to Section 3 (‘Methodology’):

“[...] spectra of the 2 Hz data (not shown), generated to identify energetic sea level variability inside the port, were dominated by energy in the IG band during these storms.”

[Comment 38] Lines 235-236: “… records of sea level oscillations (30s-1h) height…” should be changed to “… records of sea level height oscillations (30s-1h)…”

**OK, 100% accomplished**.

[Comment 39] Lines 235-238: These two paragraphs are confuse. It is not from Table 6 that we see that sea level height oscillations are below or above this or that level. This is derived from the analysis of Figure 3a and this figure is not indicated in the text. Table 6 is providing the framework that relates these observations with the thresholds used to guide port management.

**OK, 100% accomplished**. Apologies for the confusion. The paragraph referred to the IG significant height (which were not shown in the first version of the manuscript), while Figure 3a was showing IG maximum height. Now both parameters are exposed in Figure 3a (see below), following Reviewer1´s suggestion (“*It is not possible to see all these results in Table 6. Could you display them graphically?*”). Furthermore, since the thresholds for port management are also indicated in this panel, Table 6 has been removed from the document to make easier for the reader to follow the narrative.

[Figure]

[Comment 40] Line 243: "…the analysis of 20-m timeseries of agitation…" should be corrected to "…the analysis of 20 min averaged time series of agitation…"

**OK, 100% accomplished**. The sentence has been reformulated:

"The analysis hourly time series of agitation provided by Melilla tide gauge revealed that the seven extreme events exceeded the P99.9 threshold (0.56 m, Figure 3a)."

Additionally, in section 2.1.2 (Melilla port tide gauge) we have clarified:

"Finally, 20-minute estimations of $HFSLO_{max}$, $HFSLO_{13}$, IG wave energy (IGE) and agitation were subsampled at hourly intervals (Table 2) and examined to assess the impact of extreme wave storms inside the harbour."

[Comment 41] Lines 246-248:

(a) The definition of agitation was given in line 156 and, based on the Stockdon et al. 2006) work, it relates to the IG wave inshore the port that are forced by the waves offshore (measured by the Melilla buoy). Here it seems that we are using the term "agitation response" to design the sea level variability inside the port, which can be linked to the penetration of waves but also to astronomical tides or storm surge. These seem to be different concepts for the term "agitation".

**OK, 100% accomplished.** We are sorry, that was a typo. In L146 we pretended to give the definition of infragravity waves height. All the methodology section has been restructured to better clarify the concept of wave agitation response and thereby avoid any misunderstanding.

(b) Why not to include also wind forcing conditions together with astronomical tide and storm surge? Storm surge was (in the text) only associated with the isostatic response of sea surface to the atmospheric pressure. Besides this effect, the wind can force upwelling or downwelling responses along the coast, which are associated with sea level changes. In the case of the extreme events reported, the easterly winds would expectedly promote strong upwelling along the Melilla coastline, leading to a sea level fall near the coast.

**OK, 70% accomplished**. We did not include the wind forcing as there was not a wind sensor in the study area. Besides, we think that persistently strong easterly winds can hardly induce coastal upwelling in Melilla due to its specific coastal orientation. Conversely, strong and persistent southeasterly winds may induce coastal upwelling in Melilla.

[Figure]

Area where potential coastal downwelling could take place due to both intense along-shore easterly winds and Coriolis effect

Areas where potential coastal upwelling could take place due to both intense along-shore easterly winds and Coriolis effect (due to Earth's rotation), which combine to drive a near-surface layer of water offshore (Ekman transport). Such cross-shelf transport is compensated for by the vertical uplift of cold and enriched waters that fertilize the uppermost layer

Additionally, we explored daily maps of chlorophyll (CHL, not shown) and we could not find any peak of CHL concentration. Equally, we explored daily maps of sea surface temperature (SST, see below) provided by L4 satellite missions (and available through the Copernicus Marine Service catalogue) and we could not detect any SST drop in the Alborán Sea neither in the vicinity of Melilla harbour. Additionally, SST hourly timeseries from Melilla coastal buoy (attached below) did not reveal any significant decrease in SST: the SST values fluctuated between 16.2ºC and

15.7°C during the period comprised between the 31st of March and the 7th of April 2022. Therefore, we tend to conclude that there was not coastal upwelling in Melilla during the analysed extreme event.

[Figure]

In the new panel Figure 2e (shown below), the total sea level (blue line), the astronomical tide (red line) and the storm surge (non-tidal residual, green line) are now shown along with the port agitation (grey line) in order to evaluate the influence of the former factors on the port agitation.

[Figure]

Analogously, the new Annex 3 shows the same information for the six previous extreme events (E1-E6):

[Figure]

[Comment 42] Line 262: "… closest to the moored buoy…". Could you indicate at which distance from the buoy location?

**OK, 100% accomplished**. Since the MED reanalysis grid point is located at 2.916°W-35.354°N and the coastal buoy is located at 2.940°W–35.330°N, the distance between them is 3.45 km. Such distance has been indicated in the manuscript, in section 4.5 ('Trends in extreme wave climate'): "the MED reanalysis grid point (2.916°W, 35.354°N) closest to the moored buoy (located at a distance of 3450 m) was selected…"

[Comment 43] Line 272: But a similar maximum of P99 is also present in February for the buoy measurements.

**Not accomplished.** We would like to inform Reviewer-2 that panels b and c from Annex 3 have been removed in the new version of the manuscript, following Reviewer-1 suggestion. Therefore, the associated paragraph has been also deleted.

[Comment 44] Line 287: "… we select only March and July…". Although March corresponds in fact to the maximum P99 values of SWH for the time series 1993-2022, we saw that April presented the maximum values in P99 o SWH for the period 2009-2022 and for the time series of the wave buoy measurements. So why haven't you taken April as the representative month for the stormy season (and to show to the reader the spatial distribution of P50 and P99, in

Annex 4 a,c), given that this is the month for which the discussion in this paper is centred and also that this is the month that you selected to highlight the trends in figure 4a?

**OK, 100% accomplished.** Maps for March in Annex 4 have been replaced by maps for April (in the new version of the manuscript, this Figure is Annex 5).

[Comment 45] Line 298: "… of 2 cm year$^{-1}$ ", again, check if SI units should be used in the text.

**OK, 100% accomplished**. We have checked that cm is a valid unit.

[Comment 46] Lines 305-307: You state that although trend maps of P99 SWH for March and May showed relevant trends, they are not commented because they are located in areas far from the Melilla port area. But is it not that the goal of this exercise, to examine the conditions in the Alborán Sea to try to understand what conditions or change of conditions could be leading to the development of extreme storms such as the one measured in April 2022?

**OK, 100% accomplished**. Those maps have been commented. Notwithstanding, we would like to inform Reviewer-2 that Reviewer-1 requested to remove Annex 5: [Comment 38] "*Annex 5. Consider removing this annex because the most of the pixels show non-significant trend values*".

Within this context, the paragraph has been modified in the following way:

"The climate variability over the Alborán Sea was assessed by analysing the intra-annual variations in the extreme wave conditions (Figure 4). Monthly trend maps of P99 of SWHm were calculated for the period 1993-2022, revealing statistically significant changes in the vicinity of Melilla harbour for few specific months: while an increase of 2 cm·year-1 is observed for April (Figure 4a), a downward P99 trend of 1.5-2 cm·year-1 is detected for June (Figure 4b) and October (Figure 4c). The temporal trends for each month (Figure 4, d-f), computed over the subdomain surrounding Melilla harbour (black box in Figure 4), supported visually the previous statement: the trends were statistically significant at the 90% confidence interval for April, June, and October. By contrast, during both the second part of summer (July-September) and the transitional season (November - February), monthly maps of P99 trends (not shown) did not exhibit statistically significant values over the entire Alborán Sea. The trend map of P99 for March and May (not shown) exhibited large areas with positive trends and negative trends, respectively, but delimited over the eastern part (2ºW-1ºW) of the Alborán basin. "

General comment to 4.5 "Trends in Extreme Wave Climate":

[Comment 47] the analysis is presented in terms of the SWH percentiles. But the conditions affecting the Melilla port are associated not only with the offshore wave height but also with the offshore wave period and direction (for the particular environment of the port, this can be seen in the correlation diagrams that you present in figure 3 b,d). Was this analysis made for the 1993-2022 reanalysis? If so, can you introduce some of these results in the discussion in section "5. Conclusions"?

**OK, 100% accomplished**. No, such analysis was not conducted, although we agree with Reviewer-2 that offshore wave period and direction deserve a detailed exploration. We also must confess that the specific format of the Ocean State Report Special Issue (the length of the manuscript is limited to a maximum of 4 figures and 3000 words) did not provide room for deeper analysis of wave period. In this line of thought, we already described in Section 5 ('Conclusions'): "long-term historical changes in wave period and directionality are receiving increasing attention and should be further analysed to assess their specific impact on harbours operability".

In order to follow Reviewer-2´s suggestion and mitigate this shortcoming, several sentences have been introduced along the manuscript with the aim of emphasising the relevant role played by the wave period in the harbour agitation, especially during E7 extreme event:

Section 4.4 ('Sea state within the port'): "Equally, the highest values of agitation (above 1 m height) were associated with $T_m$ and $T_p$ values above 7 s and 10 s, respectively. It seems reasonable to deduce that the record-breaking harbour agitation (1.41 m) registered during E7 event was caused by the combined effect of unprecedented values of $SWH_o$ (7.32 m) and $T_m$ (9.42 s) in tandem with a very high value of $T_p$ (10.75 s) and a $MWD_o$ (55º) comprised within the aforementioned predominant sector (50º-70º). This clearly shows that compound events (i.e., multiple extreme events that occur simultaneously or in close sequence) are of particular concern for harbour operability, as their individual effects may interact synergistically."

Section 5 ('Conclusions'): "Likewise, offshore wave period also plays a primary role in the modulation of harbour agitation, as derived from equation 4 and the results exposed in Figure 3d. As a consequence, any potential increase in both wave period and SWH could lead to the so-called compound extreme events, which are considered to be a major risk of climate change since they can cause more significant damage than individual extreme events (Velpuri et al., 2023)."

Reference:

Velpuri, M., Das, J. and Umamahesh, N.V. Spatio-temporal compounding of connected extreme events: Projection and hotspot identification, Environmental Research, 235, 116615, doi:10.1016/j.envres.2023.116615, 2023.

[Comment 48] Lines 322-231: This block of text is reproducing the information that you already provided in section 4.

**OK, 100% accomplished**. This piece of text has been rephrased and shortened to avoid any redundancy with the text provided in section 4.

[Comment 49] Line 324: "…derived from the hourly time series….", to become "…derived from the hourly time series for the period 2008-2022…."

**OK, 100% accomplished**. The entire paragraph was rephrased so this sentence is no longer in the document.

[Comment 50] Line 341-342: "…this common atmospheric configuration seems to predominantly feature during the same stage of the year, a 6-weekperiod between late February and early April (Figure 1, d)". This conclusion is not derived from figure 1-d, which only show us that the extreme wave events occur during that period, without relating them to a specific atmospheric pattern. The conclusion can be derived from the analysis of figures 2a and Annex 1 a to f that link the events to the atmospheric patterns.

**OK, 100% accomplished**. Since a new bar diagram has been added as Figure 2d, the sentence has been rephrased as follows: "this common atmospheric configuration seems to predominantly feature during the same stage of the year, a 50-day period between late February and early April (Figure 2d)."

General comments to "5. Conclusion" section:

[Comment 51] Comment 1: The estimates of the return period are key to guide the design of coastal structures and the planning of port operations. Given such a huge importance, could you say something about how robust are the estimates of return period that you provided in section 4?

**OK, 100% accomplished**. An additional paragraph has been added to better underline the need to revise security protocols taking into account the updated, robust return periods:

*"Special attention should be focused on the thorough revision of security protocols and the implementation of mitigation plans within the harbour territory based on the updated return periods presented in this work. The design lifetime risk should be recalculated accordingly as coastal structures in the vicinity of the harbour must resist growing stresses during their lifespan and operations, such as wave overtopping, floodings or resonance, to name a few. While the current port layout configuration must be adapted to the increasing frequency and magnitude of these stressors, future maritime facilities at Melilla harbour should be wisely designed and constructed taking into account these outcomes in order to withstand extreme wave regimes imposed by the changing marine environment (Vanem et al., 2019). Albeit methodologically robust, the return periods exposed in this work are based on short (12-year) timeseries of quality-controlled in situ wave observations. Therefore, they should be further complemented with return periods computed by means of longer modelled time series from very a high-resolution wave reanalysis."*

[Comment 52] Comment 2: "Section 5. Conclusions" could accommodate a more extensive discussion about the conditions leading to the extreme events that affect the SW Mediterranean (in particular, the Melilla area) and about our perception regarding the future occurrence of storms such as the one of 4-5 April 2022.

The following paragraph has been added to Section 5:

*"Therefore, it might be deduced that large-scale atmospheric blocks leading to severe sea states in Melilla tend to be more probable during the winter-to-spring transition period. This outcome is in line with previous blocking climatologies for the eastern North Atlantic (Kautz et al., 2022; Barriopedro et al., 2006). In this context, previous works have also explored the dynamical links between blocking and the Nort Atlantic Oscillation (NAO), which is the leading mode of atmospheric circulation variability over the Euro-Atlantic sector and is characterized by a seesaw of atmospheric mass between the Iceland Low and the Azores High (e.g., Hurrell and Deser 2009). The NAO appeared as the leading variability pattern during winter, accounting for the 45% of the blocking frequency variance (Barriopedro et al., 2006)."*

You conducted an analysis of wave conditions (based on the SWH) for the global Alborán Sea, characterizing the mean conditions for the 1993-2022 period and the trends for each month. It would be interesting to put these results in a more global perspective, for example by comparing them with other studies that focus the larger area of the W Mediterranean Sea, such as the study of Amarouche, Akpinar and Semedo (2022) ("Wave storm events in the Western Mediterranean Sea over four decades", Ocean Modelling 170, 101933,2022.

**OK, 100% accomplished**. The discussion against previously reported results has been enriched with the inclusion of Amarouche et al. works (2021 and 2022), placing special emphasis on other areas of the western Mediterranean Sea such as the Gulf of Lyon. The following paragraph has been added into the end of Section 4.5:

" From a broader perspective focused on the entire western Mediterranean Sea, Barbariol et al. (2021) also documented a relevant positive trend (1.2 cm·year$^{-1}$) during winter in the Gulf of Lyon (denoted in Figure 1a) due to strong north-westerly Mistral winds. By contrast, Amarouche

et al. (2021) examined a 41-year (1979-2020) hindcast database and determined that the west coast of Gulf of Lyon was affected by a significant increasing trend for all seasons, with a considerable annual increase (4 cm·year$^{-1}$) of maximum values of $SWH_m$. More recently, Amarouche et al. (2022) demonstrated significant decadal increases in wave storm intensity and duration not only over the eastern part of the Alborán Sea but also in the Balearic basin. All these findings highlighted both the existence of an inter-seasonal variability of P99 of $SWH_m$ and the importance of multi-temporal scales analysis."

2.2 References

[Comment 53] All references indicated in the text and tables are reported in the reference list.

Correct the following references indicated in the list of references:

- "Chiggiato et al. 2023:" - line 435. Change to "Chiggiato, J., Artale, V.,de Madron, X. D., Schroeder, K.,Taupier-Letage. I., Velaoras, D., Vargas-Yáñez, M.:" (the data of the publication is correctly indicated in line 436, at the end of the reference). Done!

- "Fanti V., Ferreira, Ó., Kümmerer, V. et al." – line 472, not justified the use of et al. , change to "Fanti V., Ferreira, Ó., Kümmerer, V., Loureiro, C." Done!

- "Garcia-Valdecasas, J., Pérez Gómez, B., Molina, R. et al." – line 474 – not justified the use of et al., in order to maintain consistence with other references present in the table Corrected!

  (note: "Garrabou, J., Gómez-Gras, D., Medrano, A. et al."- line 476 – justified the use of et al. given the extensive list of authors). OK!

- "Eyring….., 2021: Human…." – line 466. Remove 2021, the date of publication appears correctly at the end of reference, in line 471. Done!

2.3 Tables

[Comment 54] Following the comments made in 2.1 above, a table indicating the different systems/models used, providing the geographical location covered by those systems/models (e.g. mooring position or model area of coverage), the time period covered by data and the spatial/time resolution of the data will be very helpful to the reader.

**OK, 100% accomplished**. A new Table 2 has been introduced in the manuscript the requested information is gathered.

[Comment 55] Table 5 Caption: Perhaps you could indicate which parameters of Weibull distribution corresponds to what (slope, shape, threshold).

**OK, 100% accomplished.** This has been clarified in old Table 5 (now Table 4)

[Comment 56] Table 6, 3$^{rd}$ line: negative vale "-0.15m" should be an error and correspond to "=15 m".

**OK, 100% accomplished.** Table 6 has been replaced by Figure 3a where this typo has been corrected.

[Comment 57] Table 6 caption, line 635. Indicate references for the thresholds presented or refer to the text.

**OK, 100% accomplished.** Table 6 has been replaced by Figure 3a. The caption associated to Figure 3a has been modified to indicate the references for the thresholds.

2.4 Figures

[Comment 58] Figure 1a: The change of colour scale from figure 1a to figure 1b can confuse the reader (although the colour scales are clearly indicated)

**OK, 100% accomplished.** We have changed the palette by a new one with a broader variety of colours so now both maps share a common scale to avoid any confusion to the reader.

[Comment 59] Figure 1b: Table inserted in figure:

b.1 Specific question: indicate clearly what is the period (mean period, peak period...?) and what is the direction (mean direction, peak direction...?) that are indicated in the table.

**OK, 100% accomplished**. We have clearly indicated that mean period, peak period and mean direction are the variables shown in the table.

[Comment 60] Figure 1b: General comment: This table would be better located in the "Tables" section. This would allow to include additional information that can be relevant (such as the return period for each event and the duration of SWH above P99) as proposed in the comments inserted in 2.1.

**OK, 100% accomplished**. A new Table 3 inserted in the manuscript with additional information.

Figure 2.

[Comment 61] (a) The vectors in figure 2b correspond to what? They should correspond to the wind at 10m vectors but it's confuse since the vectors lengths seem not to correspond to the wind speed colour scale. Note for example that, in the figure, vectors are longer offshore the NW Spain (offshore Cape Finisterre and Galician coast), where speeds are about 15 m/s, than in the Gulf of Cadiz or in the Alborán Sea, where speeds approach 20 m/s

**OK, 100% accomplished**. The vectors indeed represent the wind at 10 m height. In particular, we used a python function to plot the 2D field of arrows: **quiver**. Quiver is defined as follows:

**quiver**([X, Y], U, V,) where *X*, *Y* define the arrow locations, *U*, *V* define the arrow directions. As additional parameter, we can introduce "pivot" to define the part of the arrow that is anchored to the X, Y grid. The arrow rotates about this point. Pivot has these possible values: **{'tail', 'mid', 'middle', 'tip'},** where t**he default value is 'tail'.** That means that the arrow size is representative of the wind field at the tail of the arrow.

Further details can be found at:

https://matplotlib.org/stable/api/_as_gen/matplotlib.pyplot.quiver.html

Please find below the Figure 2b where the two subregions mentioned by Reviewer-2 (NW Spain and Alborán Sea) have been zoomed in order to better compare the arrows length. Please, keep in mind that the arrow length is representative of the grid point where the arrow tail is located.

As Reviewer-2 will check, arrows length in NW Spain (yellow-orange zones) are not longer than those represented in the Alborán Sea or Gulf of Cádiz (red and dark-red zones).

[Figure]

We understand Reviewer-2´s suspicion, which is motivated by the fact that in Figure 2b we only plotted one arrow of every four for the sake of readability. In ZOOM-1 and ZOOM-2 panels we plotted one arrow of every two so it is easier to notice the differences between arrows representing 13 m/s speeds (yellow zones) and those representing 18 m/s speed (red zones). We hope this explanation satisfies Reviewer-2 and we can keep Figure 2b in its current form. Otherwise, should we offer as possible solution the following approach: we could plot again the maps imposing the value '**mid**' for the aforementioned parameter **pivot**. We leave the final decision at Reviewer-2´s discretion.

**Update: we finally used 'mid' option for pivot parameter and recomputed all the maps affected (Figure 1 and Annex 3).**

[Comment 62] (b) Sea level pressure units: used the classical units used in meteorology (hPa), change if consistency with SI units along the text is required

**OK, 100% accomplished**. We have checked that Sea level pressure units can be expressed in hPa, it is accepted by the journal.

[Comment 63] Figure 2 Caption: "wind at 10m.." should become "… wind at 10m height".

**OK, 100% accomplished**. Corrected in Figure 2 caption.

[Comment 64] Annex 1: Sea level pressure units: used the classical units used in meteorology (hPa), change if consistency with SI units along the text is required.

**OK, 100% accomplished**. We have checked that Sea level pressure units can be expressed in hPa, it is accepted by the journal.

[Comment 65] Annex 2: The vectors in figure (a) to (f) correspond to what? They should represent the wind at 10m vectors but then the vector length does not correspond to the wind speed color scale. Note for example that, in figure a, the vectors along the W Moroccan coast are similar with the ones represented in the Alborán Sea although there is a strong difference in the speeds represented by the colors.

**OK, 100% accomplished**. The vectors indeed represent the wind at 10 m height. In particular, we used a python function to plot the 2D field of arrows: **quiver**. Quiver is defined as follows:

**quiver**([X, Y], U, V,) where *X*, *Y* define the arrow locations, *U*, *V* define the arrow directions. As additional parameter, we can introduce "pivot" to define the part of the arrow that is anchored to the X, Y grid. The arrow rotates about this point. Pivot has these possible values: **{'tail', 'mid', 'middle', 'tip'},** where t**he default value is 'tail'.** That means that the arrow size is representative of the wind field at the tail of the arrow.

Further details can be found at:

https://matplotlib.org/stable/api/_as_gen/matplotlib.pyplot.quiver.html

Please find below the Annex 2a where the two subregions mentioned by Reviewer-2 (Morocco coast and Alborán Sea) have been zoomed in order to better compare the arrows length. Please, keep in mind that the arrow length is representative of the grid point where the arrow tail is located.

As Reviewer-2 will check, arrows length in Morocco (yellow zones) are not longer than those represented in the Alborán Sea (red zones).

[Figure]

We understand reviewer-2´s suspicion, which is motivated by the fact that in Annex 2a we only plotted one arrow of every four for the sake of readability. In ZOOM-1 and ZOOM-2 panels we plotted one arrow of every two, so it is easier to notice the differences between arrows representing 13 m/s speeds (yellow zones) and those representing 18 m/s speed (red zones). We hope this explanation satisfies the Reviewer-2 and we can keep Annex 2a in its current form. Otherwise, should we offer as possible solution the following approach: we could plot again the maps imposing the value '**mid**' for the aforementioned parameter **pivot**. We leave the final decision at Reviewer-2´s discretion.

**Update: we finally used 'mid' option for pivot parameter and recomputed all the maps affected (Figure 1 and Annex 3).**

[Comment 66] Annex 3 Caption: please indicate to what corresponds the red and blue fits in figure 3a.

**OK, 100% accomplished**. The caption has been modified: "best linear fit (cyan line) of scatter plot between hourly estimations of significant wave heigh observed by Melilla coastal buoy ($SWH_o$) and modelled by MED reanalysis ($SWH_m$) in the grid point closest to the moored buoy for an 12-year period (2011-2022). The red line represents the result of perfect agreement with slope 1.0 and intercept 0. Statistical metrics are adhered in white box."

1. Technical Aspects

[Comment 67] Line 122: "… the wave climate Melilla area… "should read "… the wave climate affecting Melila area…" or equivalent.

**OK, 100% accomplished** (in section 4.5)

---

## Referee Report (RR1)

**Review of the article titled "Monitoring the record-breaking wave event in Melilla harbour (SW Mediterranean Sea)" by Lorente, P., et al. 2023**

Upon reviewing the revised version of your manuscript, it is evident that the quality has significantly improved due to several key revisions. The inclusion of new tables detailing the characteristics of the different datasets used is particularly noteworthy, enhancing the manuscript's clarity and the reader's ability to understand the scope of your analysis.

Furthermore, the rewrite of the abstract provides a clearer and more impactful summary of the research findings. The reorganization of the data section enhances the manuscript's logical flow, facilitating a better understanding of the methodologies employed. Additionally, the introduction of new figures offers a concise visual summary of the data, further elevating the manuscript's quality.

However, there is a specific aspect that requires attention to prevent potential misinterpretation. In Figure 2d, the presentation of event numbers near to the legend could be mistakenly interpreted as representing event durations. It is also unclear whether the percentage of time exceeding the 99th percentile is calculated considering the duration of each event. To clarify this, I recommend revising the figure's labeling or the x-axis configuration. Options for addressing the labeling issue include labeling the x-axis with actual time units (e.g., years) or adjusting the axis label to "Events," depending on which approach better suits the data's representation. The decision on how to implement this change is left to the authors' discretion. Furthermore, please specify in the figure caption whether the percentage of time exceeding the 99th percentile is based on the duration of each event.

These revisions have collectively contributed to a substantial enhancement of the manuscript's overall quality, making it suitable for publication.